# Using the climate feedback response method to quantify climate feedbacks in the middle atmosphere in WACCM

Maartje Sanne Kuilman[1], Qiong Zhang[2], Ming Cai[3], Qin Wen[1,4]

1. Department of Meteorology and Bolin Centre for Climate Research, Stockholm University, Stockholm, Sweden
2. Department of Physical Geography and Bolin Centre for Climate Research, Stockholm University, Stockholm, Sweden
3. Department of Earth, Ocean and Atmospheric Science, Florida State University, Tallahassee, Florida, USA
4. Laboratory for Climate and Ocean-Atmosphere Studies (LaCOAS), Department of Atmospheric and Oceanic Sciences, School of Physics, Peking University, Beijing, China

Corresponding author: Maartje Sanne Kuilman (maartje.kuilman@misu.su.se)

## Abstract

Over recent decades it has become clear that the middle atmosphere has a significant impact on surface and tropospheric climate. A better understanding of the middle atmosphere and how it reacts to the current increase of the concentration of carbon dioxide ($CO_2$) is therefore necessary. In this study, we investigate the response of the middle atmosphere to a doubling of the $CO_2$-concentration and the associated changes in sea surface temperatures (SSTs) using the Whole Atmosphere Community Climate Model (WACCM). We use the climate feedback response analysis method (CFRAM) to calculate the partial temperature changes due to an external forcing and climate feedbacks in the atmosphere. As this method has the unique feature of additivity, these partial temperature changes are linearly addable. In this study, we discuss the direct forcing of $CO_2$ and the effects of the ozone, water vapour, cloud, albedo and dynamical feedbacks.

As expected, our results show that the direct forcing of $CO_2$ cools the middle atmosphere. This cooling becomes stronger with increasing height: the cooling in the upper stratosphere is about three times as strong as the cooling in the lower stratosphere. The ozone feedback yields a radiative feedback that mitigates this cooling in most regions of the middle atmosphere. However, in the tropical lower stratosphere and in some regions of the mesosphere, the ozone feedback has a cooling effect. The increase in $CO_2$-concentration causes the dynamics to change. The temperature response due to this dynamical feedback is small in the global average, although there are large temperature changes due to this feedback locally. The temperature change in the lower stratosphere is influenced by the water vapour feedback and to a lesser degree by the cloud and albedo feedback. These feedbacks play no role in the upper stratosphere and the mesosphere. We find that the effects of the changed SSTs on the middle atmosphere are relatively small as compared to the effects of changing the $CO_2$. However, the changes in SSTs

are responsible for dynamical feedbacks that cause large temperature changes. Moreover, the temperature response to the water vapour feedback in the lower stratosphere is almost solely due to changes in the SSTs. As CFRAM has not been applied to the middle atmosphere in this way before, this study also serves to investigate the applicability as well as the limitations of this method. This work shows that CFRAM is a very powerful tool to study climate feedbacks in the middle atmosphere. However, it should be noted that there is a relatively large error term associated with the current method in the middle atmosphere, which can be for a large part be explained by the linearization in the method.

## 1. Introduction

The increase of concentration of carbon dioxide in the atmosphere forms a major perturbation to the climate system. It is commonly associated with lower-atmospheric warming. However, in the middle atmosphere, the increase of $CO_2$ leads to a cooling of this region instead. This cooling has been well documented and is found by both model studies and observations (e.g. *Manabe and Wetherald*, 1975; *Ramaswamy et al.*, 2001; *Beig et al.,* 2003).

The middle atmosphere is not only affected by the increase in $CO_2$-concentration, but also by the decrease in ozone-concentration. The depletion of ozone ($O_3$) also effects the temperature in the stratosphere and leads to a cooling (*Shine et al,* 2003). A better understanding of the effect of the increased $CO_2$-concentration on the middle atmosphere, will help to distinguish the effects of the changes $CO_2$- and $O_3$-concentration.

Another major motivation for this study is the emerging evidence that the middle atmosphere has an important influence on surface and tropospheric climate (*Shaw and Shepherd*, 2008). It has, for example, been shown that cold winters in Siberia are linked to changes in the stratospheric circulation (*Zhang et al.*, 2018).

*Nowack et al*. (2015) has found that there is an increase in global mean surface warming of about 1°C when the ozone is prescribed at pre-industrial levels, as compared with when it is evolving in response to an abrupt 4xCO$_2$ forcing. It should be noted that the exact importance of changes in ozone seems to be dependent on both the model and the scenario (*Nowack et al*., 2015) and is not found by all studies (*Marsh et al.*, 2016).

As the effect is found to be rather large in some studies, and absent in other, there is a need for a better understanding of the behaviour of the middle atmosphere in response to changing $CO_2$ conditions, as the ozone concentration is influenced by this. Ozone is an example of a climate feedback, a process that changes in response to a change in $CO_2$-concentration and in turn dampens or amplifies the climate response to the $CO_2$ perturbation.

These climate feedbacks are a challenging subject of study, as observed climate variations might not be in equilibrium, multiple processes are

operating at the same time and moreover the geographical structures and timescales of different forcings differ. However, feedbacks form a crucial part of understanding the response of the atmosphere to changes in the $CO_2$-concentration.

Various methods have been developed to study these feedbacks, such as the partial radiative perturbation (PRP) method, the online feedback suppression approach and the radiative kernel method (*Bony et al.,* 2006 and the references therein). These methods study the origin of the global climate sensitivity (*Soden and Held*, 2006; *Caldwell et al.*, 2016; *Rieger et al.*, 2017). The focus of these methods is on changes in the global mean surface temperature, global mean surface heat and global mean sensible heat fluxes (*Ramaswamy et al., 2019*).

These methods are powerful for this purpose; however, they are not suitable to explain temperature changes on spatially limited domains. They neglect non-radiative interactions between feedback processes and they only account for feedbacks that directly affect the radiation at the top of the atmosphere (TOA).

The climate feedback-response analysis method (CFRAM) is an alternative method which takes into account that the climate change is not only determined by the energy balance at the top of the atmosphere, but is also influenced by the energy flow within the Earth's system itself (*Cai and Lu, 2009, Lu and Cai*, 2009). The method is based on the energy balance in an atmosphere-surface column. It solves the linearized infrared radiation transfer model for the individual energy flux perturbations. This makes it possible to calculate the partial temperature changes due to an external forcing and the internal feedbacks in the atmosphere. It has the unique feature of additivity, such that these partial temperature changes are linearly addable.

As a practical diagnostic tool to analyse the role of various forcings and feedbacks, CFRAM has been used widely in climate change research on studying surface climate change (*Taylor et al.,* 2013; *Song and Zhang,* 2014; *Hu et al.,* 2017; *Zheng et al.,* 2019). CFRAM has been applied to study the middle atmosphere climate sensitivity as well (*Zhu et al.,* 2016). In their study, Zhu et al. *(2016)* have adapted CFRAM and applied it to both model output, as well as observations. The atmospheric responses during solar maximum and minimum were studied and it was found that the variation in solar flux forms the largest radiative component of the middle atmosphere temperature response.

In the present work, we apply CFRAM to climate sensitivity experiments performed with the Whole Atmosphere Community Climate Model (WACCM), which is a high-top global climate system model, including the full middle atmosphere chemistry.

We investigate the middle atmosphere response to $CO_2$-doubling. We acknowledge that such an idealized equilibrium simulation cannot reproduce the complexity of the atmosphere, in which the $CO_2$-concentration is changing

gradually. However, simulating a double $CO_2$-scenario still allows us to
identify robust feedback processes in the middle atmosphere.
There are two aspects of the middle atmosphere response to $CO_2$-doubling:
there is the effect of the changes in $CO_2$-concentration directly, as well as the
changes in sea surface temperature (SST) which are in itself caused by the
changes in $CO_2$-concentration. It is useful to investigate these aspects
separately, as former should be robust, while the effect of the changed SST
depends on the changes in tropospheric climate, which can be expected to
depend more on the model.
In this study, we investigate the effects of doubling the $CO_2$-concentration and
the accompanying sea surface temperature change on the temperature in the
middle atmosphere as compared to the pre-industrial state. We use CFRAM
to calculate the radiative contribution to the temperature change due to
changes in carbon dioxide directly as well as due to changes in ozone, water
vapour, albedo and clouds. We refer to the changes in ozone, water vapour,
albedo and clouds in response to changes in the $CO_2$-concentration as the
ozone, water vapour, albedo and cloud feedbacks.
The circulation in the middle atmosphere is driven by waves. Wave forcing
drives the temperatures in the middle atmosphere far away from radiative
equilibrium. In the mesosphere, there is a zonal forcing, which yields a
summer to winter transport. In the polar winter stratosphere, there is a strong
forcing that consists of rising motion in the tropics, poleward flow in the
stratosphere and sinking motion in the middle and high latitudes. This
circulation is referred to as the 'Brewer-Dobson circulation' (*Brewer,* 1949;
*Dobson*, 1956).
Dynamical effects make important contributions to the middle-atmosphere
energy budget, both through eddy heat flux divergence and through adiabatic
heating due to vertical motions. It is therefore important that we also consider
changes to the middle-atmosphere climate due to dynamics. We refer to this
as the 'dynamical feedback' (*Zhu et al*., 2016).
The main goal of this paper is to calculate the contribution to the temperature
change due to changes in carbon dioxide directly as well as due to changes in
ozone, water vapour, albedo, clouds and dynamics in the middle atmosphere
under a double $CO_2$-scenario using CFRAM. Our intention is not to give a
complete account of the exact mechanisms behind the changes in ozone,
water vapour, albedo, clouds and dynamics.
**2. The model and methods**
**2.1 Model description**
The Whole Atmosphere Community Model (WACCM) is a chemistry-climate
model, which spans the range of altitudes from the Earth's surface to about
140 km (*Marsh et al.,* 2013). The model consists of 66 vertical levels with
irregular vertical resolution, which ranges from ~1.1 km in the troposphere,
1.1–1.4 km in the lower stratosphere, 1.75 km at the stratosphere and 3.5 km
above 65 km. The horizontal resolution is 1.9° latitude by 2.5° longitude.
WACCM is a superset of the Community Atmospheric Model version 4
(CAM4) developed at the National Center for Atmospheric Research (NCAR).
Therefore, WACCM includes all the physical parameterizations of CAM4
(*Neale et al.,* 2013), and a well-resolved high-top middle atmosphere. The
orographic gravity wave (GW) parameterization is based on *McFarlane*
(1987). WACCM also includes parameterized non-orographic GWs, which are
generated by frontal systems and convection (*Richter et al.,* 2010). The
parameterization of non-orographic GW propagation is based on the
formulation by *Lindzen* (1981).
The chemistry in WACCM is based on version 3 of the Model for Ozone and
Related Chemical Tracers (MOZART3). This model represents chemical and
physical processes from the troposphere until the lower thermosphere.
(*Kinnison et al.,* 2007). In addition, WACCM simulates chemical heating,
molecular diffusion and ionization and gravity wave drag.
**2.2 Experimental set-up**
In this study, the F_1850 compset (component set) of the model is used, i.e.
the model assumes pre-industrial (PI) conditions. This compset simulates an
equilibrium state, which means that it runs a perpetual year 1850. Four
experiments have been performed for this study (see Table 1).
Experiment C1 is the control run, with the pre-industrial $CO_2$ concentration
(280 ppm) and forced with pre-industrial ocean surface conditions such as
sea surface temperatures and sea ice. These SSTs are generated from the
CMIP5 pre-industrial control simulation by the fully coupled Earth system
model CESM. The atmospheric component of CESM is the same as WACCM,
but does not include stratospheric chemistry (*Hurrell et al.,* 2013). The SSTs
might be slightly different when they would be generated using a model that
also includes atmospheric chemistry, however, this aspect is not considered
in this study.
Experiment S1 represents the experiment with the $CO_2$ concentration doubled
as compared to the pre-industrial state (560 ppm) and forced with the same
pre-industrial SSTs as in experiment C1. In WACCM, the $CO_2$-concentration
does not double everywhere in the atmosphere. Only the surface level $CO_2$
mixing ratio is doubled, and elsewhere in the atmosphere is calculated
according to WACCM's chemical model.
The compset used in this experiment and all the following ones is still F_1850,
which means that other radiatively and chemically active gases, such as
ozone, will change only because of the changes in the $CO_2$-concentration,
due to WACCM's interactive chemistry. This also means that the effects of
chlorofluorocarbons (CFCs) are not considered in our experiments, as
anthropogenic production of CFCs started later than 1850.
In experiment S2, we simulate the scenario, in which there is the SSTs forcing
from the coupled CESM for double $CO_2$ condition. This means that the sea
surface temperatures are higher than in the PI run, and there is less sea ice.
However, in this experiment the $CO_2$-concentration is kept at the pre-industrial
value of 280 ppm. S3 represents the experiment with the $CO_2$-concentration
in the atmosphere doubled to 560 ppm and the SSTs prescribed for the
double $CO_2$-climate. Experiment C1, S1, S2 and S3 will be also referred to
hereafter by PI, the simulation with high CO2, the simulation with high SSTs
and the simulation with high $CO_2$ and SSTs, respectively.
The experimental setup of this study is similar to the setup performed with the
Canadian Middle Atmosphere Model (CMAM) *by Fomichev et al*. (*2007*) and
with the Hamburg Model of the Neutral and Ionized Atmosphere
(HAMMONIA) by *Schmidt et al.* (2006). The HAMMONIA model is coupled to
the same chemical model as WACCM: MOZART3. The setup in their study is
similar, however, in their study, they double the $CO_2$-concentration from 360
ppm to 720 ppm, while in our study, we double from the pre-industrial level of
$CO_2$ (280 ppm).
Note that experiment S2 and S1 are not representing scenarios that could
happen in the real atmosphere. These experiments have been used to study
the effect of the SSTs separately. Experiment S3 doesn't take into account
other (anthropogenic) changes in the atmosphere not caused by changes in
the $CO_2$-concentration and the SSTs.
All the simulations are run for 50 years, of which the last 40 years are used for
analysis. In the all results shown, we have used the 40 year mean of our
model data.
Table 1. Set-up of the model experiments.

| Experiment | CO$_2$ | SSTs from CESM equilibrium run |
|---|---|---|
| C1 | PI | PI |
| S1 | Double | PI |
| S2 | PI | High |
| S3 | Double | High |

**2.3 Climate feedback-response analysis method (CFRAM)**
In this study, we aim to quantify the different climate feedbacks that may play
a role in the middle atmosphere in a double $CO_2$-climate. For this purpose, we
apply the climate feedback-response analysis method (CFRAM) (*Lu and Cai*,
264 2009).
As briefly discussed in the introduction, traditional methods to study climate
feedbacks are based on the energy balance at the top of the atmosphere
(TOA). This means that the only climate feedbacks that are taken into
consideration are those that affect the radiative balance at the TOA. However,
there are other thermodynamic and dynamical processes that do not directly
affect the TOA energy balance, while they do yield a temperature response in
the atmosphere.
Contrary to TOA-based methods, CFRAM considers all the radiative and non-
radiative feedbacks that result from the climate system due to response to an
external forcing. This means that CFRAM starts from a slightly different
definition of a feedback process. Note also that as the changes in temperature
are calculated simultaneously, the vertical mean temperature or lapse rate
feedback per definition do not exist in CFRAM.
Another advantage of CFRAM is that it allows for measuring the magnitude of
a certain feedback in units of temperature. We can actually calculate how
much of the temperature change is due to which process. The *'climate*
*response'* in the name of this method refers to the changes in temperature in
response to the climate forcings and climate feedbacks.
We refer to the Appendix for the complete formulation of CFRAM diagnostics
using outputs of WACCM. Based on the linear decomposition principle, we
can solve Eq. (A12) for each of the terms on its right-hand side. This yields
the partial temperature changes due to each specific process namely:

$\Delta T_{CO_2} = \left(\frac{\partial \vec{R}}{\partial \vec{T}}\right)^{-1} \Delta\left(\vec{S} - \vec{R}\right)_{CO_2}$                                         (1)

$\Delta T_{O_3} = \left(\frac{\partial \vec{R}}{\partial \vec{T}}\right)^{-1} \Delta\left(\vec{S} - \vec{R}\right)_{O_3}$                                          (2)

$\Delta T_{H_2O} = \left(\frac{\partial \vec{R}}{\partial \vec{T}}\right)^{-1} \Delta\left(\vec{S} - \vec{R}\right)_{H_2O}$                                       (3)

$\Delta T_{albedo} = \left(\frac{\partial \vec{R}}{\partial \vec{T}}\right)^{-1} \Delta\left(\vec{S} - \vec{R}\right)_{albedo}$                                   (4)

$\Delta T_{cloud} = \left(\frac{\partial \vec{R}}{\partial \vec{T}}\right)^{-1} \Delta\left(\vec{S} - \vec{R}\right)_{cloud}$                                    (5)

In which $\vec{R}$ represents the vertical profile of the net long-wave radiation
emitted by each layer in the atmosphere and by the surface. $\vec{S}$ is the vertical
profile of the solar radiation absorbed by each layer. The matrix $\left(\frac{\partial \vec{R}}{\partial \vec{T}}\right)$
is the Planck feedback matrix, in which the vertical profiles of the changes in
the divergence of radiative energy fluxes due to a temperature change are
represented. $\Delta T$ represents the temperature change.

The factors $\Delta\left(\vec{S} - \vec{R}\right)_{CO_2}$ , $\Delta\left(\vec{S} - \vec{R}\right)_{O_3}$, $\Delta\left(\vec{S} - \vec{R}\right)_{H_2O}$ , $\Delta\left(\vec{S} - \vec{R}\right)_{albedo}$ and
$\Delta\left(\vec{S} - \vec{R}\right)_{cloud}$ are calculated by inserting the output variables from WACCM in
the radiation code of CFRAM. Here, one takes the output variables from the
control run, apart from the variable that is related to the direct forcing or the
feedback. Table A1 in the Appendix shows which variables from the
perturbation runs have been inserted in the radiation code of CFRAM in order
to calculate $\Delta(\vec{S}-\vec{R})_{CO_2}$, $\Delta(\vec{S}-\vec{R})_{O_3}$, $\Delta(\vec{S}-\vec{R})_{H_2O}$, $\Delta(\vec{S}-\vec{R})_{albedo}$ and
$\Delta(\vec{S}-\vec{R})_{cloud}$ and eventually the associated temperature changes.


Similarly, to equations (1)-(5), we also calculate the temperature change due
to non-local thermal equilibrium (non-LTE) processes and the dynamical
feedback. We calculate the terms $\Delta(\vec{S}-\vec{R})_{non-LTE}$ and $\Delta dyn$ in Eq. (A4) and
(A7).

$$\Delta T_{non-LTE} = \left(\frac{\partial \vec{R}}{\partial \vec{T}}\right)^{-1} \Delta(\vec{S}-\vec{R})_{non-LTE} \qquad (6)$$
$$\Delta T_{dyn} = \left(\frac{\partial \vec{R}}{\partial \vec{T}}\right)^{-1} \Delta dyn \qquad (7)$$

The calculated partial temperature changes can be added, their sum being
equal to the total temperature change. It is important to note that this does not
mean that the individual processes are physically independent of each other.

$$\Delta T_{CFRAM} = + \Delta T_{O_3} + \Delta T_{H_2O} + \Delta T_{albedo} + \Delta T_{cloud} + \Delta T_{non-LTE} + \Delta T_{dyn}$$
$$(8)$$

The linearization done for equations (A9) and (A10) introduces an error
between the temperature difference as calculated by CFRAM and as seen in
the model output. Another source of error is that the radiation code of the
CFRAM calculations is not exactly equal to the radiation code of WACCM.

$$\Delta T_{CFRAM} = \Delta T_{WACCM} - \Delta T_{error} \qquad (9)$$

For more details on the CFRAM method, please refer to *Lu and Cai* (2009).

Note that the method used in this study differs from the Middle Atmosphere
Climate Feedback Response Analysis Method (MCFRAM) used by *Zhu et al.*
(2016). The major difference is that in this study, we perform the calculations
using the units of energy fluxes (Wm$^{-2}$) instead of converting to heating rates
(Ks$^{-1}$). In other words, we use Wm$^{-2}$ as the units of heating rates for the layer
between two adjacent vertical levels. Because the radiative heating rates are
the net radiative energy fluxes entering the layer, it is rather natural and
straightforward (i.e., without dividing the mass in the layer to convert it to units
of Ks$^{-1}$) to have the same units of heating rates (convergence) as the radiative
energy fluxes. Another difference is that our method is not applicable above
0.01 hPa (~80 km), while *Zhu et al.* (2016) added molecular thermal
conduction to the energy equation, to perform the calculations beyond the
mesopause.

**3. Temperature responses in a double CO$_2$ scenario**

As described in section 2.2, four experiments were performed with WACCM: a
simulation with pre-industrial conditions (experiment C1) , a simulation with
changed SSTs only (experiment S2), a simulation with only a changed CO$_2$-
concentration (experiment S1) and a final simulation with both changed SSTs
and $CO_2$-concentration (experiment S3).
Figure 1 shows the zonal mean temperature changes for the different
experiments with respect to the pre-industrial state, as modelled by WACCM.
The results reach a statistical significance of 95% for the whole middle
atmosphere domain in the experiments S3-C1 and S1-C1, and most of the
middle atmosphere for experiment S2-C1. For this figure, as well as for all the
results shown in this paper, we have used the 40 year mean of our data.
In line with what was shown in earlier studies (e.g. *Akmaev*, 2006; *Fomichev*
*et al.*, 2007), we observe that an increase in $CO_2$ causes a cooling in the
middle atmosphere with the exception of the cold summer upper mesosphere
region. We also observe that changing the SSTs alone, while leaving the
$CO_2$-concentration at the pre-industrial levels (Fig 1c and 1f) also yields
significant temperature changes over a large part of the middle atmosphere
and contributes to the observed warming in the cold summer mesopause
region.
As found previously by *Fomichev et al.* (2007) and *Schmidt et al.* (2006), we
find that the sum of the two separate temperature changes in the experiment
with changed $CO_2$ only and changed SSTs only (experiment S1 and S2) is
approximately equal to the changes observed in the combined simulation
(experiment S3). *Shepherd* (2008) has explained this phenomenon as follows:
climate change affects the middle atmosphere in two ways: either radiatively
through in situ changes associated with changes in $CO_2$ or dynamically
through changes in stratospheric wave forcing, which are primarily a result of
changing the SSTs (*Shepherd,* 2008). Even though the radiative and dynamic
processes are not independent, these processes are seen to be
approximately additive (*Sigmond et al.*, 2004, *Schmidt et al.,* 2006, *Fomichev*
*et al.,* 2007).

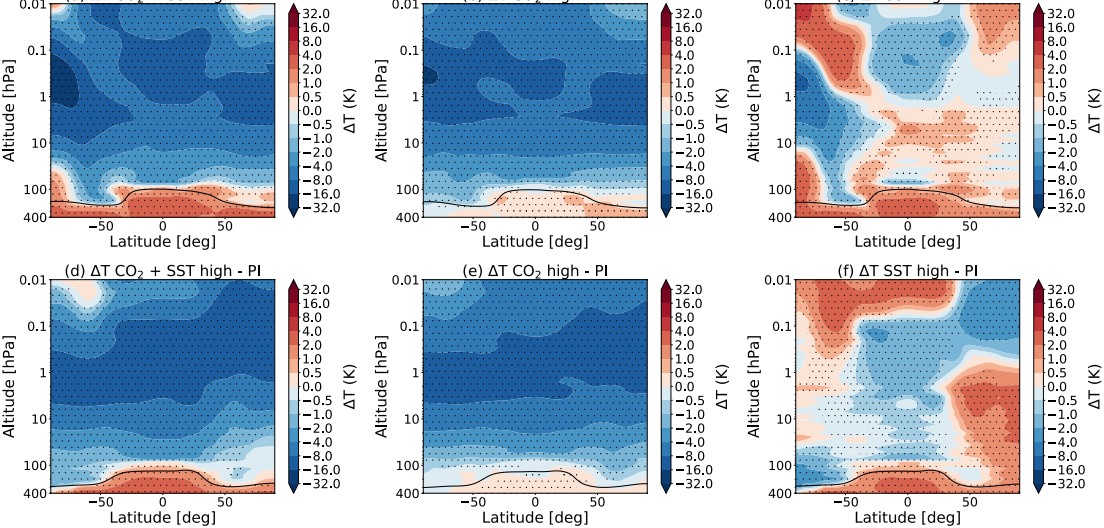

Figure 1: The total change in temperature in July (top) and January (bottom)
for (a,d) the simulation with high $CO_2$ and SSTs (S3), (b,e) the simulation with
high $CO_2$ (S1), (c,f) the simulation with high SSTs (S2), all as compared to the

pre-industrial control simulation (C1). The dotted regions indicate the regions where the data reaches a confidence level of 95%. The black line indicates the tropopause height for the experiments S3 (a,d), S1 (b,e) and S2 (c,f).

## 4. Meridional-vertical profiles of partial temperature changes

The CFRAM makes it possible to separate and estimate the temperature responses due to an external forcing and various climate feedbacks, such as ozone, water vapour, cloud, albedo and dynamical feedbacks. Note that for the ozone, water vapour, cloud and albedo feedback, we can only calculate the radiative part of the feedback. The response to dynamical changes is calculated in a separate term.

This can be understood as we use the Fu-Liou radiative transfer model (*Fu and Liou,* 1992, 1993) to do offline calculations of the total local thermal equilibrium (LTE) radiative heating rate perturbation fields between the control experiment C1 and one of the other three experiments (i.e, S1, or S2, or S3). We use the standard outputs of atmospheric compositions (e.g., $CO_2$ and $O_3$) and thermodynamic fields (e.g., pressure, temperature, water vapour, clouds, surface albedo) as well as partial LTE radiative heating rate perturbation fields due to perturbations in individual atmospheric composition or thermodynamic fields (e.g., the terms on the right hand side of (A.9) except the first term).

We use the difference between the offline calculation of the total LTE radiative heating rate perturbations and the original total LTE radiative heating rate perturbations derived directly from the standard WACCM outputs as the error term of our offline LTE radiative heating perturbations. We note that the standard WACCM output fields also include non-LTE radiative heating fields, but do not include non-radiative heating rates. Therefore, we use the sum of the total LTE radiative heating rate perturbations and non-LTE radiative heating fields derived from the standard WACCM output fields to infer non-radiative heating rate perturbations under the equilibrium condition, namely Eq. (A.8).

We should also note that, because we are using an atmosphere-only model, in our experiment, the external forcing is either the change in $CO_2$-concentration or the change in SSTs or both. In an atmosphere-ocean model (such as CESM) and, of course, in reality, the changes in sea surface temperature and sea ice distributions are responses to the changed $CO_2$-concentration.

In the following subsections 4.1-4.5, we will discuss the meridional-vertical profiles of the temperature responses to the direct forcing and the various feedbacks during July and January. In section 5, we will discuss regional and global means of partial temperature changes due to feedbacks.

### 4.1 Direct temperature response to $CO_2$

Figure 2 shows the zonal mean temperature change due to the increase in $CO_2$. We see that increasing $CO_2$ leads to a cooling almost everywhere in the

middle atmosphere, except at the high latitudes in the cold summer upper
mesosphere, where we see a warming instead. The higher the temperature,
the more cooling due to the increasing $CO_2$-concentration is found (*Shepherd,*
*2008*). The reason for this is that the outgoing longwave radiation strongly
depends on the Planck blackbody emission (*Zhu et al.,* 2016).

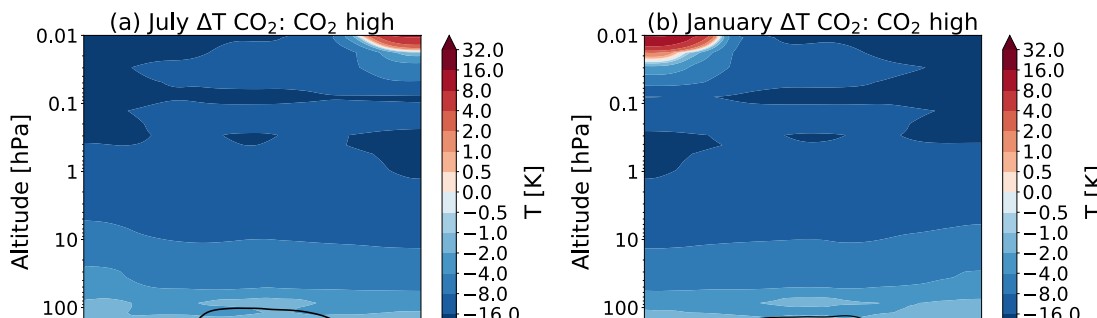

Figure 2: Partial temperature change due to the direct forcing of $CO_2$ for July
(top) and January (bottom) due to the doubling of the atmospheric $CO_2$-
concentration, as calculated by CFRAM, using experiment S1 and C1. The
black line indicates the tropopause height for the S1 run (with double $CO_2$-
concentration).
Changing the SSTs does not lead to a change in $CO_2$-concentration, therefore
the temperature response to changes in $CO_2$ is not present for the run with
only changed SST (Figures not shown).

**4.2 Ozone feedback**

Ozone plays a crucial role in the chemical and radiative budget of the middle
atmosphere. The distribution of ozone in the middle atmosphere is determined
by both chemical and dynamical processes. Most of the ozone production
takes place in the tropical stratosphere, as a result of photochemical
processes, which involve oxygen. Meridional circulation then transports ozone
to other parts of the middle atmosphere (*Langematz*, 2019). The production of
ozone is largely balanced by catalytic destruction cycles involving $NO_x$, $HO_x$
and $Cl_x$ radicals. $HO_x$ dominates ozone destruction in the mesosphere and
lower stratosphere, while $NO_x$ and $Cl_x$ dominate this process in the middle and
upper stratosphere (e.g. *Cariolle*, 1983).
Since the 1970s ozone in the middle atmosphere began to decline globally,
due to increased production of ozone depleting substances (ODSs) (*Brühl*
*and Crutzen*, 1988). The Montreal Protocol, adopted in 1987 to stop this
threat, eventually led to a slow recovery of the stratospheric ozone over the
recent two decades (*WMO*, 2018; *Langematz*, 2019).  In our study, we don't
consider the effect of anthropogenic ODSs since pre-industrial times
(*Langematz*, 2019).
In this study, we are interested in the temperature response to changes in
ozone concentration induced by the increased $CO_2$ concentration and/or the
changes in SST in WACCM. Under enhanced $CO_2$ concentrations, the ratio
between $O_3$ and O mixing ratios is generally shifted toward a higher
concentration of ozone, which is caused by the strong temperature
dependency of the ozone production reaction (O + $O_2$ + M $\rightarrow$ $O_3$ + M).
Fig. 3 shows the percentage changes in $O_3$-concentration when the $CO_2$-
concentration and/or the SSTs change. The results reach a statistical
significance of 95% for the whole middle atmosphere domain in the
experiments S3-C1 and S1-C1, and most of the middle atmosphere for
experiment S2-C1.
We find, as expected, that an increase in $CO_2$, leads to an increase of ozone
in most of the middle atmosphere. The increase of $O_3$ is about 20% around 2
hPa in the tropical region for experiment S3 with respect to C1. This
corresponds with what is seen by *Fomichev et al., (2007),* however they find
that the increase in ozone in January is a bit lower in this region (around 15%,
see their Figure 7).
There are some regions where the $O_3$-concentration is decreasing. In the
tropical lower stratosphere, a decrease of about 20% is seen, in the summer
polar mesosphere (around 0.01 hPa) ozone decreases by 3%, while in the
mesosphere (around 0.02 hPa), ozone decreases by over 30%. Fig. 3c and f
show that changing the SSTs also has a significant impact on the ozone
concentration. A complete account of the ozone changes is out of the scope
of this paper.

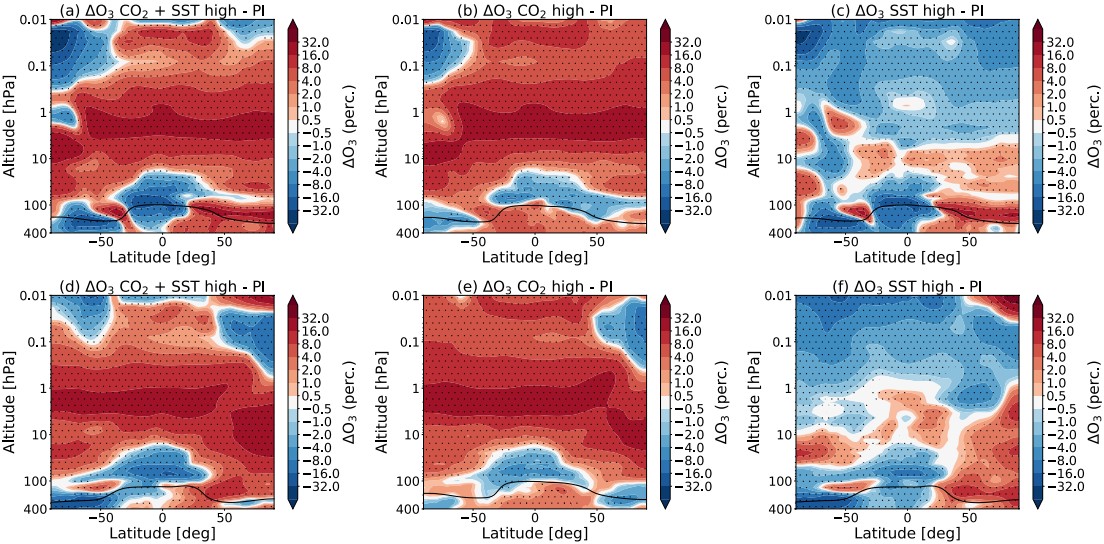

Figure 3: The percentage change in the zonal and monthly mean ozone
concentration for July (top) and January (bottom) due to (a,d) combined effect
of the $CO_2$ increase and SSTs changes (experiments S3 -C1), (b,e) the
doubling of the atmospheric $CO_2$-concentration (experiments S1 - C1) and the
(c,f) SSTs (experiments S2 - C1), as simulated by WACCM. The dotted
regions indicate the regions where the data reaches a confidence level of
95%. The tropopause height is indicated as in Fig. 1.
As we will discuss in the next section, an enhanced concentration of $CO_2$ also
leads to changes in the dynamics in the middle atmosphere. The stratospheric
Brewer-Dobson circulation is projected to strengthen, which would lead to an
increase in the poleward transport of ozone. We will also see that an increase
in $CO_2$-concentration leads to stronger summer pole-to-winter pole flow in the
mesosphere.
Figure 4 shows the percentage change in the zonal and monthly mean
concentration of Cl, NO, O, OH, $CH_3$, $NO_x$ and $N_2O$ in July due to the
combined effect of the $CO_2$ increase and SSTs changes (experiment S3 vs
C1), as simulated by WACCM. The patterns in January look similar (not
shown). These results reach a statistical significance of 95% for the whole
middle atmosphere domain.
We would like to point out that the changes in these constituents are only
brought about by the $CO_2$-concentration and/or the SSTs. We still use the
F_1850 compset and the only difference between the runs is the forcing in
$CO_2$ and SSTs. The changes in chemical constituents look very similar to
those found by *Schmidt et al.* (2006) who performed a similar experiment as
discussed in section 2.2, see their Figure 20. Note that Fig. 4 shows the
changes due to both the $CO_2$ increase and SSTs changes, while their Figure
20 shows the percentage changes due to the changes in $CO_2$-concentration
only and also only above 1 hPa.
As in *Schmidt et al.* (2006), we see an decrease in atomic oxygen (O) mixing
ratio at high summer latitudes around 0.01 hPa (see Fig. 4c), which results
from increased upwelling. This increase in O leads to a decrease in ozone in
this region. We also see decrease of ozone concentration in the winter polar
region around 0.1 hPa (approximately 65 km). This could be caused by an
increase of NO and for a small part by Cl mixing ratios, which result from a
stronger subsidence of NO and Cl rich air, as suggested in Schmidt et al.
*2006*. As stated before, complete discussion of the changes in ozone
concentration is out of the scope of this paper and the changes in other
constituents shown in Figure 4 are shown for reference only.

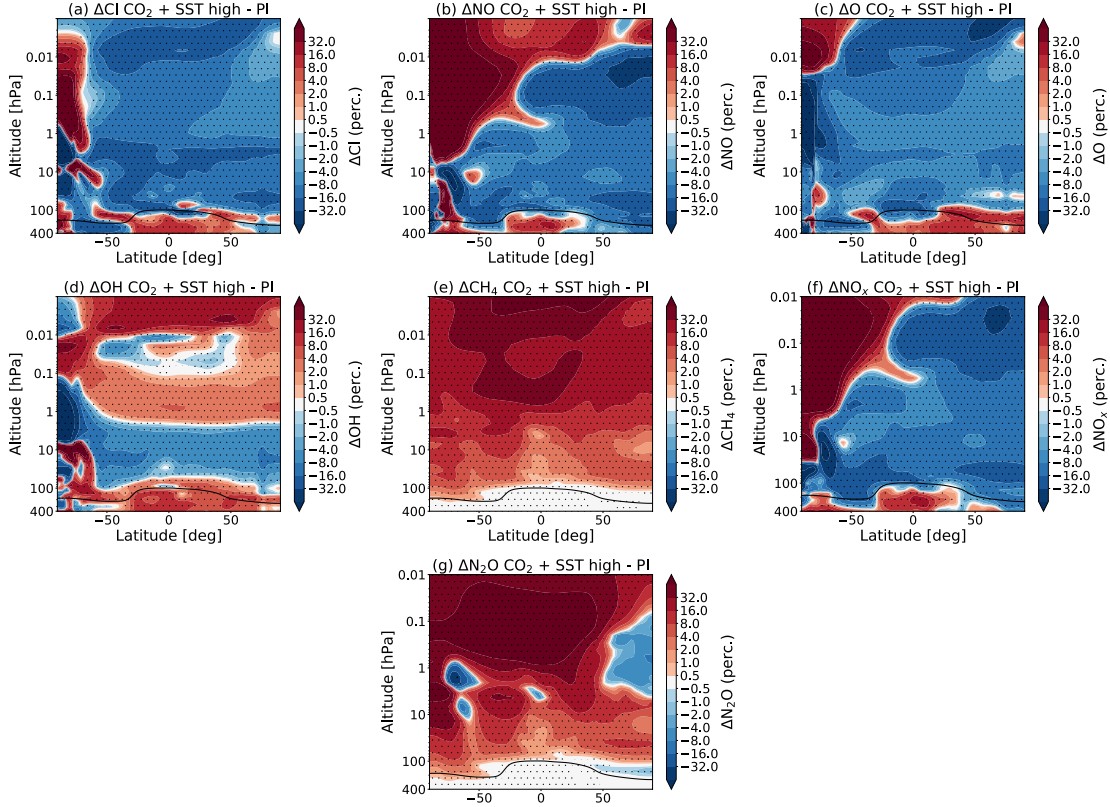

Figure 4: The percentage change in the zonal and monthly mean
concentration of Cl (a), NO (b), O (c), OH (d), CH$_4$ (e) and NO$_x$ (f) and N$_2$O (g)
in July due to the combined effect of the CO$_2$ increase and SSTs changes
(experiment (S3 vs C1), as simulated by WACCM. The dotted regions indicate
the regions where the data reaches a confidence level of 95%. The
tropopause height is indicated as in Fig. 1.

What is new in this study, is that we can calculate the temperature responses
due to the changes in ozone concentration. These temperature responses are
shown in Figure 5. It can be seen that there is a warming in the regions where
there is an increase of the O$_3$-concentration, while there is a cooling for the
regions with a decrease of the O$_3$-concentration. However, this is not the case
for the winter polar region, where there is no sunlight. Note that the
temperature responses to the changes in CO$_2$- and O$_3$- concentration behave
differently in this respect: the temperature responses due to the direct forcing
of CO$_2$ follow the temperature distribution quite closely, while the temperature
responses due to O$_3$ follow the ozone concentration, as also seen by Zhu et
al., *(2016).*

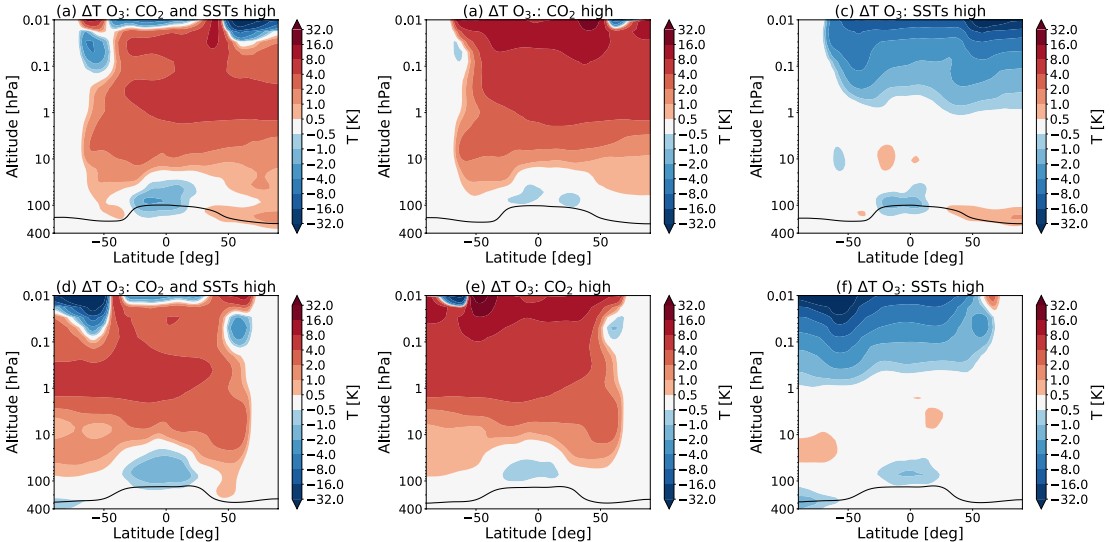

Figure 5: Partial temperature responses to changes in $O_3$-concentration, as calculated by CFRAM, in July (top) and January (bottom) due to the (a,d) combined effect of the $CO_2$ increase and SSTs changes (experiment S3), (b,e) the doubling of the atmospheric $CO_2$-concentration (experiment S1) and the (c,f) SSTs (experiment S2). The tropopause height is indicated as in Fig. 1.

## 4.2 Dynamical feedback

The zonal mean residual circulation forms an important component of the mass transport by the Brewer-Dobson circulation (BDC). It consists of a meridional ($\bar{v}^*$) and a vertical ($\bar{w}^*$) component as defined in the Transformed Eulerian Mean (TEM) framework. The residual circulation consists of a shallow branch, which controls the transport of air in the tropical lower stratosphere, as well as a deep branch in the mid-latitude upper stratosphere and mesosphere.

Both of these branches are driven by atmospheric waves. In the winter hemisphere, planetary Rossby waves propagate upwards into the stratosphere, where they break and deposit their momentum on the zonal mean flow, which in turns induces a meridional circulation. The two-cell structure in the lower stratosphere, which is present all-year round, is driven by synoptic scale waves. The circulation is also affected by orographic gravity wave drag in the stratosphere and by non-orographic gravity wave drag in the upper mesosphere (*Oberländer et al.*, 2013).

Most climate models show that the BDC and the upwelling in the equatorial region will speed up due to an increase in $CO_2$-concentration (*Butchart el al.,* 2010). It has been shown that the strengthening of the Brewer-Dobson circulation in the lower stratosphere is caused by changes in transient planetary and synoptic scale waves, while the upper stratospheric changes are due to changes in the propagation properties for gravity waves (*Oberländer et al.*, 2013).

It has been explained that the increased stratospheric resolved wave drag is
caused by an increase of the meridional temperature gradient in the
stratosphere, which leads to a strengthening of the upper flank of the
subtropical jets. This in turn shifts the critical layers for Rossby wave breaking
upward, which allows for more Rossby waves to reach the lower stratosphere,
where they break and deposit their momentum, enhancing the BDC
(*Shepherd and McLandress*, 2011)
The differences in the meridional component of the residual circulation ($\bar{v}^*$)
between the different simulations are shown in Fig. 6. These data are
averaged over the 40 years of data.  The results reach a statistical
significance of 95% almost the whole area above 1 hPa for the experiments
S1-C1, for the experiment S2-C1 the results reach a statistical significance of
95% in most of the area below this level. The experiments S3-C1 show the
largest region of statistical significance, apart from some regions below 1 hPa.
Figure 6b and 6e show that only doubling the $CO_2$-concentration leads to a
stronger pole-to-pole flow in the mesosphere. Changing the SSTs also leads
to changes in the residual circulation as can be seen in Fig. 6c and 6f.
Oberländer et al. (*2013*) have shown that the rising $CO_2$-concentration affects
the upper stratospheric layers, while the signals in the lower stratosphere are
almost completely due to changes in sea surface temperature.
The warmer sea surface temperatures affect the dynamics in the middle
atmosphere. It has for example been shown that higher SSTs in the tropics
leads to an amplification in deep convection, which enhances the generation
of quasi-stationary waves (*Deckert and Dameris, 2008*). Enhanced SSTs lead
to an enhanced dissipation of planetary waves, as well as an enhanced
dissipation of orographic and non-orographic waves in the upper stratosphere
(*Oberländer et al., 2013*).

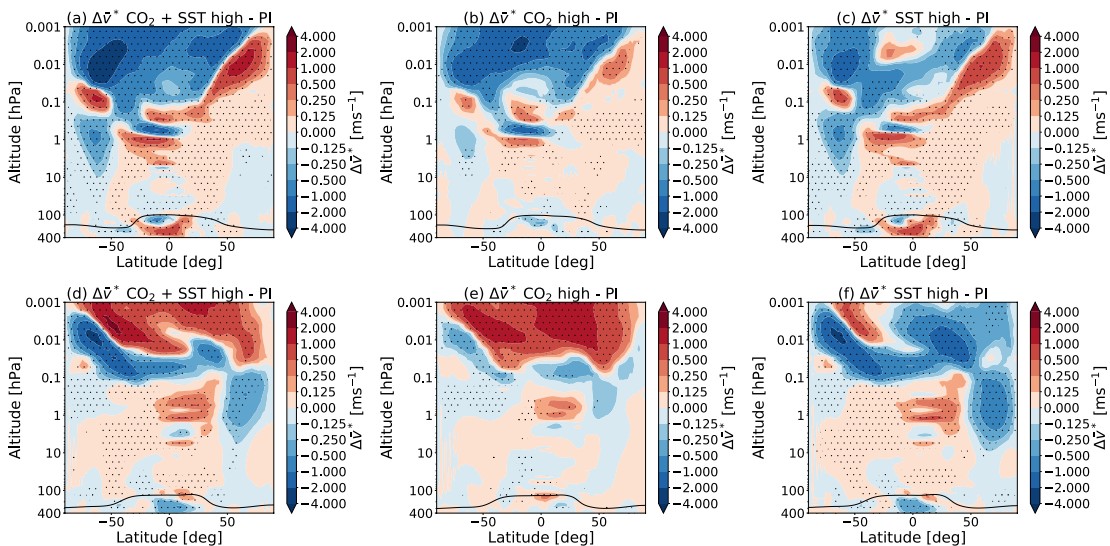

Figure 6: Changes in the zonal and monthly mean transformed Eulerian-mean
residual circulation horizontal velocity $\bar{v}^*$ for July (top) and January (bottom)
due to (a,d) combined effect of the $CO_2$ increase and SSTs changes
(experiments S3 -C1), (b,e) the doubling of the atmospheric $CO_2$-
concentration (experiments S1 - C1) and the (c,f) SSTs (experiments S2 -
C1), as simulated by WACCM. The dotted regions indicate the regions where
the data reaches a confidence level of 95%. The tropopause height is as
indicated in Fig. 1.
We are interested in the temperature responses due to the dynamical
feedbacks in the different experiments. These temperature responses are
shown in Figure 7. Figure 7b and 7e show that there is cooling in the summer
mesosphere, while there is warming in the winter mesosphere, which is
consistent with a stronger summer-to-winter pole flow.
Figure 7c and 7f show the temperature responses due to changes in the
SSTs. It is seen that there is mostly a warming in the summer mesosphere
and mostly a cooling in the winter hemisphere, which would weaken the effect
of the changed $CO_2$-concentration. Most of the temperature responses in the
lower stratosphere are caused by the changes in SSTs, as expected.

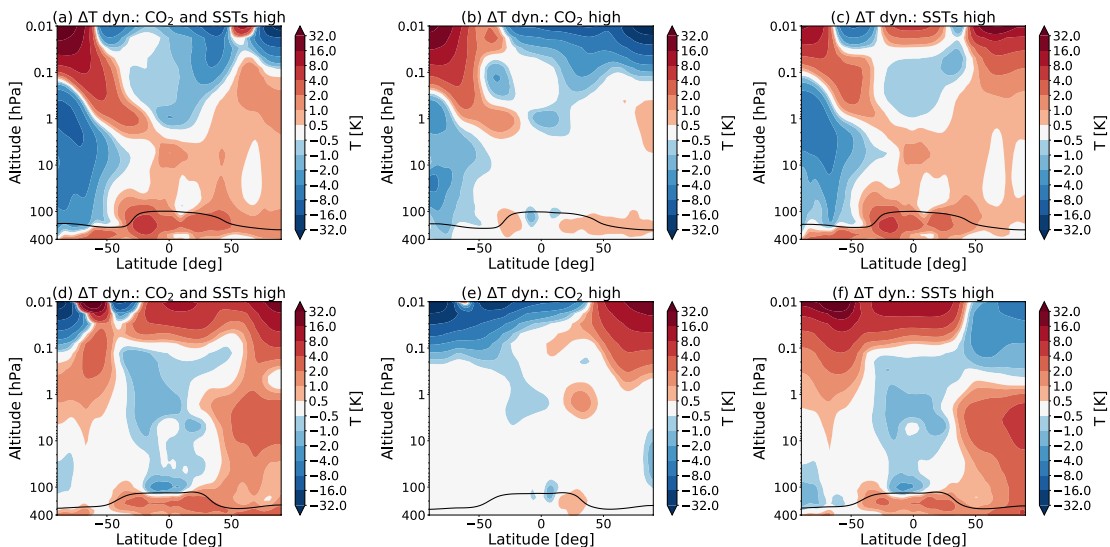


Figure 7: Partial temperature responses to changes in dynamics, as
calculated by CFRAM, in July (top) and January (bottom) due to the (a,d)
combined effect of the $CO_2$ increase and SSTs changes (experiment S3),
(b,e) the doubling of the atmospheric $CO_2$-concentration (experiment S1) and
the (c,f) SSTs (experiment S2). The tropopause height is indicated as in Fig.
652 1.

In summary, doubling the $CO_2$-concentration leads to a stronger pole-to-pole
flow in the mesosphere, which leads to cooling of the summer mesosphere
and a warming of the winter mesosphere. Changing the SSTs weakens this
effect, but leads to temperature changes in the stratosphere and lower
mesosphere.
**4.4 Water vapour feedback**

Figure 8 shows how the water vapour is changing in the middle atmosphere if
the $CO_2$-concentration is increased and/or the SSTs are changed with respect
to the pre-industrial control run. In WACCM, increasing the $CO_2$-concentration
alone leads to a decrease of water vapour in most of the middle atmosphere
(Fig. 8b and f). The results reach a statistical significance of 95% for the
whole middle atmosphere domain in the experiments S3-C1 and S2-C1, and
most of the middle atmosphere for experiment S1-C1, apart from the winter
hemisphere region around 0.1 hPa.
The amount of water vapour in the stratosphere is determined by transport
through the tropopause as well as by the oxidation of methane in the
stratosphere itself. The transport of the water vapour in the stratosphere is
mainly a function of the tropopause temperature (*Solomon et al.,* 2010). In
WACCM, we see a decrease in temperature in the tropical tropopause for the
double $CO_2$ experiment of about -0.25 K. The cold temperatures in the tropical
tropopause lead to a reduction of water vapour of between 2 and 8% due to
freeze-drying in this region.
It can be seen that using the SSTs from the doubled $CO_2$-climate leads to an
increase in water vapour almost everywhere in the middle atmosphere as
compared to PI (Fig. 8c and f). In WACCM, forcing with SSTs from a double
$CO_2$-climate is observed to lead to a higher and warmer tropopause, which
can explain this increase of water vapour. However, it should be noted that
models currently have a limited representation of the processes determining
the distribution and variability of lower stratospheric water vapour. Minimum
tropopause temperatures are not consistently reproduced by climate models
(*Solomon et al.,* 2010; *Riese et al.,* 2012). At the same time, observations are
not completely clear about whether there is a persistent positive correlation
between the SST and the stratospheric water vapour (*Solomon et al.*, 2010).

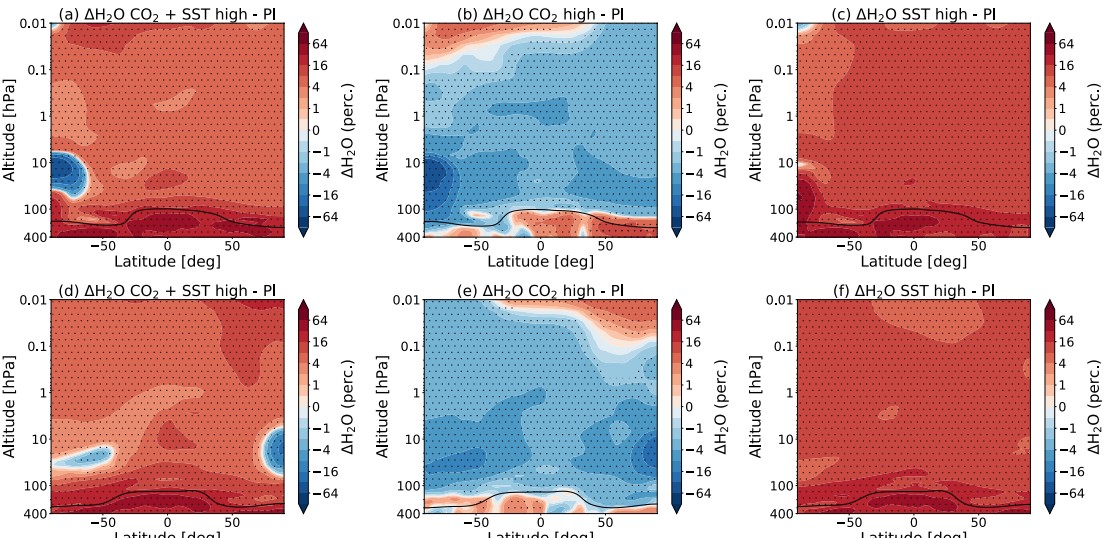

Figure 8: The percentage changes in the zonal and monthly mean water
vapour mixing ratio for July (top) and January (bottom) due to (a,d) combined
effect of the $CO_2$ increase and SSTs changes (experiments S3 - C1), (b,e) the
doubling of the atmospheric $CO_2$-concentration (experiments S1 - C1) and the
(c,f) SSTs (experiments S2 - C1), as simulated by WACCM. The dotted
regions indicate the regions where the data reaches a confidence level of
95%. The tropopause height is as indicated in Fig. 1.
Figure 9 shows the temperature responses due to the changes in water
vapour as calculated by CFRAM. It can be seen that in the regions where
there is an increase in the water vapour, there is a cooling, and vice versa.
This can be understood as increasing the water vapour in the middle
atmosphere leads to an increase in longwave emissions in the mid and far-
infrared by water vapour. This in turns leads to a cooling of the region.
Similarly, a decrease in water vapour leads to a warming of the region
(*Brasseur and Solomon,* 2005). Fig. 8 shows that above 1 hPa, there are also
large percentage changes in water vapour. However, the absolute
concentration of water vapour is small there, which explains why there is no
temperature response to these changes.

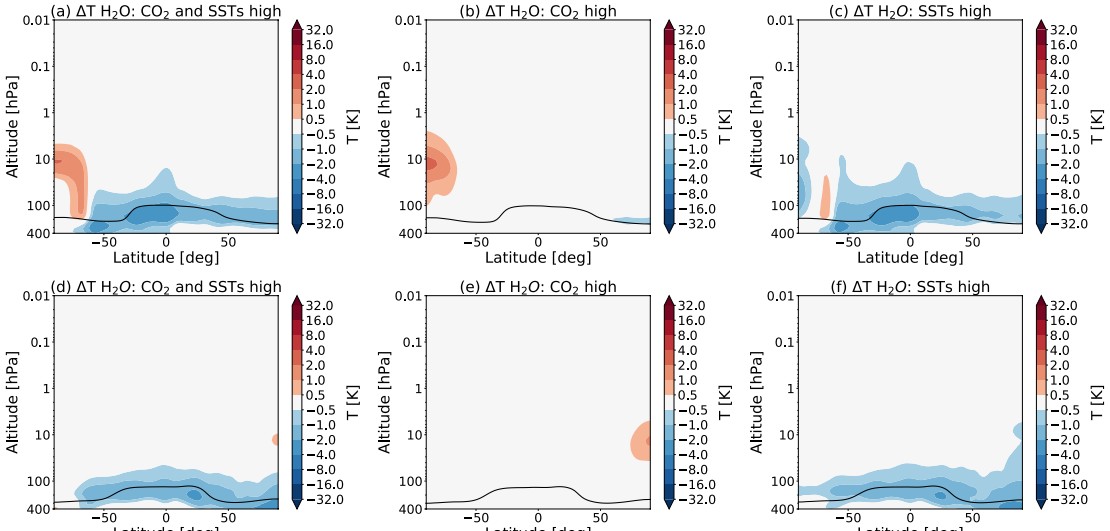

Figure 9: Partial temperature responses to changes in water vapour, as
calculated by CFRAM, in July (top) and January (bottom) due to the (a,d)
combined effect of the $CO_2$ increase and SSTs changes (experiment S3),
(b,e) the doubling of the atmospheric $CO_2$-concentration (experiment S1) and
the (c,f) SSTs (experiment S2). The tropopause height is indicated as in Fig.
717 1.
Water vapour plays a secondary but not negligible role in determining the
middle atmosphere climate sensitivity. In the lower stratosphere, $H_2O$
contributes considerably to the cooling in this region. Above 30 hPa**,** the water
vapour contribution to the energy budget is negligible, as also seen by
*Fomichev et al.* (2007).
**4.5 Cloud and albedo feedback**
Forcing the model with SSTs from the double $CO_2$-climate (as in experiment
S2 and S3) yields an overall increase in the cloud cover in the upper
troposphere, while this is not the case if one only increases the $CO_2$
concentration (as in experiment S1). Figure 10 shows the temperature

responses to changes in cloud (left) and albedo (right) in July (top) and
January (bottom) for experiment S2, as calculated by CFRAM.

Fig. 10 shows in the tropical region, there is a warming due to changes in
clouds, while there is a cooling at higher latitudes in July (see Figure 10a). In
January, the pattern looks slightly different (see Figure 10c). These
temperature changes are due to changes in the balance between the
increased reflected shortwave radiation and the decrease of outgoing
longwave radiation.

We also see an effect of the changes in surface albedo in the stratosphere
(see Figure 10b and d). The cooling in the summer polar stratosphere shown
in Figure 10b and d is due to radiative changes. We suggest that this cooling
is due to a decrease in surface albedo, which would lead to less shortwave
radiation being reflected. However, more research is needed.

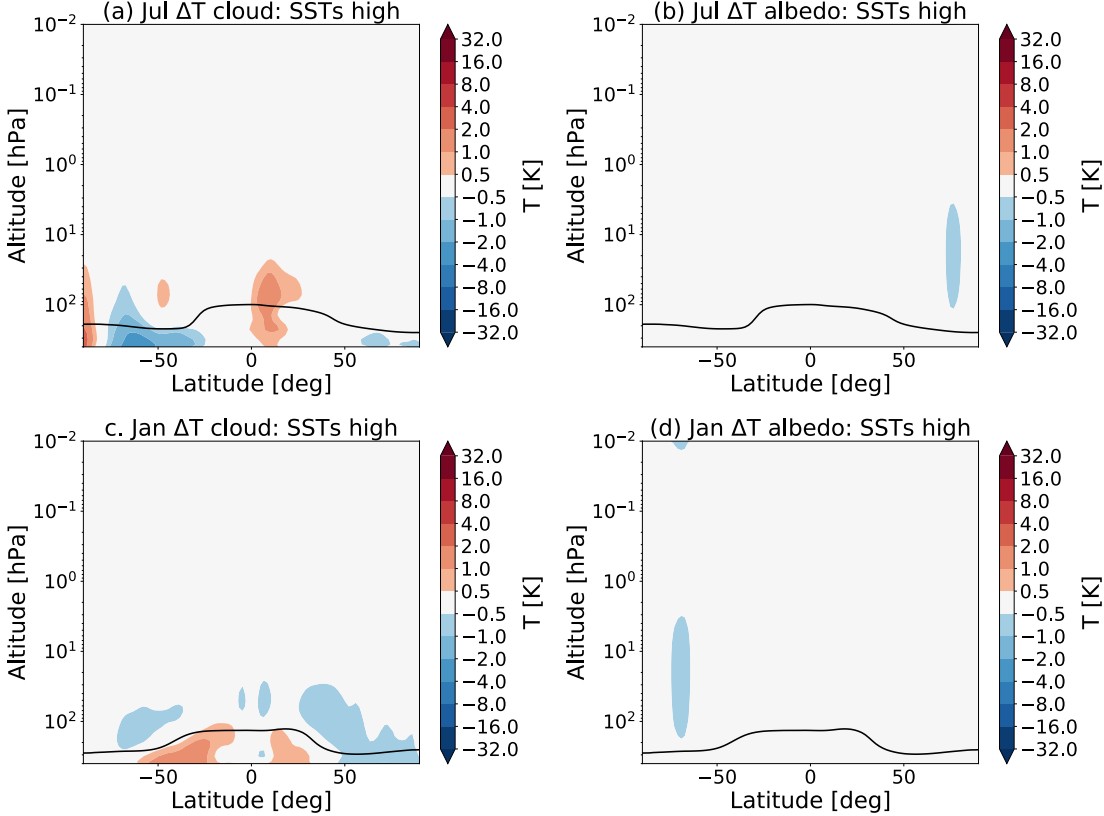

Figure 10: Partial temperature responses to changes in clouds (left) and
albedo (right), as calculated by CFRAM, in July (top) and January (bottom)
due to the SSTs (experiment S2). The tropopause height is indicated as in
Fig. 1.

Cloud and albedo feedbacks due to changes in clouds and surface albedo
play a crucial role in determining the tropospheric and surface climate
(*Boucher et al.,* 2013, *Royer et al.,* 1990). However, it is clear that these
feedbacks play only a very small role in the middle atmosphere temperature
response to the doubling of $CO_2$ and SSTs.

## 5. Regional and global means of partial temperature changes due to feedbacks

To study the relative importance of the different feedback processes globally we show the average change in global mean temperature for the lower stratosphere, the upper stratosphere and the mesosphere for the S3 experiment with the changed $CO_2$-concentration and changed SSTs in Figure 11. We also show the average change in temperature in the polar regions (90°S-70°S and 70°N-90°N), the tropics (20°S-20°N) for the lower and upper stratosphere and the mesosphere.

In order to calculate the lower stratospheric temperature changes, we take the average value of the temperature change from the tropopause up to 24 hPa. The pressure level of the tropopause is simulated in WACCM for each latitude and longitude, we use this pressure level to demarcate between the troposphere and stratosphere. We consider 24 hPa as a crude estimate for the boundary between the lower and upper stratosphere.

The tropopause is not exactly at the same pressure level in the perturbation experiments as compared to the pre-industrial control run (C1). We always take the tropopause of the perturbation experiment which is a bit higher at some latitudes, to make sure that we do not use values from the troposphere. We add the values for each latitude up and take the average. This average is not mass weighted. By calculating the average in this way, we can directly compare the vertical values in different regions of the atmosphere. The temperature changes in the upper stratosphere and in the mesosphere are calculated in the same way, but then for the altitudes 24 hPa-1 hPa and 1 hPa-0.01 hPa respectively.

Figure 11 shows the radiative feedbacks due to ozone, water vapour, clouds, albedo and the dynamical feedback, as well as the small contribution due to the Non-LTE processes in column 'NLTE', as calculated by CFRAM. The 'total'-column shows the temperature changes in WACCM, while the column 'error' shows the difference between temperature change in WACCM and the sum of the calculated temperature responses in CFRAM. Note that the range of values on the y-axis is not the same for the different subplots.

Figure 11 shows that the temperature change in the lower stratosphere due to the direct forcing of $CO_2$ is around 3 K in the global mean. There is a stronger cooling in the tropical region of about 4 K in July and 3.5 K in January. We also observe that there is a cooling of about 1 K due to ozone feedback in the tropical region while there is a slight warming taking place in the summer hemispheres in both January and July. We also see that the temperature change in the lower stratosphere is influenced by the water vapour feedback. There is a cooling of about 0.5 K in the lower stratosphere, apart from in the southern polar area. There is some small influence from the cloud and albedo feedback, which can be negative or positive (see also Fig. 9).

In the upper stratosphere, the cooling due to the direct forcing of $CO_2$ is with about 9 K in the global mean considerably stronger than in the lower

stratosphere. The cooling is stronger in the summer polar regions, where the
cooling due to the direct forcing of $CO_2$ reaches 11K. In the winter polar
region, this cooling is only about 8K.
The water vapour, cloud and albedo feedback play no role in the upper
stratosphere nor in the mesosphere. The ozone feedback results in the
positive partial temperature changes in the upper stratosphere, of about 2 K in
the global mean. The changes in ozone don't result in temperature changes in
the winter hemisphere, as discussed in section 4.2.
The picture in the mesosphere is similar as in the upper stratosphere. The
main difference is that the temperature changes are larger. The global
temperature change due to direct forcing of $CO_2$ is about 15 K. The $O_3$-
feedback results in a partial temperature changes of about 3 K in the
mesosphere in the global mean. The temperature change due to ozone in the
equatorial mesosphere is about 4 K, while the warming due to ozone in the
summer polar region is a bit smaller: around 3K.  Just like in the upper
stratosphere water vapour, cloud and albedo feedback play no role.
We see, that the ozone feedback generally yields a radiative feedback that
mitigates the cooling, which is due to the direct forcing of $CO_2$. This has been
suggested in earlier studies, such as *Jonsson et al., 2004, Dietmüller et al.,*
*2014*. With CFRAM, it is possible to quantify this effect and to compare it with
the effects of other feedbacks in the middle atmosphere. Note that no other
method before has been able to quantify how much of the temperature
change in the middle atmosphere is due to the different feedback processes.
The temperature response due to dynamical feedbacks is small in global
average: less than 1 K. This can be understood as waves generally do not
generate momentum and heat, but redistribute these instead (*Zhu et al.,*
2016). However, the local responses to dynamical changes in the high
latitudes are large, as we have seen in section 4.2. There are some very small
temperature responses due to non-LTE effects as well, which mostly
contribute to the temperature change in the mesosphere.
The error term is relatively large, as can be seen from the rightmost column in
Fig.11. This term shows the difference between temperature change in
WACCM and the sum of the calculated temperature responses in CFRAM
(see eq. 9 in section 2.3). In CFRAM, we assumed that the radiative
perturbations can be linearized by neglecting the higher order terms of each
thermodynamic feedback and the interactions between these feedbacks, this
yields an error.
Cai and Lu (*2009*) show that this error is larger in the middle atmosphere than
for similar calculations in the troposphere. In the middle atmosphere, the
density of the atmosphere is smaller, which leads to smaller numerical values
of the diagonal elements of the Planck feedback matrix. As a result, the linear
solution is very sensitive to forcing in the middle atmosphere. Another part of
the error is due to the fact that the radiative transfer model used in the offline
CFRAM calculations is different than the radiative transfer model used in the
climate simulations with WACCM.

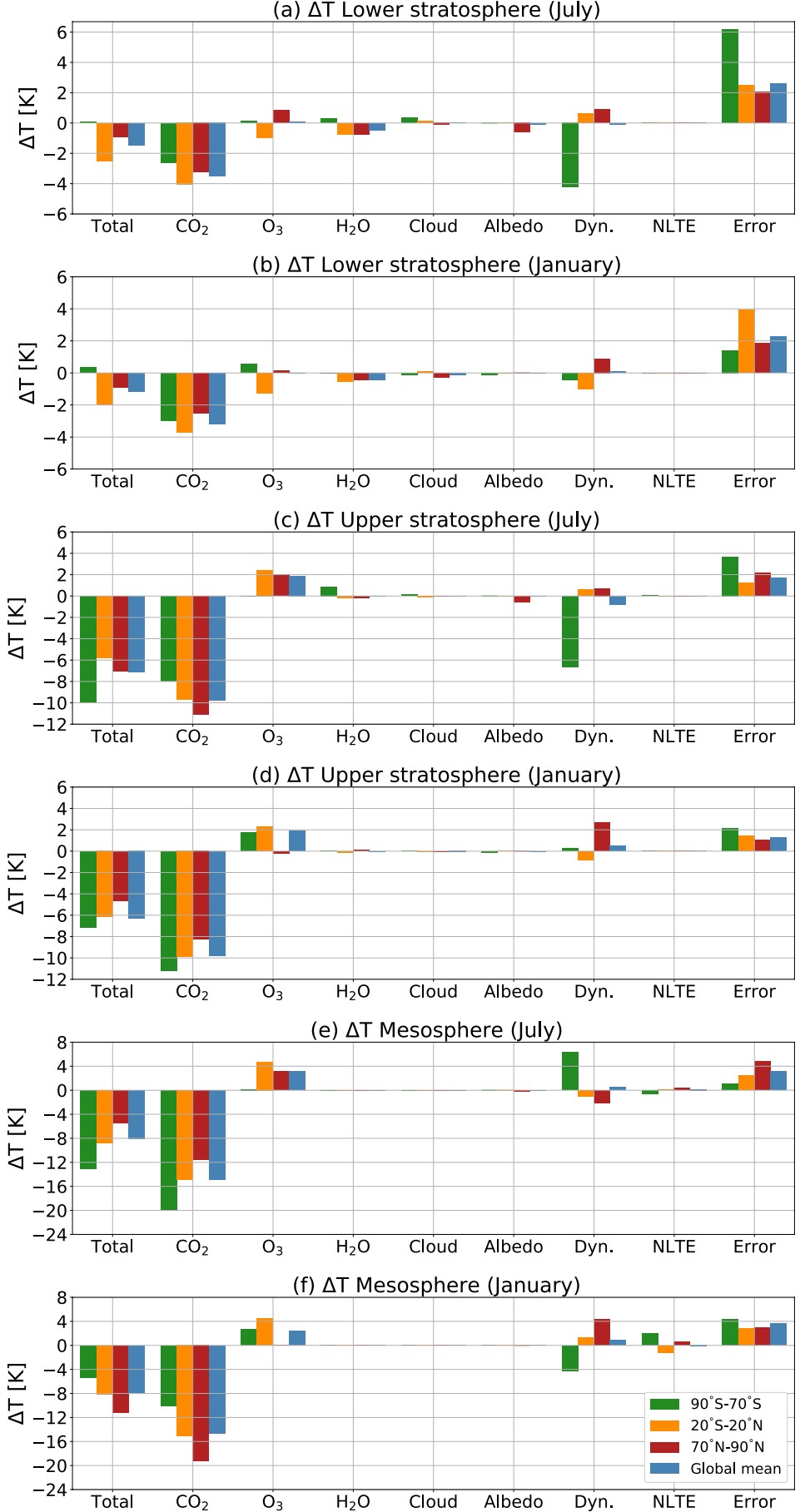


Figure 11: The mean temperature responses to the changes in $CO_2$ and
various feedback processes in the lower stratosphere from the tropopause
height up to 24 hPa (a,b), upper stratosphere from 24-1 hPa (c, d) and in the
mesosphere from 1-0.01 hPa (e,f) in July (a, c, e) and January (b, d, f) in the
polar regions (90°S-70°S and 70°N-90°N), the tropics (20°S-20°N) and the
global mean, for S3 experiment (double $CO_2$ and changed SSTs). Note that
the range of values on the y-axis is not the same for the different subplots.
In addition, the vertical profiles of the temperature responses to the direct
forcing of $CO_2$ and the feedbacks are shown in Figure 12. Here, one can see
that the increase in $CO_2$ leads to a cooling over almost the whole middle
atmosphere; an effect that increases with height. We also observe that in the
summer upper mesosphere regions, the increased $CO_2$-concentration leads
to a warming. The changes in ozone concentration in response to the
doubling of $CO_2$ lead to a warming almost everywhere in the atmosphere. In
some places, this warming exceeds 5 K. In the polar winter the effect of ozone
is small due to lack of sunlight.
There is also a relatively large temperature response to the changes in
dynamics. In Fig. 12, it can be seen that there is a cooling in the summer
mesosphere, while there is warming in the winter mesosphere. The water
vapour, cloud and albedo feedback play only a very small role in the middle
atmosphere, as we observed in Figure 11. We find that there are also some
small temperature changes due to non-LTE effect above 0.1 hPa. How the
non-LTE effects exactly cause the small temperature changes in this region is
outside the scope of this paper and needs further investigation.

Temperature responses to different feedback processes

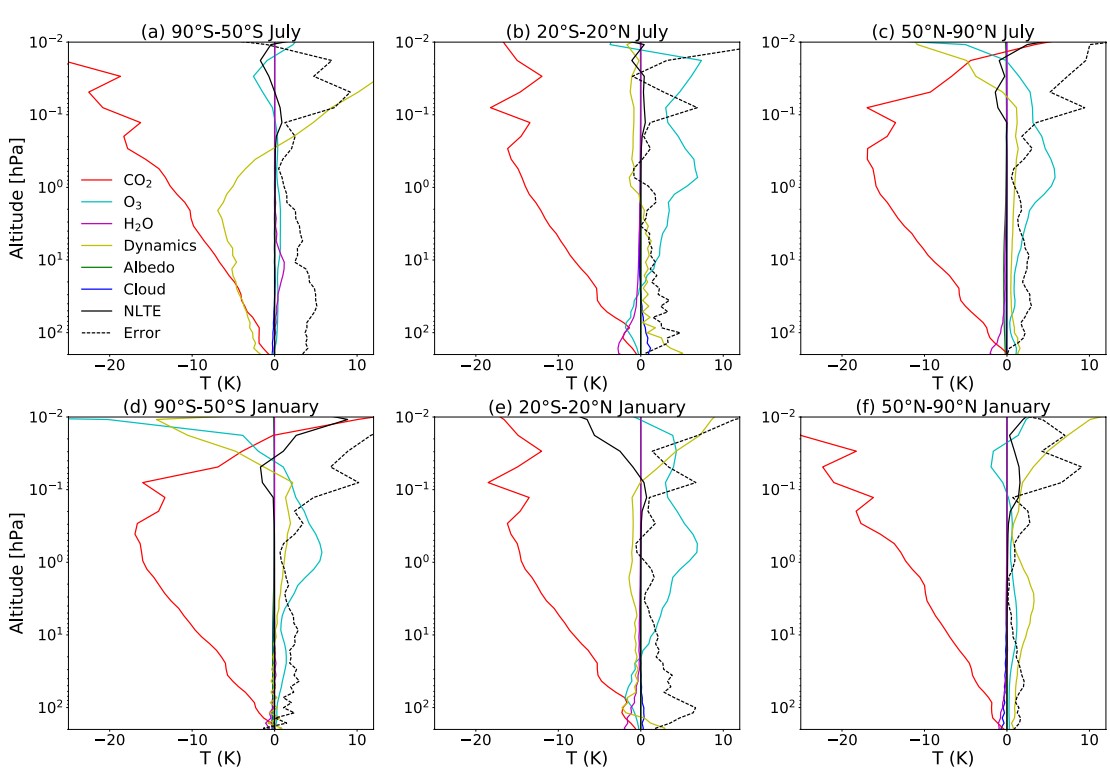


Figure 12: Vertical profiles of the temperature responses to the changes in $CO_2$ and various feedback processes in July (top) and January (bottom) due to double $CO_2$ and changed SSTs in the atmosphere between 200 and 0.01 hPa, for regions from 50° N/S poleward and the tropics (20°S-20°N), as calculated by CFRAM.

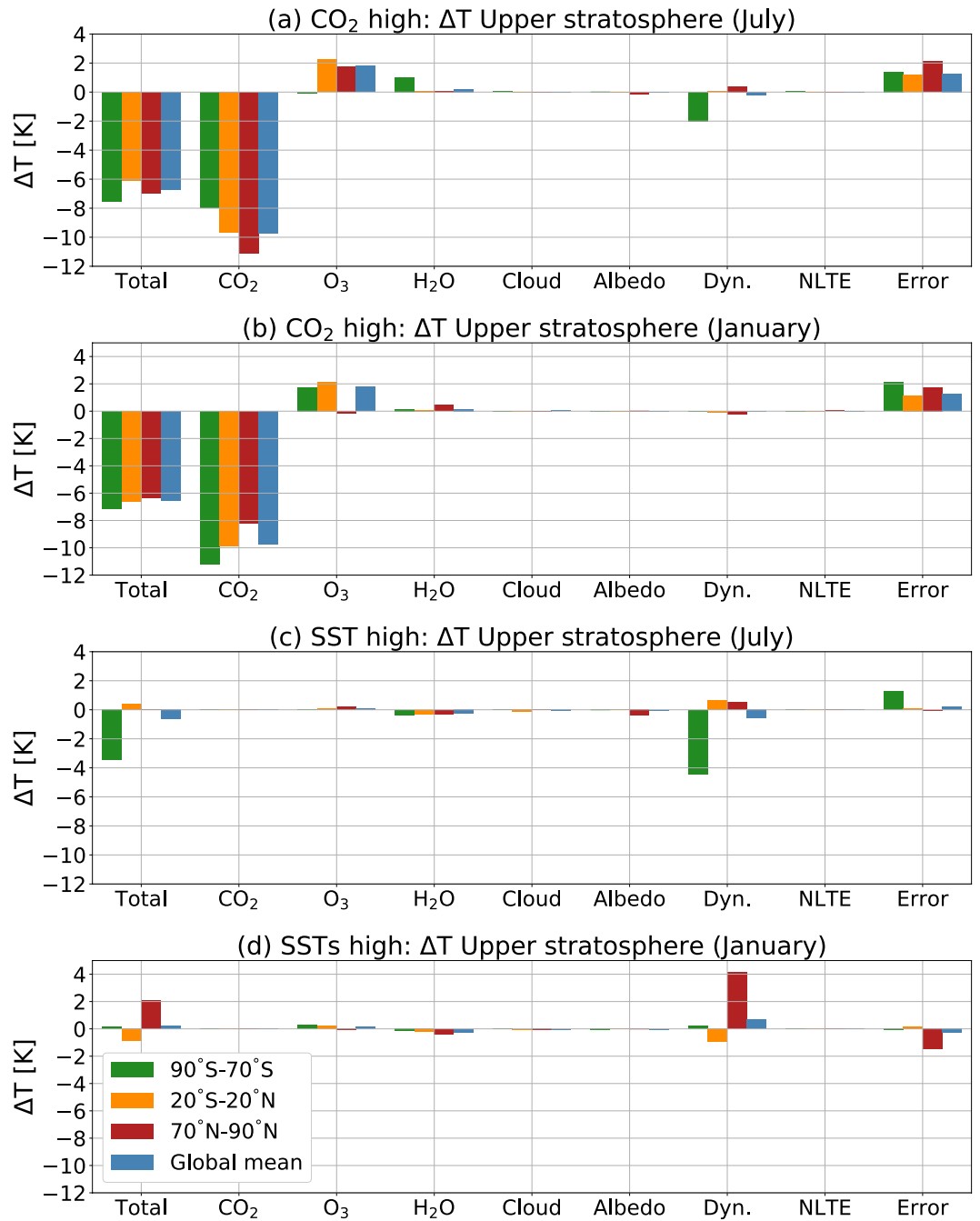

Figure 13: The mean temperature responses to the changes in $CO_2$ and various feedback processes in July (a,c) and January (b,d) in the upper stratosphere between 24 and 1 hPa, for polar regions (90°S-70°S and 70°N-90°N), the tropics (20°S-20°N) and the global mean for the experiment with double $CO_2$ (S1) (a,b) and changed SSTs (S2) (c,d) separately.

Figure 13 shows the temperature responses in the upper stratosphere for the
experiment with double $CO_2$ (a,b) and changed SSTs (c,d) separately. This
has been done to give insight in the temperature response of the $CO_2$ and the
SST separately. These temperature changes were calculated in the same
way as for Fig. 11. Again also, the 'total'-column shows the temperature
changes as simulated by WACCM, the columns $CO_2$, $O_3$, $H_2O$, cloud, albedo,
dynamics, the column 'NLTE' shows the temperature responses due to non-
LTE processes as calculated by CFRAM. As in Fig. 11, the 'Error'-column in
Fig.13 shows the difference between temperature change in WACCM and the
sum of the calculated temperature responses in CFRAM.
We learn from this figure that the effects of the changed SSTs on the upper
stratosphere are relatively small as compared to the effects of changing the
$CO_2$. We show the temperature changes for the upper stratosphere as an
example. In the lower stratosphere and the mesosphere, we see the same
pattern: the effect of the $CO_2$ on the temperature is generally much larger than
the effect of the SSTs on the temperature. This finding is consistent with the
study of *Fomichev et al.* (2007), where it is concluded that the impact of
changes in SSTs on the middle atmosphere is relatively small and localized
as compared to the combined response of changing the $CO_2$-concentration
and the SSTs.
The changes in SSTs are, however, responsible for large temperature
changes as a result of the dynamical feedbacks, especially in the winter
hemispheres, where there is a temperature response of 4K. A similar figure
for the lower stratosphere (not shown) shows that the temperature response
to the water vapour feedback is almost solely due to changes in the SSTs and
not the direct forcing of $CO_2$.
Earlier, we discussed that the sum of the two separate temperature changes
in the experiment with double $CO_2$ and changed SSTs is approximately equal
to the changes observed in the combined simulation. We find that the same is
true for the temperature responses to the different feedback processes.
**5. Discussion and conclusions**
In this study, we have applied the climate feedback response analysis method
to climate sensitivity experiments performed by WACCM. We have examined
the middle atmosphere response to $CO_2$ doubling with respect to the pre-
industrial state. We investigated the combined effect of doubling $CO_2$ and
subsequent warming SSTs, as well as the effects of separately changing the
$CO_2$ and the SSTs. It is important to note that no other method before has
been able to quantify how much of the temperature change in the middle
atmosphere is due to the different feedback processes.
It was found before that the sum of the two separate temperature changes in
the experiment with only changed $CO_2$ and only changed SSTs is, at first
approximation, equal to the changes observed in the combined simulation
(see e.g. *Fomichev et al. (2007) and Schmidt et al. (2006)*). This is also the
case for WACCM.
We have found that, even though changing the SSTs yields significant
temperature changes over a large part of the middle atmosphere, the effects
of the changed SSTs on the middle atmosphere are relatively small as
compared to the effects of changing the $CO_2$ without changes in the SSTs.
We have given an overview of the mean temperature responses to the
changes in $CO_2$ and various feedback processes in the lower stratosphere,
upper stratosphere and in the mesosphere in January and July. We find that
the temperature change due to the direct forcing of $CO_2$ increases with
increasing height in the middle atmosphere. The temperature change in the
lower stratosphere due to the direct forcing of $CO_2$ is around 3 K. There is a
stronger cooling in the tropical lower stratosphere of about 4 K in July and 3.5
K in January.
In the upper stratosphere, the cooling due to the direct forcing of $CO_2$ is about
9 K, which is considerably stronger than in the lower stratosphere. The
cooling is stronger in the summer polar regions, where the cooling reaches a
value of 11K, than in the winter polar region, where the cooling is only about
8K. In the mesosphere, the cooling due to the direct forcing of $CO_2$ is even
stronger: 15 K.
The ozone concentration changes due to changes in the $CO_2$-concentration
as well as by changes in the SSTs. The temperature changes caused by this
change in ozone concentration generally mitigate the cooling caused by the
direct forcing of $CO_2$. However, in the tropical lower stratosphere and in some
regions of the mesosphere, the ozone feedback cools these regions further. In
the tropical lower stratosphere, for example, there is a cooling of 1K due to
the ozone feedback.
We also have seen that the global mean temperature response due to
dynamical feedbacks is small in the global average in all regions: less than 1
K. However, local responses to the changes in dynamics can be large.
Doubling the $CO_2$-concentration leads to a stronger summer-to-winter-pole
flow, which leads to a cooling of the summer mesosphere and a warming of
the winter mesosphere. Changing the SSTs weakens this effect in the
mesosphere, but affects the temperature response in the stratosphere and
lower mesosphere.
Using CFRAM on WACCM data shows that the change in water vapour leads
to a cooling of up to 2 K in the lower stratosphere. It should be noted that
climate models currently have a limited representation of the processes
determining the distribution and variability of lower stratospheric water vapour.
This means that the temperature response to the water vapour feedback
might be different using a different model. We have also seen a small effect of
the cloud and albedo feedback on the temperature response in the lower
stratosphere, while these feedbacks play no role in the upper stratosphere
and the mesosphere.
The results seen in this study are consistent with earlier findings. As in
*Shepherd et al.,* (2008), we find that the higher the temperature at a region in
the atmosphere, the more cooling there is seen due to the direct feedback of
$CO_2$. We find, as in *Zhu et al.,* (2016) that the temperature responses due to
the direct forcing of $CO_2$ follow the temperature distribution quite closely, while
the temperature responses due to $O_3$ follow the changes in ozone
concentration instead.
We have also seen that the ozone feedback generally yields a radiative
feedback that mitigates the cooling, which is due to the direct forcing of $CO_2$,
which is consistent with earlier studies such as *Jonsson et al., (2004),*
*Dietmüller et al., (2014)*. CFRAM is the first study that allows for calculating
how much of the temperature response is due to which feedback process.
The next step would be to investigate the exact mechanisms behind the
feedback processes in more detail. Some processes can influence the
different feedback processes, such as ozone depleting chemicals influencing
the ozone concentration and thereby the temperature response of this
feedback. A better understanding of the effect of the increased $CO_2$-
concentration on the middle atmosphere, will help to distinguish the effects of
the changes $CO_2$- and $O_3$-concentration.
There is also a need for a better understanding of how different feedbacks in
the middle atmosphere affect the surface climate. As discussed in the
introduction, the exact importance of ozone feedback on the global mean
temperature is currently not clear (*Nowack et al.*, 2015, *Marsh et al.*, 2016). A
similar analysis as in this paper can be performed to quantify the effects of
feedbacks on the surface climate.
In conclusion, we have seen that CFRAM is an efficient method to quantify
climate feedbacks in the middle atmosphere, although there is a relatively
large error due to the linearization in the model. The CFRAM allows for
separating and estimating the temperature responses due to an external
forcing and various climate feedbacks, such as ozone, water vapour, cloud,
albedo and dynamical feedbacks. More research into the exact mechanisms
of these feedbacks could help us to understand the temperature response of
the middle atmosphere and their effects on the surface and tropospheric
climate better.
**Appendix: Formulation of CFRAM diagnostics using outputs of WACCM**
The mathematical formulation of CFRAM is based on the conservation of total
energy (*Lu and Cai*, 2009). At a given location in the atmosphere, the energy
balance in an atmosphere-surface column can be written as:
$$\vec{R} = \vec{S} + \vec{Q}^{conv} + \vec{Q}^{turb} - \vec{D}^v - \vec{D}^h + \vec{W}^{fric} \qquad (A1)$$
$\vec{R}$ represents the vertical profile of the net long-wave radiation emitted by each
layer in the atmosphere and by the surface. $\vec{S}$ is the vertical profile of the solar
radiation absorbed by each layer. $\vec{Q}^{turb}$ is the convergence of total energy
fluxes in each layer due to turbulent motions, $\vec{Q}^{conv}$ is convergence of total
energy fluxes into the layers due to convective motion. $\vec{D}^{v}$ is the large-scale
vertical transport of energy from different layers to others. $\vec{D}^{h}$ is the large-
scale horizontal transport within the layers and $\vec{W}^{fric}$ is the work done by
atmospheric friction. All terms in (A1) have units of $Wm^{-2}$.
Due to an external forcing (in this study, the change in $CO_2$-concentration
and/or change in SSTs), the difference in the energy flux terms then
becomes:
$$\Delta\vec{R} = \Delta\vec{F}^{ext} + \Delta\vec{S} + \Delta\vec{Q}^{conv} + \Delta\vec{Q}^{turb} - \Delta\vec{D}^{v} - \Delta\vec{D}^{h} + \Delta\vec{W}^{fric} \qquad (A2)$$
In which the delta ($\Delta$) stands for the difference between the perturbation run
and the control run.
CFRAM takes advantage of the fact that the infrared radiation is directly
related to the temperatures in the entire column. The temperature changes in
the equilibrium response to perturbations in the energy flux terms can be
calculated. This is done by requiring that the temperature-induced changes in
infrared radiation balance the non-temperature induced energy flux
perturbations.
Equation (A2) can also be written as:
$$\Delta(\vec{S} - \vec{R})_{total} + \Delta dyn = 0 \qquad (A3)$$
The term $\Delta(\vec{S} - \vec{R})$ can be calculated as the longwave heating rate and the
solar heating rate are output variables of the model simulations. We take the
time mean of the WACCM data and perform the calculations for each grid
point of the WACCM data. This means that in the end, we will have the
temperature changes at each latitude, longitude and height.
We then calculate the difference in these heating rates for the perturbation
simulation and the control simulation.
We use the term $\Delta(\vec{S} - \vec{R})_{total}$ to calculate the dynamics term $\Delta dyn$.
$$\Delta dyn = -\Delta(\vec{S} - \vec{R})_{total} \qquad (A4)$$
WACCM provides us with a heating rate in $Ks^{-1}$. For the CFRAM calculations,
we need the energy flux in $Wm^{-2}$. We can calculate the energy flux by
multiplying with the mass of different layers in the atmosphere and the specific
heat capacity.
$$\Delta(\vec{S} - \vec{R}) = \Delta(\vec{S} - \vec{R})_{(WACCM)} * mass_k * c_p \qquad (A5)$$

In which $\Delta(\vec{S} - \vec{R})$ is the difference in the shortwave radiation $(\vec{S})$ and
longwave radiation $(\vec{R})$) between the perturbation run and the control run as a
flux in Wm$^{-2}$, while $\Delta(\vec{S} - \vec{R})_{(WACCM)}$ is this difference as heating rate in Ks$^{-1}$ in
WACCM, with $mass_k = \frac{p_{k+1} - p_k}{g}$ with p in Pa, $c_p = 1004$ J $kg^{-1}$ $K^{-1}$ the
specific heat capacity at constant pressure and $g$ the gravitational
acceleration 9.81 $ms^{-2}$.
WACCM includes a non-local thermal equilibrium (non-LTE) radiation scheme
above 50 km. It consists of a long-wave radiation (LW) part and a short-wave
radiation (SW) part which includes the extreme ultraviolet (EUV) heating rate,
chemical potential heating rate, $CO_2$ near-infrared (NIR) heating rate, total
auroral heating rate and non-EUV photolysis heating rate.
Therefore, we split the term $\Delta(\vec{S} - \vec{R})_{total}$ in an LTE and a non-LTE term:
$$\Delta(\vec{S} - \vec{R})_{total} = \Delta(\vec{S} - \vec{R})_{LTE} + \Delta(\vec{S} - \vec{R})_{non-LTE} \qquad \text{(A6)}$$
WACCM provides us with the total longwave heating rate as well as the total
solar heating rate and the non-LTE longwave and shortwave heating rates for
the different runs. This means that we can calculate the term $\Delta(\vec{S} - \vec{R})_{non-LTE}$
as well, where we again need to convert our result from Ks$^{-1}$ to Wm$^{-2}$:
$$\Delta(\vec{S} - \vec{R})_{non-LTE} = \Delta(\vec{S} - \vec{R})_{non-LTE(WACCM)} \; mass_k * c_p \qquad \text{(A7)}$$
This term can be inserted in equation (3):
$$\Delta(\vec{S} - \vec{R})_{LTE} + \Delta(\vec{S} - \vec{R})_{non-LTE} + \Delta dyn = 0 \qquad \text{(A8)}$$
The central step in CFRAM is to decompose the radiative flux vector, using a
linear approximation.
We start by decomposing the LTE infrared radiative flux vector $\Delta \vec{R}$
$$\Delta \vec{R}_{LTE} = \frac{\partial \vec{R}}{\partial \vec{T}} \Delta T + \Delta \vec{R}_{CO_2} + \Delta \vec{R}_{O_3} + \Delta \vec{R}_{H_2O} + \Delta \vec{R}_{albedo} + \Delta \vec{R}_{cloud} \qquad \text{(A9)}$$
where $\Delta \vec{R}_{CO_2}$ , $\Delta \vec{R}_{O_3}$ , $\Delta \vec{R}_{H_2O}$ , $\Delta \vec{R}_{albedo}$ , $\Delta \vec{R}_{cloud}$ are the changes in infrared
radiative fluxes due to the changes in $CO_2$, ozone, water vapour, albedo and
clouds, respectively.
For equation (A9), we assumed that radiative perturbations can be linearized
by neglecting the higher order terms of each thermodynamic feedback and
the interactions between these feedbacks. This is also commonly done in the
partial radiative perturbation (PRP) method, in which partial derivatives of the
model top of the atmosphere radiation are evaluated with respect to changes
in model parameters by diagnostic rerunning the model's radiation code (*Bony*
*et al.,* 2006).

The term $\frac{\partial \vec{R}}{\partial \vec{T}}\Delta T$ represents the changes in the IR radiative fluxes related to the
temperature changes in the entire atmosphere-surface column. The matrix $\frac{\partial \vec{R}}{\partial \vec{T}}$
is the Planck feedback matrix, in which the vertical profiles of the changes in
the divergence of radiative energy fluxes due to a temperature change are
represented.
We calculate this feedback matrix using the output variables of the
perturbation and the control run of WACCM and inserting these in the CFRAM
radiation code: atmospheric temperature, surface temperature, reference
height temperature, ozone, surface pressure, solar insolation, downwelling
solar flux at the surface, net solar flux at the surface, dew point temperature,
cloud fraction, cloud ice amount, cloud liquid amount, ozone and specific
humidity.
Similarly, the changes in the LTE shortwave radiation flux can be written as
the sum of the change in shortwave radiation flux due to the direct forcing of
$CO_2$ and the different feedbacks:

$$\Delta \vec{S}_{LTE} = \Delta \vec{S}_{CO_2} + \Delta \vec{S}_{O_3} + \Delta \vec{S}_{H_2O} + \Delta \vec{S}_{albedo} + \Delta \vec{S}_{cloud} \qquad \text{(A10)}$$
Similarly, to equation (A9), we perform a linearization.
Substituting (A9) and (A10) in equation (A8) yields:

$$\Delta(\vec{S} - \vec{R})_{CO_2} + \Delta(\vec{S} - \vec{R})_{O_3} + \Delta(\vec{S} - \vec{R})_{H_2O} + \Delta(\vec{S} - \vec{R})_{albedo} + \Delta(\vec{S} - \vec{R})_{cloud} - \frac{\partial \vec{R}}{\partial \vec{T}}\Delta T$$
$$+\Delta(\vec{S} - \vec{R})_{non-LTE} + \Delta dyn = 0 \qquad \text{(A11)}$$
This can be written as:

$$\Delta T = \left(\frac{\partial \vec{R}}{\partial \vec{T}}\right)^{-1}\left\{\Delta(\vec{S} - \vec{R})_{CO_2} + \Delta(\vec{S} - \vec{R})_{O_3} + \Delta(\vec{S} - \vec{R})_{H_2O} + \Delta(\vec{S} - \vec{R})_{albedo} + \right.$$
$$\left. \Delta(\vec{S} - \vec{R})_{cloud} + \Delta(\vec{S} - \vec{R})_{non-LTE} + \Delta dyn\right\} \qquad \text{(A12)}$$
As described in the main text of this paper, we can solve Eq. (A12) for each of
the terms on its right-hand side, based on the linear decomposition principle.
This yields the partial temperature changes due to each specific process. The
factors $\Delta(\vec{S} - \vec{R})_{CO_2}$ , $\Delta(\vec{S} - \vec{R})_{O_3}$, $\Delta(\vec{S} - \vec{R})_{H_2O}$ , $\Delta(\vec{S} - \vec{R})_{albedo}$ and
$\Delta(\vec{S} - \vec{R})_{cloud}$ in eqs (1-5) are calculated by inserting the output variables
from WACCM in the radiation code of CFRAM. Here, one takes the output
variables from the control run, apart from the variable that is related to the
direct forcing or the feedback. The table below shows which variables have
been taken from the perturbation runs for each feedback.

| Direct forcing/feedback | Changed variables in the radiation code |
|---|---|
| $CO_2$ | $CO_2$ |

| | |
|---|---|
| Ozone | $O_3$ |
| Water vapour | Specific humidity |
| | Surface pressure |
| | Surface temperature |
| | Dew point temperature |
| Albedo | Downwelling solar flux at surface |
| | Net solar flux at surface |
| Cloud | Cloud fraction |
| | Cloud ice |
| | Cloud liquid amount |

Table A1: The variables from the perturbation runs inserted in the radiation
code of CFRAM to calculate the temperature change in response to the
changes in $CO_2$, $O_3$, water vapour, cloud and albedo.

**Acknowledgements**

The computations and simulations were performed on resources provided by
the Swedish National Infrastructure for Computing (SNIC) at National
Supercomputer Center (NSC) in Linköping.

Hamish Struthers NSC is acknowledged for assistance concerning technical
aspects in making the WACCM code run on NSC supercomputer Tetralith.
We thank Qiang Zhang for helping to make the radiation model code
applicable to WACCM model data.

**Competing interests**

The authors have no competing interests to declare.

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
