# Peer review of "Using the climate feedback response method to quantify climate feedbacks in the middle atmosphere in WACCM"

_Atmospheric Chemistry and Physics, 2019_

## Referee Comment (RC1) · Anonymous Referee #3 · 9 Mar 2020

I am sorry to say that I am still not happy with this paper in its present form. The paper is missing a clear message and many statements remain vague. For example in the abstract: "feedback processes" (l 51) but which processes? CFRAM (l 54) is not known to me; I am not sure how helpful this statement is without further explanation to the readers of ACP. Response to CO2 doubling but at what time has the doubling been reached – would that not be important for the issue of stratospheric ozone? Ozone feedback is mentioned (l 63), but what is assumed for ozone in the upper stratosphere? We know upper stratospheric ozone is "recovering" over the coming decades (WMO, 2018); is this the point here? And a "warming by 1.5 K, but in which region? Changes in dynamics play a large role (l 66) but which changes, which role? And above 0.1 hPa, which is certainly a region where an ozone feedback is expected. Above 0.1 hpa

is above 60 km; this is the mesosphere. Several tropospheric issues are "of minor importance" (l 69), but why is this issue discussed in an abstract of a paper on the "middle atmosphere"?

Further comments:

l. 85-90: It should be clearly said that the middle atmosphere is *not* in radiative equilibrium in most regions; downwelling in the polar regions in winter is part of the BD circulation, and not only an "example". See e.g. Dunkerton, JAS, 1978.

l 115-122: Ozone is mentioned here, but it is well established that stratospheric ozone responds to changing halogen levels (WMO 2018). This aspect can not be ignored in this study.

Sec. 2.3: Is this a new formulation of CFRAM or is this section just reiterating a technique used before? It looks like a new description here as there are no references to previous description of CFRAM at the top of sec 2.3.

l 731-735: No, the ozone concentration is *not* controlled by the Chapman reactions "for a large part". Depending on altitude, NOx and HOx cycles are important; this is well known (check the textbook you are citing). Also chlorine compounds are relevant.

l 774: The direct influence is only possible where the chemistry not too fast, again check the Brasseur and Solomon textbook.

Sec. 3.5: The feedback for stratospheric H2O is not that simple, see eg. Solomon et al., Science, 2010; Riese et al, JGR, 2012). Several points are mixed here. There might be a change in stratospheric water vapour based on changing climatic conditions and this change could have an impact on local heating/cooling but also an effect of the surface radiative forcing. These issues should be disentangled.

l 820: warmer tropopause: also in the tropics? (where water vapour is entering the stratosphere). Would you not expect higher SSTs causing more wave activity and thus a stronger tropical upwelling? Why is this argument not correct?

l 815-816: Is this reduction expected from simple water vapour equilibrium (over ice) arguments? Citations?

Sec 3.6 discusses feedbacks that "play only a very small role" for the middle atmosphere. Fig 12 shows that the middle atmosphere is largely grey (zero effect). But the caption tells me that the comparison is to pre-industrial conditions, so the reported "delta" is due to a change relative to pre-industrial? Then one would expect a larger signal – correct? And the discussion starting in l. 810 is not discussing "pre-industrial"; this is confusing.

L 964-972: these lines seem to describe the overall conclusions of the paper; when I read these lines they seem to tell me that CFRAM is okay, but that some refinement is necessary. This is a rather technical statement (which would be more helpful if statements like "some" would be more specific). Most importantly however, the paper promises to talk about "quantifying (!) climate feedbacks in the middle atmosphere" – in my view this has not been achieved in this manuscript.

References:

* The current report on stratospheric ozone is WMO 2018, I suggest using this most current information (see above)

Brasseur and Solomon 2005: this is an excellent text book but might not give the most up to date information required here on upper stratospheric ozone

---

## Referee Comment (RC2) · Anonymous Referee #2 · 1 Apr 2020

**1   General Comments**

The paper presents another application of CFRAM, a one-dimensional (mostly) radiative climate model for offline feedback analysis, in a more than a decade long series, now using output of the high top chemical climate model WACCM. In other studies it was used for example for a low top GCM (Taylor et al., 2013) or a CCM up to thermosphere (Zhu et al., 2016). When it was applied to global radiative models more than a decade ago, data transfer was straight forward but for use with complex 3D models it is essential to provide information on averaging of model output. I suppose CFRAM was applied for zonal averages at every meridional grid point and for every month of the 40 year time slices, am I right from hints only in the references? There are also a

lot of other options. This has to be documented since this can contribute significantly to errors (see for example TEM-analysis mentioned by authors, line 652; Zhu et al., 2016).

The shown results are not new, they almost resemble what was found with chemical radiative convective models more than 30 years ago (e.g. Brühl and Crutzen, 1988). Concerning upper stratospheric ozone chemistry, the authors should for example read Cariolle (1983), what is written in the manuscript is a mess.

In general, the paper needs much more clear definitions what has been done.

**2  Specific comments**

The presented averages of temperature change from 12 to 80km altitude in the abstract and also key points are confusing and not physically meaningful because several different regimes are envolved. Here it would be more useful to focus on the upper stratosphere. Is the averaging mass weighted or not?

More than a decade ago is not recently (paragraph beginning with line 136). What is the spatially limited domain, please define. Provide references earlier. Merge with next paragraph and rearrange.

The paragraph beginning with line 157 is confusing concerning the statements on ozone here, skip that or define clearly what are the ozone changes due to, including the altitude dependence.

In section 2.2 the assumptions for the other radiatively and chemically active gases should be provided. Is the double $CO_2$ scenario with preindustrial conditions for $N_2O$, CFCs, $CH_4$ and NOx in the troposphere? This is also critical for the SST.

Section 2.3 should include how WACCM output is implemented into CFRAM, i.e. the

averaging methods for space and time.

Split Fig. 2 and Fig. 4 into more vertical sections (e.g. lower stratosphere, upper stratosphere, mesosphere). What kind of averaging?

The paragraph beginning with line 564 is confusing. I suppose you mean ozone changes induced by $CO_2$ cooling. Ozone matters also in the infrared window.

Shorten the paragraph beginning with line 628.

You may improve the paragraph beginning with line 645 by the use of the textbook by Holton.

Don't forget to mention convection around line 688.

Does Fig. 6 show the average of the 40yr time slice?

Section 3.4 has to be rearranged and improved, the key processes are missing. The reaction $O+O_3$, the sink reaction in the Chapman chemistry, is strongly temperature dependent (see Brühl und Crutzen, 1988; Cariolle, 1983; or JPL). NO and Cl catalytic cycles matter mostly in the upper and mid stratosphere, in the mesosphere hydrogen species (e.g. OH) are most important. Check if in all calculations only $CH_3Cl$ acts as chlorine source (pre-industrial!).

The statement in line 820 is quite controversial, a lot of models lead to different results here; please check.

Split the averaging region in line 903ff consistent with the new figures and the revised abstract.

**3  Technical corrections**

Please define all formula letters in line 298.

In line 546 something is missing or twice.

Add 'high latitude' in line 616.

Typo in line 688.

Captions of Fig. 7 to 11 can be shortened (tropopause as in Fig. 6).

**4   References**

Brühl, C. and P.J. Crutzen: Scenarios of possible changes in atmospheric temperatures and ozone concentrations due to man's activities, estimated with a one-dimensional coupled photochemical climate model. Climate Dynamics, 2: 173-203, 1988.

Cariolle, D.: The ozone budget in the stratosphere: results of a one-dimensional photochemical model. Planet. Space Sci., 31: 1033-1052, 1983.

Others see discussion paper.

---

## Short Comment (SC1) · 20 Apr 2020

I hereby answer some of the major questions asked by reviewer #3 that are crucial in understanding what we have done. I note that all these issues also already have been raised and addressed in the pre-review. I am working at the same time to rewrite the paper in a way that makes it more clear what the purpose of the work is.

What we try to do in this paper is to apply a new method to quantify the temperature responses to different feedback processes that arise in response to changing the $CO_2$-concentration. This is one of the first studies with which can calculate how much of the temperature change in a specific place in the atmosphere is attributed to which feedback process. The method we applied here can quantify the temperature response,

but to provide a complete explanation of all the responses and the exact mechanism behind all the feedback processes is outside the scope of this paper.

Just in the abstract: "feedback processes" (l 51) but which processes? The "feedback processes" we mean chemical, physical and dynamical processes, which can feedback to the radiation and further change the temperature.

Ozone feedback is mentioned (l 63), but what is assumed for ozone in the upper stratosphere? We know upper stratospheric ozone is "recovering" over the coming decades (WMO, 2018); is this the point here?

We are not speaking here about the changes in O3-concentration due to the ozone hole, but rather changes in ozone concentration that are resulting from changes in the $CO_2$ concentration. The ozone concentration in the control run is for pre-

Several tropospheric issues are "of minor importance", but why is this issue discussed in an abstract of a paper on the "middle atmosphere"?

We calculate the temperature changes in the middle atmosphere, due to different feedback processes, such as the ozone feedback. We know the albedo and cloud feedback are important for the troposphere, but with CFRAM, we can quantify the importance of these feedback processes for the middle atmosphere. We imagine that some processes in the troposphere also affect the temperature changes in the middle atmosphere. In fact, this is what we see is happening in the lower stratosphere, where we do see effects of the cloud, albedo and water vapour feedback.

---

## Short Comment (SC2) · 20 Apr 2020

**1 General Comments**

The paper presents another application of CFRAM, a one-dimensional (mostly) radiative climate model for offline feedback analysis, in a more than a decade long series, now using output of the high top chemical climate model WACCM. In other studies it was used for example for a low top GCM (Taylor et al., 2013) or a CCM up to thermosphere (Zhu et al., 2016). When it was applied to global radiative models more than a decade ago, data transfer was straight forward but for use with complex 3D models it is essential to provide information on averaging of model output. I suppose CFRAM was applied for zonal averages at every meridional grid point and for every month of the 40 year time slices, am I right from hints only in the references? There are also a lot of other options. This has to be documented since this can contribute significantly to errors (see for example TEM-analysis mentioned by authors, line 652; Zhu et al., 2016).

The calculations are done at every grid point of the WACCM model after temporal means of the data of the perturbation run and the control run are taken. This means the data that go into the CFRAM calculations have the dimensions of latitude, longitude and height. A zonal mean has been done only after the calculations. This information has now been added to section 2.3:

*We take the time mean of the WACCM data and perform the calculations for each grid point of the WACCM data. This means that in the end, we will have the temperature changes at each latitude, longitude and height.*

The shown results are not new, they almost resemble what was found with chemical radiative convective models more than 30 years ago (e.g. Brühl and Crutzen, 1988). Concerning upper stratospheric ozone chemistry, the authors should for example read Cariolle (1983), what is written in the manuscript is a mess.

Thanks the references. The section on the changes in the ozone concentration due to the changed $CO_2$ concentration has now been rewritten and significantly shortened, as this is not new and not the main point of our work: we are interested in the temperature response as a result of the changes in the $O_3$ concentration in WACCM.

In general, the paper needs much more clear definitions what has been done.

First of all, the authors would like to thank the reviewer for the helpful comments.

**2 Specific comments**

The presented averages of temperature change from 12 to 80 km altitude in the abstract and also key points are confusing and not physically meaningful because several different regimes are involved. Here it would be more useful to focus on the upper stratosphere. Is the averaging mass weighted or not?

The authors agree it is better to show the temperature changes for the specific regions in the middle atmosphere. When calculating the temperature changes, we do take into account the mass of the layers as can be seen in equation (7). When doing the averaging over the different heights and latitudes, we just take a ordinary average and account for this again.

More than a decade ago is not recently (paragraph beginning with line 136). What is the spatially limited domain, please define. Provide references earlier. Merge with next paragraph and rearrange.

This has been done now.

The climate feedback-response analysis method (CFRAM) is an alternative method which takes into account that the climate change is not only determined by the energy balance at the top of the atmosphere, but is also influenced by the energy flow within the Earth's system itself (*Cai and Lu, 2009, Lu and Cai*, 2009). The method is based on the energy balance in an atmosphere-surface column. It solves the linearized infrared radiation transfer model for the individual energy flux perturbations. This makes it possible to calculate the partial temperature changes due to an external forcing and these internal feedbacks in the atmosphere. It has the unique feature of additivity, such that these partial temperature changes are linearly addable.

The paragraph beginning with line 157 is confusing concerning the statements on ozone here, skip that or define clearly what are the ozone changes due to, including the altitude dependence.

Right, this is how it is stated in their paper but I understand it misses more information here. As I don't want to do too much in detail of their study, I have decided to skip this part.

In section 2.2 the assumptions for the other radiatively and chemically active gases should be provided. Is the double $CO_2$ scenario with preindustrial conditions for $N_2O$, CFCs, $CH_4$ and NOx in the troposphere? This is also critical for the SST.

This information has been added to section 2.2:

*Other radiatively and chemically active gases, such as ozone, will change because of the changes in the $CO_2$-concentration, due to WACCM's chemical model as well.*

As we used a fixed SST from the CSEM model as forcing. The atmosphere component of CESM is the same as WACCM, but does not include stratospheric

chemistry. So you are right that this is an element that we are not including in our analysis.

Section 2.3 should include how WACCM output is implemented into CFRAM, i.e. the averaging methods for space and time.

*This information has been added to section 2.3:*

*We take the time mean of the WACCM data and perform the calculations for each grid point of the WACCM data. This means that in the end, we will have the temperature changes at each latitude, longitude and height.*

Split Fig. 2 and Fig. 4 into more vertical sections (e.g. lower stratosphere, upper stratosphere, mesosphere). What kind of averaging?

*This has now been done: see Figure on the next page. The averaging procedure has also been explained in the text:*

*Figure 2 shows the average change in global mean temperature for the lower stratosphere, the upper stratosphere and the mesosphere for the experiment with the changed $CO_2$-concentration and changed SSTs. To calculate the lower stratosphere temperature changes, we take the average value of the temperature change from the tropopause – the pressure level of which is an output of WACCM – until about 24 hPa for each latitude.*

*The tropopause is not exactly at the same pressure level, we always take the one for the perturbation experiment which is a bit higher at some latitudes, to make sure that we don't use values from the troposphere.  We add the values for each latitude up and take the average. This average is not mass weighted. The temperature changes in the upper stratosphere and in the mesosphere are calculated in the same way, but then for the altitudes 24 hPa-1 hPa and 1 hPa-0.01 hPa respectively.*

*Figure 2 shows the radiative feedbacks due to ozone, water vapour, clouds, albedo and the dynamical feedback, as well as the small contribution due to the Non-LTE processes, as calculated by CFRAM. The 'total'-column shows the temperature changes in WACCM, while the column 'error' shows the difference between temperature change in WACCM and the sum of the calculated temperature responses in CFRAM. In sections 3.3-3.6, we will discuss the different feedbacks separately in more detail, at this point we give an overview of the general effects and relative importance of the different feedback processes.*

*Figure 2 shows that the temperature change in the lower stratosphere due to the direct forcing of $CO_2$ is around 3 K. We also observe that there is a cooling of about 1 K due to ozone feedback in the tropical region while there is a slight warming taking place in the summer hemispheres in both January and July. We also see that the temperature change in the lower stratosphere is influenced by the water vapour feedback and to a lesser degree by the cloud and albedo feedback.*

*In the upper stratosphere, the cooling due to the direct forcing of $CO_2$ is with about 9 degrees considerably stronger than in the lower stratosphere. The water vapour,*

*cloud and albedo feedback play no role in the upper stratosphere and mesosphere. The ozone feedback results in the positive partial temperature changes, of about 2 K. This means that the ozone feedback yields a radiative feedback that mitigates the cooling, which is due to the direct forcing of $CO_2$. This has been suggested in earlier studies, such as Jonsson et al., 2004. With CFRAM, it is possible to quantify this effect and to compare it with the effects of other feedbacks in the middle atmosphere.*

*The picture in the mesosphere is similar. The main difference is that the temperature changes are larger, note the difference of the range for the temperature change between Fig. 2c, d and Fig. 2e,f.*

*The temperature response due to dynamical feedbacks is small in global average. This can be understood as waves generally do not generate momentum and heat, but redistribute these instead (Zhu et al., 2016). However, the local responses to dynamical changes in the high latitudes are large, as we will see in section 3.3. There are some small temperature responses due to non-LTE effects as well.*

[Figure]

(a) ΔT Lower stratosphere (July)

(b) ΔT Lower stratosphere (January)

(c) ΔT Upper stratosphere (July)

(d) ΔT Upper stratosphere (January)

(e) ΔT Mesosphere (July)

(f) ΔT Mesosphere (January)

*Figure 2: The mean temperature responses to the changes in CO₂ and various feedback processes in the lower stratosphere (a,b), upper stratosphere (c, d) and the in the mesosphere (e,f) in July (a, c, e) and January (b, d, f) in the polar regions (90°S-70°S and 70°N-90°N), the tropics (20°S-20°N) and the global mean, for experiment with double CO₂ and changed SSTs.*

Figure 4 has also been changed:

[Figure]

*Figure 4: The mean temperature responses to the changes in CO₂ and various feedback processes in July (a,c) and January (b,d) in the upper stratosphere between 20  and 1 hPa, for polar regions (90°S-70°S and 70°N-90°N), the tropics*

*(20°S-20°N) and the global mean for the experiment with double $CO_2$ (a,b) and changed SSTs (c,d) separately.*

*Figure 4 shows temperature responses in the upper stratosphere for the experiment with double $CO_2$ (a,b) and changed SSTs (c,d) separately. These temperature changes were calculated as they were for Fig. 2. Again also, the 'total'-column shows the temperature changes as found by WACCM, the columns $CO_2$, $O_3$, $H_2O$, cloud, albedo, dynamics, Non-LTE shows the temperature responses as calculated by CFRAM. Error shows the difference between temperature change in WACCM and the sum of the calculated temperature responses in CFRAM.*

*We learn from this figure that the effects of the changed SSTs on the middle atmosphere are relatively small as compared to the effects of changing the $CO_2$. The changes in SSTs are responsible for large temperature changes as a result of the dynamical feedbacks especially in the winter hemispheres. A similar figure for the lower stratosphere (not shown) shows that the*
*the temperature response to the water vapour feedback is, however, almost solely due to changes in the SSTs.*

*Earlier, we discussed that the sum of the two separate temperature changes in the experiment with double $CO_2$ and changed SSTs is approximately equal to the changes observed in the combined simulation. We find that the same is true for the temperature responses to the different feedback processes.*

The paragraph beginning with line 564 is confusing. I suppose you mean ozone changes induced by $CO_2$ cooling. Ozone matters also in the infrared window.

This was indeed what was meant, the paragraph has now been rewritten.

*In addition, the vertical profile of the temperature responses to the direct forcing of CO2 and the feedbacks is shown in Figure 3. Here, one can see that the increase in $CO_2$ leads to a cooling over almost the whole middle atmosphere: an effect that increases with height. We also observe that in the summer upper mesosphere regions, the increased $CO_2$-concentration leads to a warming. The changes in ozone concentration in response to the doubling of $CO_2$ leads to warming almost everywhere in the atmosphere. In some places, this warming exceeds 5 K. In the polar winter the effect of ozone is small due to lack of sunlight.*

Shorten the paragraph beginning with line 628.

*This paragraph has now been shortened. Changing the SSTs does not lead to a change in $CO_2$-concentration, therefore the temperature response to changes in $CO_2$ is not present for the run with only changed SST (Figures not shown).*

You may improve the paragraph beginning with line 645 by the use of the textbook by Holton.

This section has been rewritten:

*As discussed in the introduction, in the middle atmosphere there is a wave driven circulation which drives the temperatures away from radiative equilibrium. Large departures from this radiative equilibrium state are seen in the mesosphere and in the polar winter stratosphere. In the mesosphere, there is a zonal forcing which yields a summer to winter transport. In the polar winter stratosphere, there is a strong forcing that consists of rising motion in the tropics, poleward flow in the stratosphere and sinking motion in the middle and high latitudes. This circulation is referred to as the Brewer-Dobson circulation (Butchart et al. 2010).*

Don't forget to mention convection around line 688.

This has now been mentioned.

*The warmer sea surface temperatures enhance the activity of transient planetary waves and orographic gravity waves in the lower and middle stratosphere, for example via the amplification of deep convection (Deckert and Damaris, 2008). The changed SSTs also leads to enhanced dissipation of planetary waves, as well as orographic and non-orographic waves in the upper stratosphere.*

Does Fig. 6 show the average of the 40yr time slice?

Yes, that is right. This is now mentioned explicitly in the text:

*The differences in the meridional component of the residual circulation ($\bar{v}^*$) between the different simulations are shown in Fig. 6. These data are averaged over the 40 years of data.*

Section 3.4 has to be rearranged and improved, the key processes are missing. The reaction $O+O_3$, the sink reaction in the Chapman chemistry, is strongly temperature dependent (see Brühl und Crutzen, 1988; Cariolle, 1983; or JPL). NO and Cl catalytic cycles matter mostly in the upper and mid stratosphere, in the mesosphere hydrogen species (e.g. OH) are most important. Check if in all calculations only $CH_3Cl$ acts as chlorine source (pre-industrial!).

Thanks for this comment and the helpful references. The section on the changes in the ozone concentration due to the changed CO2 concentration has now been rewritten and significantly shortened, as this is not new and not the main point of our work: we are interested in the temperature response as a result of the changes in the O3 concentration in WACCM.

*Ozone plays a major role in the chemical and radiative budget of the middle atmosphere. The ozone distribution in the mesosphere is maintained by a balance between transport processes and various catalytic cycles involving nitrogen oxides, $HO_x$ and $Cl_x$ radicals. In the upper stratosphere, $NO_x$ and $Cl_x$ cycles dominate (Cariolle, 1982), while OH is of utmost importance in the mesosphere (Jonsson et al., 2004).*

*In this paper, we are interested in the changes in ozone concentration induced by the increased $CO_2$ concentration and/or the changes in SST in WACCM. In the real*

*world, the ozone concentration is not only affected by the changing $CO_2$ concentration, but also by CFC and $NO_x$ emissions.*

*Fig. 8 shows the percentage changes in $O_3$-concentration when the $CO_2$-concentration and/or the SSTs change. An increase in $CO_2$, leads to an increase of ozone in most of the middle atmosphere. However, in the tropical lower stratosphere, the summer polar mesosphere, the winter and equatorial mesosphere, a decrease in ozone is seen. Fig. 8c and f show that changing the SSTs also has a significant impact on the ozone concentration. A complete account of the ozone changes is out of the scope of this paper, but the main processes responsible for ozone changes will be discussed.*

[Figure]

*Figure 8: The percentage changes in ozone concentration in July (top) and January (bottom) for (a,d) the simulation with high CO2 and SSTs, (b,e) the simulation with high CO2, (c,f) the simulation with high SSTs, all as compared to the pre-industrial control simulation, as found by WACCM. The statistical signifance and tropopause height are indicated as in Fig. 6.*

*Ozone chemistry is complex, however the ozone increase between 30 - 70 km can be understood primarily as a result of the negative temperature dependence of the reaction $O + O_2 + M \rightarrow O_3 + M$. The fractional contribution of this processes and other loss cycles involving $NO_x$, $CLO_x$ and $NO_x$ varies with altitude.*

*At altitudes between 50 and 60 km, the ozone increase is understood by less effective $HO_X$ odd oxygen destruction. The increase in $O_3$ between 45 and 50 km can be understood as the reaction rate coefficient of the sink reaction $O + O_3 \rightarrow 2O_2$ decreases. At altitudes lower than 45 km, there is a decrease of NOx abundance, which can explain the increase (Jonsson et al., 2004).*

*Schmidt et al. (2006) show that the decrease of ozone at the high latitudes in the summer mesosphere, is due to a decrease in atomic oxygen which results from increased upwelling. The decrease in $O_3$ concentration in the polar winter around 0.1 hPa is due to a stronger subsidence of NO and Cl, which are both ozone-destroying constituents.*

The statement in line 820 is quite controversial, a lot of models lead to different results here; please check.

The caveat that all models don't consistently reproduce the tropopause temperatures has been added.

*It can be seen that changing the SSTs leads to an increase in water vapour almost everywhere in the middle atmosphere (Fig. 10c and f). In WACCM, the increase in SSTs is observed to lead a higher and warmer tropopause, which can explain this increase of water vapour. However, it should be noted that models currently have a limited representation of the processes determining the distribution and variability of lower stratospheric water vapour. Minimum tropopause temperatures aren't consistently reproduced by climate models (Solomon et al., 2010; Riese et al., 2012). At the same time, observations are not completely clear about whether there is a persistent positive correlation between the SST and the stratospheric water vapour neither (Solomon et al., 2010).*

Split the averaging region in line 903ff consistent with the new figures and the revised abstract.

The abstract and the conclusion have been revised.

**4. Discussion and conclusions**

*In this study, we have applied the climate feedback response analysis method to climate sensitivity experiments performed by WACCM. We have examined the middle atmosphere response to $CO_2$ doubling with respect to the pre-industrial state. We also investigated the combined effect of doubling $CO_2$ and subsequent warming SSTs, as well as the effects of separately changing the $CO_2$ and the SSTs.*

*It was seen before that the sum of the two separate temperature changes in the experiment with only changed $CO_2$ and only changed SSTs is, at first approximation, equal to the changes observed in the combined simulation (see e.g. Fomichev et al. (2007) and Schmidt et al. (2006)). This is also the case for WACCM.*

*We have found that, even though changing the SSTs yields significant temperature changes over a large part of the middle atmosphere, the effects of the changed SSTs on the middle atmosphere are relatively small as compared to the effects of changing the $CO_2$ without changes in the SSTs.*

*We find that the temperature change due to the direct forcing of $CO_2$ increases with increasing height in the middle atmosphere. The temperature change in the lower stratosphere due to the direct forcing of $CO_2$ is around 3 K while in the upper stratosphere, the cooling due to the direct forcing of $CO_2$ is with about 9 K considerably stronger than in the lower stratosphere. In the mesosphere, the cooling due to the direct forcing of $CO_2$ is even stronger.*

*Ozone responds to changes in respond to changes in $CO_2$ and/or SSTs due to changes in chemical reaction rate constants and due to the strength of the up- and downwelling. The temperature changes caused by these changes in ozone*

*concentration generally mitigate the cooling caused by the direct forcing of $CO_2$. However, we have also seen in that in the tropical lower stratosphere and in some regions of the mesosphere, the ozone feedback cools these regions further.*

*We also have seen that the global mean temperature response due to dynamical feedbacks is small, while the local responses to the changes in dynamics are large. Doubling the $CO_2$ leads to a stronger summer-to-winter-pole flow, which leads to cooling of the summer mesosphere and a warming of the winter mesosphere. Changing the SSTs weakens this effect in the mesosphere, but leads to temperature changes in the stratosphere and lower mesosphere.*

*The temperature change in the lower stratosphere is influenced by the water vapour feedback and to a lesser degree by the cloud and albedo feedback, while these feedbacks play no role in the upper stratosphere and the mesosphere.*

*It would also be interesting to investigate the exact mechanisms behind the feedback processes in more detail. Some processes can influence the different feedback processes, such as ozone depleting chemicals influencing the ozone concentration and thereby the temperature response of this feedback. Other studies have shown that a surface albedo change, which is associated with sea ice loss, can influence the middle atmosphere dynamics, which in turn influences the temperature response (Jaiser et al., 2013). The CFRAM cannot unravel the effects of these different processes.*

*There is also a need for a better understanding of how different feedbacks in the middle atmosphere affect the surface climate. As discussed in the introduction, the exact importance of ozone feedback is currently not clear While this paper focused on the temperature changes in the middle atmosphere, similar analysis can be done to quantify the effects of feedbacks on the surface climate.*

*In conclusion, we have seen that CFRAM is an efficient method to quantify climate feedbacks in the middle atmosphere, although there is a relatively large error due to the linearization in the model. The CFRAM allows for separating and estimating the temperature responses due an external forcing and various climate feedbacks, such as ozone, water vapour, cloud, albedo and dynamical feedbacks. More research into the exact mechanisms of these feedbacks could help us to understand the temperature response of the middle atmosphere and their effects on the surface and tropospheric climate better.*

**3 Technical corrections**

Please define all formula letters in line 298.

This has now been done:

$$\Delta(\vec{S} - \vec{R}) = \Delta(\vec{S} - \vec{R})_{(WACCM)} * mass_k * c_p \qquad (5)$$

In which $\Delta(\vec{S} - \vec{R})$ is the difference in the shortwave radiation $\overrightarrow{(S)}$ and longwave radiation $\overrightarrow{(R)})$ between the perturbation run and the control run as a flux in $Wm^{-2}$,

while $\Delta(\vec{S} - \vec{R})_{(WACCM)}$ is this difference as heating rate in Ks$^{-1}$ in WACCM, with $mass_k = \frac{p_{k+1} - p_k}{g}$ with p in Pa and $c_p = 1004$ J $kg^{-1}$ $K^{-1}$ the specific heat capacity at constant pressure.

In line 546 something is missing or twice. Right, this has been corrected.

Add 'high latitude' in line 616. This has been added.

Typo in line 688. This has been corrected.

Captions of Fig. 7 to 11 can be shortened (tropopause as in Fig. 6).

Thanks for this comment, this has been implemented now.

**4 References**

Brühl, C. and P.J. Crutzen: Scenarios of possible changes in atmospheric temperatures and ozone concentrations due to man's activities, estimated with a one-dimensional coupled photochemical climate model. Climate Dynamics, 2: 173-203, 1988.

Cariolle, D.: The ozone budget in the stratosphere: results of a one-dimensional photochemical model. Planet. Space Sci., 31: 1033-1052, 1983.

Others see discussion paper.

---

## Short Comment (SC3) · 20 Apr 2020

I am sorry to say that I am still not happy with this paper in its present form.

First of all, we would like to thank the reviewer for his or her comments. We do point out that almost all of these questions were already addressed in the pre-review for this paper. Large parts of the paper have been rewritten (see also comment on reviewer #2) and we hope that the reviewer is happier with the paper.

The paper is missing a clear message and many statements remain vague.

What we do in this paper is to apply a new method to quantify the temperature responses to different feedback processes that arise in response to changing the $CO_2$-concentration. This is one of the first studies in which it is calculated how much of the temperature change in a specific place in the atmosphere is attributed to which feedback process. The method we applied here can quantify the temperature response, but to provide a complete explanation of all the responses and the exact mechanism behind all the feedback processes is outside the scope of this paper.

For example in the abstract: "feedback processes" (l 51) but which processes?

The "feedback processes" we mean chemical, physical and dynamical processes, which can feedback to the radiation and further change the temperature, in our study these processes arise due to a change in CO2 concentration.

We understand that the formulation in the abstract was maybe not very clear and the abstract has now been rewritten.

*The importance of the middle atmosphere for surface and tropospheric climate is increasingly realized. In this study, we aim at a better understanding of climate feedbacks in response to a doubling of $CO_2$ in the middle atmosphere using the climate feedback response analysis method (CFRAM). This method allows one to calculate the partial temperature changes due to an external forcing and climate feedbacks in the atmosphere. It has the unique feature of additivity, such that these partial temperature changes are linearly addable. We find that the temperature change due to the direct forcing of $CO_2$ increases with increasing height in the middle atmosphere, with the cooling in the upper stratosphere about three times as strong as in the lower stratosphere. The ozone feedback yields a radiative feedback that generally mitigates this cooling, however in the tropical lower stratosphere and in some regions of the mesosphere, the ozone feedback cools these regions further. The temperature response due to dynamical feedbacks is small in global average, although the temperature changes due to the dynamical feedbacks are large locally. The temperature change in the lower stratosphere is influenced by the water vapour feedback and to a lesser degree by the cloud and albedo feedback, while these feedbacks play no role in the upper stratosphere and the mesosphere. We find that*

*the effects of the changed SSTs on the middle atmosphere are relatively small as compared to the effects of changing the $CO_2$. However, the changes in SSTs are responsible for large temperature changes as a result of the dynamical feedbacks and the temperature response to the water vapour feedback in the lower stratosphere is almost solely due to changes in the SSTs. As CFRAM has not been applied to the middle atmosphere in this way, this study also serves to investigate the applicability as well as the limitations of this method. This work shows that CFRAM is a very powerful tool to study climate feedbacks in the middle atmosphere. However, it should be noted that there is a relatively large error term associated with the current method in the middle atmosphere, which can be for a large part be explained by the linearization in the method.*

CFRAM (l 54) is not known to me; I am not sure how helpful this statement is without further explanation to the readers of ACP.

Further explanation of the method is now added in the abstract (see above).

I would like to refer to Taylor et al., 2013; Song and Zhang, 2014; Hu et al., 2017; Zheng et al., 2019, who have used CFRAM as a practical diagnostic tool to analyse the role of various forcing and feedback studying surface climate change.

Response to CO2 doubling but at what time has the doubling been reached – would that not be important for the issue of stratospheric ozone? Ozone feedback is mentioned (l 63), but what is assumed for ozone in the upper stratosphere? We know upper stratospheric ozone is "recovering" over the coming decades (WMO, 2018); is this the point here?

We are not speaking here about the changes in $O_3$-concentration due to the ozone hole, but rather changes in ozone concentration that are resulting from changes in the $CO_2$ concentration. The ozone concentration in the control run is for pre-industrial conditions. We change the CO2 and/or the SSTs in the model and compare the two equilibrium states. In runs with the changed CO2 and/or the SSTs the ozone concentration is changed due to the changes in CO2 concentration only. The model has interactive chemistry which calculates the amount of ozone concentration.

And a "warming by 1.5 K, but in which region? Changes in dynamics play a large role (l 66) but which changes, which role? And above 0.1 hPa, which is certainly a region where an ozone feedback is expected. Above 0.1 hpa
is above 60 km; this is the mesosphere.

In the earlier version of the paper, average temperature changes over the whole middle atmosphere region were taken. Although this can learn us something, we have now divided Fig. 2 for the regions: lower, upper stratosphere and mesosphere.

Several tropospheric issues are "of minor importance" (l 69), but why is this issue discussed in an abstract of a paper on the "middle atmosphere"?

We investigate the temperature responses to feedback processes. Some of these processes might have an effect in the middle atmosphere, and we see that this is indeed the case for the lower stratosphere, however not for the regions above.

Further comments:

l. 85-90: It should be clearly said that the middle atmosphere is *not* in radiative equilibrium in most regions; downwelling in the polar regions in winter is part of the BD circulation, and not only an "example". See e.g. Dunkerton, JAS, 1978.

This formulation was written similarly in the PhD thesis 'The middle atmosphere and its sensitivity to climate change' by Andreas Jonson, which I read several times while working on this paper (https://www.diva-portal.org/smash/get/diva2:198863/FULLTEXT01.pdf).

I have changed this paragraph to emphasize the role of dynamics for the temperatures in middle atmosphere.

*The circulation in the troposphere is thermally driven, however this is quite different for the middle atmosphere. The air in the middle atmosphere is out of reach for convection and is not in direct contact with the Earth's surface, which means that the middle atmosphere is dynamically stable. In the absence of eddy motions the zonal-mean temperature would relax to a radiatively determined state. However, a wave driven motions of the air drive the flow away from this state of radiative balance and in this way determine the heating and cooling patterns in the middle atmosphere (Shepherd, 2010).*

l 115-122: Ozone is mentioned here, but it is well established that stratospheric ozone responds to changing halogen levels (WMO 2018). This aspect can not be ignored in this study.

While the reaction of ozone to CFCs and the consequent ozone hole are very important and interesting topics in itself, this is not the topic of this study. What we do here is investigate the temperature changes that arise from the direct forcing of $CO_2$ and the feedback processes that result from this increase in $CO_2$. We have not investigate the effect of the depletion/recovery of ozone.

Sec. 2.3: Is this a new formulation of CFRAM or is this section just reiterating a technique used before? It looks like a new description here as there are no references to previous description of CFRAM at the top of sec 2.3.

Partly this is a new formulation. The method is based on the CFRAM method as described in Lu and Cai (2009), however the method needed to be adjusted in order to be applicable on WACCM output data. I referred to Lu and Cai at the end of the section, but have now added an additional reference at the beginning of the section.

l 731-735: No, the ozone concentration is *not* controlled by the Chapman reactions "for a large part". Depending on altitude, NOx and HOx cycles are important; this is

well known (check the textbook you are citing). Also chlorine compounds are relevant.

This part has been rewritten:

*Ozone plays a major role in the chemical and radiative budget of the middle atmosphere. The ozone distribution in the mesosphere is maintained by a balance between transport processes and various catalytic cycles involving nitrogen oxides, $HO_x$ and $Cl_x$ radicals. In the upper stratosphere, $NO_x$ and $CL_x$ cycles dominate, while hydrogen species are of most importance in the mesosphere (Cariolle, 1982).*

l 774: The direct influence is only possible where the chemistry not too fast, again check the Brasseur and Solomon textbook.

The section on the changes in the ozone concentration due to the changed CO2 concentration has now been rewritten and significantly shortened, as this is not new and not the main point of our work: we are interested in the temperature response as a result of the changes in the O3 concentration in WACCM.

Sec. 3.5: The feedback for stratospheric H2O is not that simple, see eg. Solomon et al., Science, 2010; Riese et al, JGR, 2012).

The caveat that all models don't consistently reproduce the tropopause temperatures has been added.

*It can be seen that changing the SSTs leads to an increase in water vapour almost everywhere in the middle atmosphere (Fig. 10c and f). In WACCM, the increase in SSTs is observed to lead a higher and warmer tropopause, which can explain this increase of water vapour. However, it should be noted that models currently have a limited representation of the processes determining the distribution and variability of lower stratospheric water vapour. Minimum tropopause temperatures aren't consistently reproduced by climate models (Solomon et al., 2010; Riese et al., 2012). At the same time, observations are not completely clear about whether there is a persistent positive correlation between the SST and the stratospheric water vapour neither (Solomon et al., 2010).*

Several points are mixed here. There might be a change in stratospheric water vapour based on changing climatic conditions and this change could have an impact on local heating/cooling but also an effect of the surface radiative forcing. These issues should be disentangled.

The CFRAM method here calculates the temperature change on the basis of the radiative changes due to changes in water vapour alone. This method doesn't allow further disentanglement. Our results show that that the regions where there is an increase in the water vapour, there is a cooling, and vice versa. The water vapour feedback as calculated here only takes into account radiative processes, if the water vapour feedbacks on the temperature via dynamical feedbacks this would be shown as a result of the dynamical feedback.

l 820: warmer tropopause: also in the tropics? (where water vapour is entering the stratosphere). Would you not expect higher SSTs causing more wave activity and thus a stronger tropical upwelling? Why is this argument not correct?

Yes, WACCM a warmer tropopause in the tropics as well (the tropopause temperature comes as an output of the model). See here the changes in tropopause temperature for the different latitudes in July.

[Figure]

A study by Deckert and Dameris (2008) indeed shows that higher tropical SSTs can indeed strengthen the tropical upwelling into the stratosphere. However, as explained by Solomon et al. (2010) the transport of the water vapour in the stratosphere is mainly a function of the tropopause temperature. We see that this is changing, which can explain what is seen in the model.

l 815-816: Is this reduction expected from simple water vapour equilibrium (over ice) arguments? Citations?

I assume the reviewer is referring to an earlier version of the manuscript. A suggestion was made, but based on the comments of other reviewers this was taken out.

Sec 3.6 discusses feedbacks that "play only a very small role" for the middle atmosphere. Fig 12 shows that the middle atmosphere is largely grey (zero effect). But the caption tells me that the comparison is to pre-industrial conditions, so the reported "delta" is due to a change relative to pre-industrial? Then one would expect a larger signal – correct? And the discussion starting in l. 810 is not discussing "pre-industrial"; this is confusing.

Yes, the delta T is the temperature change between the run with the changed SSTs and/or CO2 and pre-industrial conditions. Exactly, we see that these feedbacks (albedo and cloud) basically only effect the temperatures in the lower stratosphere and not in the upper stratosphere and the mesosphere. It would have been interesting to see a larger signal higher up in the atmosphere, but it is not there.

I speak about middle atmosphere climate sensitivity which is here used to refer to how much the middle atmosphere will cool or warm after the doubling of CO2 as compared to the pre-industrial state. Figure 12 is no different from Fig. 5 ,7, 9 and 11. It shows the temperature response to the changes in SSTs for the run with high SSTs as compared to pre-industrial conditions.

L 964-972: these lines seem to describe the overall conclusions of the paper; when I read these lines they seem to tell me that CFRAM is okay, but that some refinement is necessary. This is a rather technical statement (which would be more helpful if statements like "some" would be more specific). Most importantly however, the paper promises to talk about "quantifying (!) climate feedbacks in the middle atmosphere" – in my view this has not been achieved in this manuscript.

This is the first study in which it is calculated how much of the temperature change in a specific place in the middle atmosphere is attributed to which feedback process in response to a doubling of CO2. We succeeded in performing the calculations and indeed quantifying these temperature responses. Fig 2, 3, 4, 5, 7, 9, 11 and 12 show the quantification we aimed at achieving. Another goal of the paper was to investigate the applicability of this traditional form of method in the middle atmosphere (contrary to what is done by Zhu et al.). We have learnt that it can be used, but indeed it is not perfect. The linearization is a fundamental part of the method and leads inevitably to errors.

References:

* The current report on stratospheric ozone is WMO 2018, I suggest using this most current information (see above)

Brasseur and Solomon 2005: this is an excellent text book but might not give the most up to date information required here on upper stratospheric ozone

These references are no longer there.

---

## Editor Decision (ED1)

**Specific comments:**

Title: Since CFRAM play a major role in your study it should also appear in the title.

Key points: Here you clearly summarize your study, however, as you write it here it does not appear in the abstracts. Since ACP does not use key points these will be simply lost after publication. Therefore, I would strongly suggest that you include these in your abstract.

Abstract: An abstract should be clearly written and summarize the idea and results of a study. Here are too many weird or complicated sentences that distract from the content of the study. Further, not all what you have done is summarized here. Therefore, the abstract should be revised.

The abbreviation "CO2" is used throughout the paper but has never been introduced.

P2, L51-52: Already the first sentence is rather weird formulated. I would never "increasingly realized". Please rephrase.

P2, L58-59: "We find the……" This sentence is also not clear. I would suggest to split it into two sentences.

P2, L72-73: Same here. It would be better to split this sentence.

General comment: Why do we consider doubled CO2 atmospheres? One sentence should be included to motivate why such scenarios are of interest.

P2, L89: "ozone is responsible for the existence of the stratosphere…….". This may be indirectly correct, but it sounds really weird and thus should be rephrased.

P2, L93-95: Also this is a really weird paragraph. This paragraph should be completely rewritten.

P3, L102: I am not aware of that. These are generally roughly parameterized. Which processes are you referring to? Please give some examples.

P3, L109ff: How is ozone represented in the climate models? This should be added.

P3, L116ff: How does CO2 affect the middle atmosphere? What role does ozone play in this context?

General comment on the introduction: The first part of the introduction (L84-L131) is really not well written. I do not see here the relation to your study. There is unfortunately no clear line. Therefore, I would suggest to completely revise this part.

P4, L176: Nothing mentioned here that the feedbacks are discussed separately.

P4, L184: What is the resolution in the boundary layer? As you write it sounds as they use a different resolution in the boundary layer then in the remainder of the atmosphere.

P5, L219: "……from a double CO2 equilibrium simulation by CESM." Why has this been done? From the description here I count three simulations, but in the table and later there are four simulations.

P5, L223: Add the years for which the simulation has been performed.

Table 1: What is "PI"? What SSTs have been used? Please add numbers in the table.

P7, L287: we can calculate as -> can be calculated since ……

General on this section: The description is much too long for the main part of the manuscript. You should shorten this Section and put the other parts to an Appendix.

General on Section 3: Also here I would suggest some restructuring. I would add subsections, on for the temperature responses and one for the feedbacks or directly use different Sections.

P10, L448: In section 2.2, it was….. -> As described in Section 2.2 four experiments…..

P10, L453: Are you using a 40 year mean? This should be clearly stated in the manuscript.

P10, L455: Add references to the earlier studies.

P11, L478-479: "all as compared the pre-industrial control simulation". That means the difference between these? Clearly state this. As it is written now it is rather confusing.

P11, L488: Why can only the radiative feedback been calculated?

P12, L498: It would be quite helpful if you would give your experiments names as it usually done in the modelling community and then use these names throughout the manuscript. It would make it much easier to follow which experiment you are actually discussing.

P12, L500-501: "the pressure level of which is an output of WACCM". What do you mean with that? The tropopause is an output level? But it is always at a different height.

P12, L501: Why 24 hPa?

P12, L505: Why do you not use temperature to derive the location of the tropopause?

P12, L506: Why is a mass weighting important? What is the error/uncertainty of not doing this?

General comment: How do your results agree with previous studies? Are your feedbacks higher or lower?

P13, L559: Something missing here after "radiative"?

General comment: Discuss a bit more in which altitude/latitude region the impact is highest/lowest and give numbers.

Figure 2: I see here the highest feedbacks for the mesosphere. This is not discussed like that in the main text. Further, you use different y-axis scale which masks a bit the differences between the atmospheric regions. At least you should mention the different y-scales in the figure caption.

Figure 3: Here, a strong cooling due to CO2 is visible while all others rather show a warming. This is not discussed like this in the text. At least it is not clearly stated.

P17, L606, Figure 4: Why has the upper stratosphere picked? Changes seem to be highest in the mesosphere. Why has the separation not been done there?

P17, L621: Thus, SSTs are important for the water vapour feedback on temperature, butt lower for the CO2 feedback on temperature. This could also be more clearly stated.

P17, L631: Temperature direct response -> Direct temperature response

P18, L669: Add also the paper by Brewer, 1949, QJRMS

P19, L681ff: Most of this paragraph rather belongs to the introduction.

P21, L757ff: Please quantify your results and give the percentages.

P21, L771: What symbol has been used for the statistical significance? Please add that to the figure caption.

P21, L780: Before you always wrote "ozone" but now you write "O3" without introducing the abbreviation.

P21, L787-789: Ozone is destroyed in polar winter in the lower stratosphere, not at 0.1 hPa.

P22, L804 and several other occasions: the 2 in CO2 should be written as subscript.

General comment: Discuss also the statistical significance. Which results do you derive with which significance?

Figure 10 caption: Add information what symbol has been used for the statistical significance.

P23, L853: Sentence with "This has been explained…….." is too complicated and should be rephrased, e.g. This can be explained by a ……leading to an …...

P24, L888: Discuss Figure 12 a bit more. The figures should be first described. What is shown, what do you see…….

P24, L893: Which cooling are you talking about? Up to date cooling for future cooling?

P26, L924: Add a number. How much stronger?

General comment: How do the results agree with previous studies? What is the importance of you results for future predictions or climate change etc.? This should also be discussed.

To summarize you results and for having it easier with the discussion you could make a table/matrix where you mark which feedbacks are important in which altitude/latitude region.

**Technical corrections:**

P2, 75: Add before after "in this way".

P4, L164: are -> were

P4, L169: by -> with

P5, L207: earth -> Earth

P5, L218: SSTs forcing -> SST forcings

P7, L280: delete "to" before balance

P12, L505: don't -> do not

P15, L577: leads to -> leads to a

P17, L619: One "the" obsolete.

P18, L681: synoptic -> synoptic scale

P21, L771: signifance -> significance

P21, determinged -> determined

P21, L776: CLOx -> ClOx

P23, L824: to lead a -> to lead to a

P23, L828: Delete "in WACCM", you have already written it at the beginning of the sentence.

P23, L838: aren't -> are not

P23, L841: "neither" obsolete? Or should it rather read "either".

P23, L848: found by -> simulated by or simulated with

P25, L911: at a first -> as a first

P26, L939: leads to -> leads to a

---

## Author Response (AR2)

*We thank the editor for her extensive comments.*

Specific comments:

**Title**: Since CFRAM play a major role in your study it should also appear in the title.

The title has now been changed to *"Using the climate feedback response method to quantify climate feedbacks in the middle atmosphere in WACCM"*

**Key points**: Here you clearly summarize your study, however, as you write it here it does not appear in the abstracts. Since ACP does not use key points these will be simply lost after publication. Therefore, I would strongly suggest that you include these in your abstract.

These points are included in the abstract.

**Abstract**: An abstract should be clearly written and summarize the idea and results of a study. Here are too many weird or complicated sentences that distract from the content of the study. Further, not all what you have done is summarized here. Therefore, the abstract should be revised. The abbreviation "CO2" is used throughout the paper but has never been introduced.

The abbreviation $CO_2$ has now been introduced in the abstract. The abstract has been significantly revised (see below).

P2, L51-52: Already the first sentence is rather weird formulated. I would never "increasingly realized". Please rephrase.

This sentence has been rephrased.

P2, L58-59: "We find the……" This sentence is also not clear. I would suggest to split it into two sentences.

As expected, our results show that the direct forcing of $CO_2$ cools the middle atmosphere. This cooling becomes stronger with increasing height: the cooling in the upper stratosphere is about three times as strong as the cooling in the lower stratosphere.

P2, L72-73: Same here. It would be better to split this sentence.

However, the changes in SSTs are responsible for dynamical feedbacks that cause large temperature changes. Moreover, the temperature response to the water vapour feedback in the lower stratosphere is almost solely due to changes in the SSTs.

General comment: Why do we consider doubled CO2 atmospheres? One sentence should be included to motivate why such scenarios are of interest.

This has now been added*: "A better understanding of the middle atmosphere and how it reacts to the current increase of the concentration of carbon dioxide ($CO_2$) is therefore necessary."*

**Abstract**

*Over recent decades it has become clear that the middle atmosphere has a significant impact on surface and tropospheric climate. A better understanding of the middle atmosphere and how it reacts to the current increase of the concentration of carbon dioxide is therefore necessary. In this study, we investigate climate feedbacks in response to a doubling $CO_2$ in the middle atmosphere using the climate feedback response analysis method (CFRAM). With this method, one can calculate the partial temperature changes due to an external forcing and climate feedbacks in the atmosphere. As this method has the unique feature of additivity, these partial temperature changes are linearly addable. In this study, we discuss the direct forcing of $CO_2$ and the effects of the ozone, water vapour, cloud, albedo and dynamical feedbacks.*

*As expected, our results show that the direct forcing of $CO_2$ cools the middle atmosphere. This cooling becomes stronger with increasing height: the cooling in the upper stratosphere is about three times as strong as the cooling in the lower stratosphere. The ozone feedback yields a radiative feedback that mitigates this cooling in most regions of the middle atmosphere. However, in the tropical lower stratosphere and in some regions of the mesosphere, the ozone feedback has a cooling effect. The increase in $CO_2$-concentration causes the dynamics to change. The temperature response due to this dynamical feedback is small in the global average, although there are large temperature changes due to this feedback locally. The temperature change in the lower stratosphere is influenced by the water vapour feedback and to a lesser degree by the cloud and albedo feedback. These feedbacks play no role in the upper stratosphere and the mesosphere. We find that the effects of the changed SSTs on the middle atmosphere are relatively small as compared to the effects of changing the $CO_2$. However, the changes in SSTs are responsible for dynamical feedbacks that cause large temperature changes. Moreover, the temperature response to the water vapour feedback in the lower stratosphere is almost solely due to changes in the SSTs. As CFRAM has not been applied to the middle atmosphere in this way before, this study also serves to investigate the applicability as well as the limitations of this method. This work shows that CFRAM is a very powerful tool to study climate feedbacks in the middle atmosphere. However, it should be noted that there is a relatively large error term associated with the current method in the middle atmosphere, which can be for a large part be explained by the linearization in the method.*

**Introduction:** P2, L89: "ozone is responsible for the existence of the stratosphere…….". This may be indirectly correct, but it sounds really weird and thus should be rephrased.

*The introduction has been thoroughly rewritten and this part is no longer there (see below).*

P2, L93-95: Also this is a really weird paragraph. This paragraph should be completely rewritten.

*The introduction has been thoroughly rewritten and this part is no longer there (see below).*

P3, L102: I am not aware of that. These are generally roughly parameterized. Which processes are you referring to? Please give some examples.

*The introduction has been thoroughly rewritten and this part is no longer there (see below).*

P3, L109ff: How is ozone represented in the climate models? This should be added.

The introduction has been rewritten and the study of Nowack has been described more clearly:
*Nowack et al. (2015) has found that there is an increase in global mean surface warming of about 1 °C when the ozone is prescribed at pre-industrial levels, as compared with when it is evolving in response to an abrupt 4xCO$_2$ forcing.*

P3, L116ff: How does CO2 affect the middle atmosphere? What role does ozone play in this context?

The introduction has been rewritten and the importance of the changes in CO$_2$ and O$_3$ has now been clearer.

General comment on the introduction: The first part of the introduction (L84-L131) is really not well written. I do not see here the relation to your study. There is unfortunately no clear line. Therefore, I would suggest to completely revise this part.

*The introduction has been thoroughly rewritten.*

P4, L176: Nothing mentioned here that the feedbacks are discussed separately.

This has been added.

Please find below the rewritten introduction:

[revised manuscript text omitted]

P4, L184: What is the resolution in the boundary layer? As you write it sounds as they use a different resolution in the boundary layer then in the remainder of the atmosphere.

The lowermost levels in WACCM are 1010, 992.556, 970, 929, 867, 788 hPa. The height of this levels of course dependent on the temperature and therefore it wasn't written in the paper. As the boundary layer is not of major importance for this paper, we have now rephrased this as follows.

*"The Whole Atmosphere Community Model (WACCM) is a chemistry-climate model, which spans the range of altitudes from the Earth's surface to about 140 km (Marsh et al., 2013). The model consists of 66 vertical levels with irregular vertical resolution, which ranges from ~1.1 km in the troposphere, 1.1–1.4 km in the lower stratosphere, 1.75 km at the stratosphere and 3.5 km above 65 km. The horizontal resolution is 1.9° latitude by 2.5° longitude.*

P5, L219: "……from a double CO2 equilibrium simulation by CESM." Why has this been done? From the description here I count three simulations, but in the table and later there are four simulations.

The use of prescribed SSTs is common for middle atmosphere CCMs (e.g. *Fomichev et al*., 2007).

Four experiments have been performed. Section 2.2 has now been rewritten to make clear what has been done (see below).

*In this study, the F_1850 compset (component set) of the model is used, i.e. the model assumes pre-industrial (PI) conditions. This compset simulates an equilibrium state, which means that it runs a perpetual year 1850. Four experiments have been performed (see Table 1) for this study.*

P5, L223: Add the years for which the simulation has been performed.

In this study, the F_1850 compset (component set) of the model is used, i.e. the model assumes pre-industrial (PI) conditions.

This means that we run the model for a perpetual year 1850, but then with the $CO_2$ concentration and/or the SSTs changed.

Table 1: What is "PI"? What SSTs have been used? Please add numbers in the table.

PI stands for pre-industrial, this has now been made clear in the text. The experiments are now referred to with a letter number combination.

**2.2 Experimental set-up**

*In this study, the F_1850 compset (component set) of the model is used, i.e. the model assumes pre-industrial (PI) conditions. This compset simulates an equilibrium state, which means that it runs a perpetual year 1850. Four experiments have been performed for this study (see Table 1).*

*Experiment C1 is the control run, with the pre-industrial $CO_2$ concentration (280 ppm) and forced with pre-industrial ocean surface conditions such as sea surface temperature and sea ice (referred to SSTs from now on). These SSTs are generated from the CMIP5 pre-industrial control simulation by the fully coupled Earth system model CESM. The atmospheric component of CESM is the same as WACCM, but does not include stratospheric chemistry (Hurrel et al., 2013). This latter aspect is not considered in this study.*

*Experiment C2 represents the experiment with the $CO_2$ concentration doubled as compared to the pre-industrial state (560 ppm) and forced with the same pre-industrial SSTs as in experiment C1. In WACCM, the $CO_2$-concentration does not double everywhere in the atmosphere. Only the surface level $CO_2$ mixing ratio is doubled, and elsewhere in the atmosphere is calculated according to WACCM's chemical model.*

*The compset used in this experiment and all the following ones is still F_1850, which means that other radiatively and chemically active gases, such as ozone, will change only because of the changes in the $CO_2$-concentration, due to WACCM's interactive chemistry. Chlorofluorocarbons (CFCs), which have a major impact on the ozone concentration in the real atmosphere, don't play a role in our experiments.*

*In experiment S1, we simulate the scenario, in which there is the SSTs forcing from the coupled CESM for double $CO_2$ condition, but the pre-industrial $CO_2$-concentration of 280 ppm. S2 represents the experiment with the $CO_2$-concentration in the atmosphere doubled to 560 ppm and the SSTs prescribed for the double $CO_2$-climate. Experiment C1, C2, S1 and S2 will be also referred to hereafter by PI, the simulation with high CO2, the simulation with high SSTs and the simulation with high $CO_2$ and SSTs, respectively.*

*The experimental setup of this study is similar to the setup performed with the Canadian Middle Atmosphere Model (CMAM) by Fomichev et al. (2007) and with the Hamburg Model of the Neutral and Ionized Atmosphere (HAMMONIA) by Schmidt et al. (2006). The HAMMONIA model is coupled to the same chemical model as WACCM: MOZART3. The setup in their study is similar, however, in their study, they double the $CO_2$-concentration from 360 ppm to 720 ppm, while in our study, we double from the pre-industrial level of $CO_2$ (280 ppm).*

*Note that experiment S1 and C2 are not representing scenarios that could happen in the real atmosphere. These experiments have been used to study the effect of the SSTs separately. Experiment S2 doesn't take into account other (anthropogenic) changes in the atmosphere not caused by changes in the $CO_2$-concentration and the SSTs.*

*All the simulations are run for 50 years, of which the last 40 years are used for analysis. In the all results shown, we have used the 40 year mean of our model data.*

*Table 1. Set-up of the model experiments.*

| Experiment | $CO_2$ | SSTs from CESM equilibrium run |
|---|---|---|
| C1 | 280 ppm | PI control |
| C2 | 560 ppm | PI control |
| S1 | 280 ppm | Double $CO_2$ run |
| S2 | 560 ppm | Double $CO_2$ run |

P7, L287: we can calculate as -> can be calculated since ……

This has been rephrased.

**General on this section**: The description is much too long for the main part of the manuscript. You should shorten this Section and put the other parts to an Appendix.

*We have shortened the method section substantially by putting a large part of the method in the appendix section, where we give the complete formulation of CFRAM diagnostics using outputs of WACCM.*

General on Section 3: Also here I would suggest some restructuring. I would add subsections, on for the temperature responses and one for the feedbacks or directly use different Sections.

Section 3 has now been restructured. Section 3.1 has become section 3. Figs. 5-12 have now moved to section 4 (meridional-vertical profiles of partial temperature changes), which has 5 sub-sections, which discuss the temperature response to the different feedbacks.  Fig. 2-4 are moved to section 5 (regional and global means of partial temperature changes due to feedbacks).

P10, L448: In section 2.2, it was….. -> As described in Section 2.2 four experiments…..

This has been changed as suggested.

P10, L453: Are you using a 40 year mean? This should be clearly stated in the manuscript.

We have added this now both the method section as well as in section 3.

*For this figure, as well as for all the results shown in this paper, we have used the 40 year mean of our data.*

P10, L455: Add references to the earlier studies.
References have now been added.

*In line with what was shown in earlier studies (e.g. Akmaev, 2006; Fomichev et al., 2007), we observe that an increase in $CO_2$ causes a cooling in the middle atmosphere with the exception of the cold summer upper mesosphere region.*

P11, L478-479: "all as compared the pre-industrial control simulation". That means the difference between these? Clearly state this. As it is written now it is rather confusing.

Yes, that is right. This has been rewritten for this figure as well as for all the other ones which had a similar formulation.

*Changes in the zonal and monthly mean temperature (K) for July (top) and January (bottom) due to (a,d) combined effect of the $CO_2$ increase and SSTs changes (experiments C2 -B1), (b,e) the doubling of the atmospheric $CO_2$-concentration (experiments B2 - B1) and the (c,f) SSTs (experiments C1 - B1).*

P11, L488: Why can only the radiative feedback been calculated?

This has now been explained in the text.

*This can be understood as we use the Fu-Liou radiative transfer model (Fu and Liou, 1992, 1993) to do offline calculations of the total local thermal equilibrium (LTE) radiative heating rate perturbation fields between the control experiment C1 and one of the other three experiments (i.e, C2, or S1, or S2). We use the standard outputs of atmospheric compositions (e.g., $CO_2$ and $O_3$) and thermodynamic fields (e.g., pressure, temperature, water vapour, clouds, surface albedo) as well as partial LTE radiative heating rate perturbation fields due to perturbations in individual atmospheric composition or thermodynamic fields (e.g., the terms on the right hand side of (A.9) except the first term).*

*We use the difference between the offline calculation of the total LTE radiative heating rate perturbations and the original total LTE radiative heating rate perturbations derived directly from the standard WACCM outputs as the error term of our offline LTE radiative heating perturbations. We note that the standard WACCM output fields also include non-LTE radiative heating fields, but do not include non-radiative heating rates. Therefore, we use the sum of the total LTE radiative heating rate perturbations and non-LTE radiative heating fields derived from the standard WACCM output fields to infer non-radiative heating rate perturbations under the equilibrium condition, namely Eq. (A.8).*

P12, L498: It would be quite helpful if you would give your experiments names as it usually done in the modelling community and then use these names throughout the manuscript. It would make it much easier to follow which experiment you are actually discussing.

Experiment numbers (C1, C2, S1 and S2) have been added throughout the paper.

P12, L500-501: "the pressure level of which is an output of WACCM". What do you mean with that?The tropopause is an output level? But it is always at a different height. P12, L505: Why do you not use temperature to derive the location of the tropopause?

WACCM has an output field which is the tropopause height in hPa (in the same way it outputs the temperature, water vapour etc.). This is a field that indeed varies with latitude, longitude and time. This has been taken into account, as can be seen in Fig. 2: the tropopause height varies with latitude (we have averaged over the time and longitude). WACCM also has the tropopause temperature and the temperature as separate output fields. It is indeed also possible to use the temperature profile to find the position of the tropopause, which would give you the same pressure level. We have taken the output of WACCM directly, as it is more convenient.

This has now been written in the text more clearly:

*The pressure level of the tropopause is simulated in WACCM for each latitude and longitude. We use this pressure level to demarcate between the troposphere and stratosphere.*

P12, L501: Why 24 hPa?

We consider 24 hPa as a crude estimate for the boundary between the lower and upper stratosphere. This has now also been added to text.

P12, L506: Why is a mass weighting important? What is the error/uncertainty of not doing this?

I added this sentence because a reviewer asked for this. But there is no error or uncertainty involved in such average calculations regardless of whether we consider mass weighting or not. The mass weighting would emphasize the values in the lower levels much more heavily as the mass decreases with height exponentially. By not considering mass weight, the vertical average is just simple arithmetical mean and we can directly compare the vertical average values with their counterparts showing in the vertical-latitude cross-section diagrams.

General comment: How do your results agree with previous studies? Are your feedbacks higher or lower?

The CFRAM method allows to calculate for each location in the middle atmosphere, how much of the temperature change is due to which process. No other method before could do that. We have made this more explicit in the text.

*The ozone feedback generally yields a radiative feedback that mitigates the cooling, which is due to the direct forcing of $CO_2$. This has been suggested in earlier studies, such as Jonsson et al., 2004, Dietmüller et al., 2014. With CFRAM, it is possible to quantify this effect and to compare it with the effects of other feedbacks in the middle atmosphere. Note that no other method before has been able to quantify how much*

*of the temperature change in the middle atmosphere is due to the different feedback processes before.*

P13, L559: Something missing here after "radiative"?

That is correct. What we meant to write was as follows:

*Another part of the error is due to the fact that the radiative transfer model used in the offline CFRAM calculations is different than the radiative transfer model used in the climate simulations with WACCM.*

General comment: Discuss a bit more in which altitude/latitude region the impact is highest/lowest and give numbers.

The current Section 5 has been rewritten and now includes more quantification, as shown below (the same is done for the Conclusion and Discussion section).

[revised manuscript text omitted]

*This was stated in the text as follows: The picture in the mesosphere is similar. The main difference is that the temperature changes are larger, note the difference of the range for the temperature change between Fig. 10c, d and Fig. 10e,f.*

As this was not clear enough, it has been rewritten as follows:

*The picture in the mesosphere is similar as in the upper stratosphere. The main difference is that the temperature changes are larger. The global temperature change due to direct forcing of $CO_2$ is about 15 K, which is stronger than in the upper stratosphere. The $O_3$-feedback results in a partial temperature changes of about 3 K in the mesosphere. Just like in the upper stratosphere water vapour, cloud and albedo feedback play no role.*

Further, you use different y-axis scale which masks a bit the differences between the atmospheric regions. At least you should mention the different y-scales in the figure caption.

This was done to be able to make out the relative importance of the feedbacks in the different regions. We have now made note of this point in both the text and the caption.

*Note that the range of values on the y-axis is not the same for the different subplots.*

Figure 3: Here, a strong cooling due to CO2 is visible while all others rather show a warming. This is not discussed like this in the text. At least it is not clearly stated.

This has been added to the text and is also mentioned in section 4.1 *We see that increasing $CO_2$ leads to a cooling almost everywhere in the middle atmosphere, except at the high latitudes in the cold summer upper mesosphere, where we see a warming instead.*

P17, L606, Figure 4: Why has the upper stratosphere picked? Changes seem to be highest in the mesosphere. Why has the separation not been done there?

The only reason that the separation has not been done here is that one would end up with a figure consisting of 3 pages. Or three separate big figures. This can be added, if needed, but we think it will not add much information.

The aim of this figure is to show the temperature response of the $CO_2$ and the SST separately. The upper stratosphere is a bit more interesting then showing the mesosphere as there is still some albeit small effect from the albedo and water vapour, while in the mesosphere this is not the case. The text about this has been edited:

*Figure 13 shows temperature responses in the upper stratosphere for the experiment with double $CO_2$ (a,b) and changed SSTs (c,d) separately. This has been done to give insight in the temperature response of the $CO_2$ and the SST separately. These temperature changes were calculated in the same way as for Fig. 11. Again also, the 'total'-column shows the temperature changes as simulated by WACCM, the columns $CO_2$, $O_3$, $H_2O$, cloud, albedo, dynamics, Non-LTE shows the temperature responses as calculated by CFRAM. Error shows the difference between temperature change in WACCM and the sum of the calculated temperature responses in CFRAM.*

*We learn from this figure that the effects of the changed SSTs on the upper stratosphere are relatively small as compared to the effects of changing the $CO_2$. We show the temperature changes for the upper stratosphere as an example. In the lower stratosphere and the mesosphere, we see the same pattern: the effect of the $CO_2$ on the temperature is generally much larger than the effect of the SSTs on the temperature. This finding is consistent with the study of Fomichev et al. (2007), where it is concluded that the impact of changes in SSTs on the middle atmosphere is relatively small and localized as compared to the combined response of changing the $CO_2$-concentration and the SSTs.*

P17, L621: Thus, SSTs are important for the water vapour feedback on temperature, but lower for the CO2 feedback on temperature. This could also be more clearly stated.

Here, we discuss the calculated temperature responses for the experiment with double $CO_2$ (C2) and changed SSTs separately (S1). We make this now clearer in the text.

*We learn from this figure that the effects of the changed SSTs on the upper stratosphere are relatively small as compared to the effects of changing the $CO_2$. In the lower stratosphere and the mesosphere, we see the same pattern: the effect of the $CO_2$ on the temperature is generally much larger than the effect of the SSTs on the temperature.*

*The changes in SSTs are, however, responsible for large temperature changes as a result of the dynamical feedbacks, especially in the winter hemispheres. A similar figure for the lower stratosphere (not shown) shows that the temperature response to the water vapour feedback is almost solely due to changes in the SSTs.*

P17, L631: Temperature direct response -> Direct temperature response

This has been changed.

P18, L669: Add also the paper by Brewer, 1949, QJRMS
This reference has been added.

P19, L681ff: Most of this paragraph rather belongs to the introduction.

Part of what was written here has now been moved to the introduction.

*The circulation in the middle atmosphere is driven by waves. Wave forcing drives the temperatures in the middle atmosphere far away from radiative equilibrium. In the mesosphere, there is a zonal forcing, which yields a summer to winter transport. In the polar winter stratosphere, there is a strong forcing that consists of rising motion in the tropics, poleward flow in the stratosphere and sinking motion in the middle and high latitudes. This circulation is referred to as the 'Brewer-Dobson circulation' (Brewer, 1949; Dobson, 1956).*

*Dynamical effects make important contributions to the middle-atmosphere energy budget, both through eddy heat flux divergence and through adiabatic heating due to vertical motions. It is therefore important that we also consider changes to the middle-atmosphere climate due to dynamics. We refer to this as the 'dynamical feedback' (Zhu et al., 2016).*

This is to explain why we consider the dynamical feedback and what we mean by that in this study. As for the other feedbacks, I go into the background of this feedback in each of the different sections of section 4.

P21, L757ff: Please quantify your results and give the percentages.

The percentages are given now:

*We find, as expected, that an increase in $CO_2$, leads to an increase of ozone in most of the middle atmosphere. The increase of $O_3$ is about 20% around 2 hPa in the tropical region for experiment S2 with respect to C1. This corresponds with what is seen by Fomichev et al., (2007), however they find that the increase in ozone in January is a bit lower in this region (around 15%, see their Figure 7).*

*There are some regions where the $O_3$-concentration is decreasing. In the tropical lower stratosphere, a decrease of about 20% is seen, in the summer polar mesosphere (around 0.01 hPa) ozone decreases by 3%, while in the mesosphere (around 0.02 hPa), ozone decreases by over 30%. Fig. 3c and f show that changing the SSTs also has a significant impact on the ozone concentration. A complete account of the ozone changes is out of the scope of this paper.*

P21, L771: What symbol has been used for the statistical significance? Please add that to the figure caption.

This has been added: *The dotted regions indicate the regions where the data reaches a confidence level of 95%.*

P21, L780: Before you always wrote "ozone" but now you write "$O_3$" without introducing the abbreviation.

The chemical notation $O_3$ has now been introduced in the introduction of the paper, so that is clear that it signifies ozone.

P21, L787-789: Ozone is destroyed in polar winter in the lower stratosphere, not at 0.1 hPa.

We would like to refer to Schmidt et al., 2006. They don't use the WACCM model, but HAMMONIA, however this is coupled to the same chemical model MOZART3. In their study, they perform a similar experiment as we do in our study, however they start from 360 ppm of CO2 and double to 720, while we start from the pre-industrial level of CO2 (280 ppm). Our results are very similar.

They write *"The large decrease in the atomic oxygen mixing ratio at high summer latitudes above 0.01 hPa results from increased upwelling and leads also to an ozone decrease at this level. The ozone decrease in the polar winter around 0.1 hPa (approx. 65 km) is mainly caused by the increase of NO and (to a lesser extent) Cl mixing ratios due to stronger subsidence of NO and Cl-rich air"*

P22, L804 and several other occasions: the 2 in CO2 should be written as subscript. All instances of $CO_2$ are now written with a subscript in the paper.

General comment: Discuss also the statistical significance. Which results do you derive with which significance?

Comments about the statistical significance have been added in the figures that show the difference in a field between a perturbation run and the control run (such as Fig. 1). For the temperature responses we cannot calculate the statistical significance, but we can calculate the error, which we show in Fig. 10 for example.

Figure 10 caption: Add information what symbol has been used for the statistical significance.

This has been added: *The dotted regions indicate the regions where the data reaches a confidence level of 95%.*

P23, L853: Sentence with "This has been explained…….." is too complicated and should be rephrased,e.g. This can be explained by a ……leading to an …...

This sentence has been rewritten and split into several sentences:

*This can be understood as increasing the water vapour in the middle atmosphere leads to an increase in longwave emissions in the mid and far-infrared by water vapour. This in turns leads ot a cooling of the region. Similarly, a decrease in water vapour leads to a warming of the region (Brasseur and Solomon, 2005).*

P24, L888: Discuss Figure 12 a bit more. The figures should be first described. What is shown, what do you see…

This part has been rewritten:

*Forcing the model with SSTs from the double $CO_2$-climate (as in experiment S1 and S2) yields an overall increase in the cloud cover in the upper troposphere, while this is not the case if one only increases the $CO_2$ concentration (as in experiment C2). Figure 10 shows the temperature responses to changes in cloud (left) and albedo (right) in July (top) and January (bottom) for experiment S1, as calculated by CFRAM.*

*Fig. 10 shows in the tropical region, there is a warming due to changes in clouds, while there is a cooling at higher latitudes in July (see Figure 10a). In January, the pattern looks slightly different (see Figure 10c). These temperature changes are due to changes in the balance between the increased reflected shortwave radiation and the decrease of outgoing longwave radiation.*

P24, L893: Which cooling are you talking about? Up to date cooling for future cooling?

We were talking about the cooling as calculated by CFRAM. This paragraph has now been rewritten.

*We also see an effect of the changes in surface albedo in the stratosphere (see Figure 10 b and d). The cooling in the summer polar stratosphere shown in Figure 10 b and d is due to radiative changes. We suggest that this cooling is due to a decrease in surface albedo, which would lead to less shortwave radiation being reflected. However, more research is needed.*

P26, L924: Add a number. How much stronger?

This paragraph has been rewritten: *We find that the temperature change due to the direct forcing of $CO_2$ increases with increasing height in the middle atmosphere. The temperature change in the lower stratosphere due to the direct forcing of $CO_2$ is around 3 K. In the upper stratosphere, the cooling due to the direct forcing of $CO_2$ is about 9 K, which is considerably stronger than in the lower stratosphere. In the mesosphere, the cooling due to the direct forcing of $CO_2$ is about 15 K.*

General comment: How do the results agree with previous studies? What is the importance of you results for future predictions or climate change etc.? This should also be discussed.

The discussion and conclusion section has been rewritten to make these aspects clearer.

**5.0 Discussion and conclusions**

*In this study, we have applied the climate feedback response analysis method to climate sensitivity experiments performed by WACCM. We have examined the middle atmosphere response to $CO_2$ doubling with respect to the pre-industrial state. We also investigated the combined effect of doubling $CO_2$ and subsequent warming SSTs, as well as the effects of separately changing the $CO_2$ and the SSTs. It is important to note that no other method before has been able to quantify how much of the temperature change in the middle atmosphere is due to the different feedback processes before.*

*It was seen before that the sum of the two separate temperature changes in the experiment with only changed $CO_2$ and only changed SSTs is, at first approximation, equal to the changes observed in the combined simulation (see e.g. Fomichev et al. (2007) and Schmidt et al. (2006)). This is also the case for WACCM.*

*We have found that, even though changing the SSTs yields significant temperature changes over a large part of the middle atmosphere, the effects of the changed SSTs on the middle atmosphere are relatively small as compared to the effects of changing the $CO_2$ without changes in the SSTs.*

*We have given an overview of the mean temperature responses to the changes in $CO_2$ and various feedback processes in the lower stratosphere, upper stratosphere and in the mesosphere in January and July. We find that the temperature change due to the direct forcing of $CO_2$ increases with increasing height in the middle atmosphere. The temperature change in the lower stratosphere due to the direct forcing of $CO_2$ is around 3 K. There is a stronger cooling in the tropical lower stratosphere of about 4 K in July and 3.5 K in January.*

*In the upper stratosphere, the cooling due to the direct forcing of $CO_2$ is about 9 K, which is considerably stronger than in the lower stratosphere. The cooling is stronger in the summer polar regions, where the cooling reaches a value of 11K, than in the winter polar region, where the cooling is only about 8K. In the mesosphere, the cooling due to the direct forcing of $CO_2$ is even stronger: 15 K.*

*The ozone concentration changes due to changes in the $CO_2$-concentration as well as by changes in the SSTs. The temperature changes caused by this change in ozone concentration generally mitigate the cooling caused by the direct forcing of $CO_2$. However, in the tropical lower stratosphere and in some regions of the mesosphere, the ozone feedback cools these regions further. In the tropical lower stratosphere, for example, there is a cooling of 1K due to the ozone feedback.*

*We also have seen that the global mean temperature response due to dynamical feedbacks is small in the global average in all regions: less than 1 K. However, local responses to the changes in dynamics can be large. Doubling the $CO_2$ leads to a stronger summer-to-winter-pole flow, which leads to a cooling of the summer mesosphere and a warming of the winter mesosphere. Changing the SSTs weakens this effect in the mesosphere, but affects the temperature response in the stratosphere and lower mesosphere.*

*Using CFRAM on WACCM data shows that the change in water vapour leads to a cooling of up to 2 K in the lower stratosphere. It should be noted that climate models currently have a limited representation of the processes determining the distribution and variability of lower stratospheric water vapour. This means that the temperature response to the water vapour feedback might be different using a different model. We have also seen a small effect of the cloud and albedo feedback on the temperature response in the lower stratosphere, while these feedbacks play no role in the upper stratosphere and the mesosphere.*

*The results seen in this study are consistent with earlier findings. As in Shepherd et al., (2008), we find that the higher the temperature at a region in the atmosphere, the more cooling there is seen due to the direct feedback of $CO_2$. We find, as in Zhu et al., (2016) that the temperature responses due to the direct forcing of $CO_2$ follow the temperature distribution quite closely, while the temperature responses due to $O_3$ follow the changes ozone concentration instead.*

*We have also seen that the ozone feedback generally yields a radiative feedback that mitigates the cooling, which is due to the direct forcing of $CO_2$, which is consistent with earlier studies such as Jonsson et al., (2004), Dietmüller et al., (2014). CFRAM is the first study that allows for calculating how much of the temperature response is due to which feedback process.*

*The next step would be to investigate the exact mechanisms behind the feedback processes in more detail. Some processes can influence the different feedback processes, such as ozone depleting chemicals influencing the ozone concentration and thereby the temperature response of this feedback. A better understanding of the effect of the increased $CO_2$-concentration on the middle atmosphere, will help to distinguish the effects of the changes $CO_2$- and $O_3$-concentration.*

*There is also a need for a better understanding of how different feedbacks in the middle atmosphere affect the surface climate. As discussed in the introduction, the exact importance of ozone feedback on the global mean temperature is currently not clear (Nowack et al., 2015, Marsh et al., 2016). A similar analysis as in this paper can be performed to quantify the effects of feedbacks on the surface climate.*

*In conclusion, we have seen that CFRAM is an efficient method to quantify climate feedbacks in the middle atmosphere, although there is a relatively large error due to the linearization in the model. The CFRAM allows for separating and estimating the temperature responses due an external forcing and various climate feedbacks, such as ozone, water vapour, cloud, albedo and dynamical feedbacks. More research into the exact mechanisms of these feedbacks could help us to understand the temperature response of the middle atmosphere and their effects on the surface and tropospheric climate better.*

To summarize you results and for having it easier with the discussion you could make a table/matrix where you mark which feedbacks are important in which altitude/latitude region.

*We think that such a table would basically be a copy of Fig. 10. Instead of adding a new table, we now refer to Figure 10.*

Technical corrections:

P2, 75: Add before after "in this way". Changed
P4, L164: are -> were Changed
P4, L169: by -> with Changed
P5, L207: earth -> Earth Changed
P5, L218: SSTs forcing -> SST forcings Changed
P7, L280: delete "to" before balance Changed
P12, L505: don't -> do not Changed
P15, L577: leads to -> leads to a Changed
P17, L619: One "the" obsolete. Changed
P18, L681: synoptic -> synoptic scale Changed
P21, L771: signifance -> significance Changed
P21, determinged -> determined Changed
P21, L776: CLOx -> ClOx Changed
P23, L824: to lead a -> to lead to a Changed
P23, L828: Delete "in WACCM", you have already written it at the beginning of the sentence. Changed
P23, L838: aren't -> are not Changed
P23, L841: "neither" obsolete? Or should it rather read "either". Changed
P23, L848: found by -> simulated by or simulated with Changed
P25, L911: at a first -> as a first: This has been rephrased.
P26, L939: leads to -> leads to a Changed

**Responses to Reviewer 2**

General comments

The paper is now much clearer but there are still minor revisions necessary, especially concerning atmospheric chemistry.

*We thank reviewer 2 for his/her comments.*

Specific comments

Section 2.2, line 214ff: It is still not clear if the radiatively and chemically active gases CH$_4$ and N$_2$O (sometimes called well-mixed greenhouse gases) are kept at preindustrial levels in all scenarios at the surface (include in Table 1). It should be also mentioned that there are no CFCs in the atmosphere (I hope this is the case, otherwise major revision necessary).

Are the boundary conditions for all anthropogenic emissions kept constant in all scenarios?

Section 2.2 has been rewritten completely. We explain that in this study, the F_1850 compset (component set) of the model is used, i.e. the model assumes pre-industrial (PI) conditions. This compset simulates an equilibrium state, which means that it runs a perpetual year 1850. Four experiments have been performed (see Table 1).

The compset used in all experiments is still F_1850, which means that other radiatively and chemically active gases, such as ozone, will change only because of the changes in the CO$_2$-concentration, due to WACCM's interactive chemistry. CFCs don't play a role in our experiments.

The remark on inconsistencies in SST in the reply should be included in the text.

It has now been added that the fact that CESM doesn't include atmospheric chemistry, is not consideration in our study.

Line 305ff: It is not a good idea to use the same letters for different physical quantities (power flux and heating rate).

We use W/m^2 as the units of heating rates (per unit of area) for the air layer between two adjacent vertical levels, which is equivalent to heating rate per unit volume, instead of K/day, which is heating rate per unit mass. Because the radiative heating rates are the convergence of radiative energy fluxes entering/leaving the layer, it is rather natural and straightforward (meaning "without extra steps") to have the same units of heating rates (convergence) as the radiative energy fluxes. Throughout, the symbol S_vector (denote S with the vector on the top) represents of the convergence of short wave radiative fluxes entering each layer (thereby positive for heating ), whereas R_vector represents of the divergence of long wave radiative fluxes leaving each layer (thereby positive for cooling). We use

vector to represent the heating rates or a thermodynamical variable (e.g., T) in the vertical layers for mathematical convenience such as the DR/DT would be a matrix.  In other words, the vector has no sense of direction in the physical world as it does not correspond to the upward or downward radiative fluxes.

In the manuscript, we have make it clear the S_vector is the vertical profile of the convergence of  shortwave radiative energy fluxes, corresponding to the heating rates due to absorption of shortwaves in units of W/m^2, whereas R_vector is the vertical profile of the divergence of longwave radiative energy fluxes, corresponding to the cooling rates due to net emission of longwave radiation in units of W/m^2.

Also the constant 'g' is not defined (gravitational acceleration?).

g is indeed the gravitational acceleration, this has been added now.

Line 669: Cite the original references also here, they are in the reference list.

These references have been added. (This part has been moved to the introduction).

*The circulation in the middle atmosphere is driven by waves. Wave forcing drives the temperatures in the middle atmosphere far away from radiative equilibrium. In the mesosphere, there is a zonal forcing, which yields a summer to winter transport. In the polar winter stratosphere, there is a strong forcing that consists of rising motion in the tropics, poleward flow in the stratosphere and sinking motion in the middle and high latitudes. This circulation is referred to as the 'Brewer-Dobson circulation' (Brewer, 1949; Dobson, 1956).*

*Dynamical effects make important contributions to the middle-atmosphere energy budget, both through eddy heat flux divergence and through adiabatic heating due to vertical motions. It is therefore important that we also consider  changes to the middle-atmosphere climate due to dynamics. We refer to this as the 'dynamical feedback' (Zhu et al., 2016).*

Line 708: Only non-orographic gravity waves are modulated by convection. Please rearrange paragraph.

This paragraph has been rewritten:

*The warmer sea surface temperatures affect the dynamics in the middle atmosphere. It has for example been shown that higher SSTs in the tropics leads to an amplification in deep convection, which enhances the generation of quasi-stationary waves (Deckert and Dameris, 2008). Enhanced SSTs lead to an enhanced dissipation of planetary waves, as well as an enhanced dissipation of orographic and non-orographic waves in the upper stratosphere (Oberländer et al., 2013).*

Line 745ff: please rewrite: HOx dominates ozone destruction in the mesosphere and lower stratosphere, NOx and Clx in the middle and upper stratosphere.

This part has been rewritten:

*Ozone plays a crucial role in the chemical and radiative budget of the middle atmosphere. The distribution of ozone in the middle atmosphere is determined by both chemical and dynamical processes. Most of the ozone production takes place in the tropical stratosphere, as a result of photochemical processes, which involve oxygen. Meriodional circulation then transports ozone to higher latitudes (Langematz, 2019). The production of ozone is largely balanced by catalytic destruction cycles involving $NO_x$, $HO_x$ and $Cl_x$ radicals. $HO_x$ dominates ozone destruction in the mesosphere and lower stratosphere, while $NO_x$ and $Cl_x$ dominate this process in the middle and upper stratosphere (Cariolle, 1983).*

Insert in line 754 "due to the strong temperature dependence of the Chapman reaction O+O$_3$".

This has been added:

*In this study, we are interested in the temperature response to changes in ozone concentration induced by the increased $CO_2$ concentration and/or the changes in SST in WACCM. Under enhanced $CO_2$ concentrations, the ratio between $O_3$ and $O$ mixing ratios is generally shifted toward a higher concentration of ozone, which is caused by the strong temperature dependency of the ozone production reaction ($O + O_2 + M \rightarrow O_3 + M$).*

In the preindustrial stratosphere are no
aircraft NOx-emissions and no CFCs (line 755). Cl should be only from CH$_3$Cl (forest fires etc.) This is misleading here.

This sentence has been removed.

Line 773ff: Please rewrite and include something like the following: If O/O$_3$ is shifted to lower values due to cooling the catalytic
ozone destruction cycles are slower since the reactions of the radicals (NO$_2$, OH, HO$_2$, ClO) with O are the rate limiting steps.
This has been rewritten (See below)

Line 783: Why? This is secondary if the scenarios are consistent. There are a lot of textbooks and review papers on stratospheric and mesospheric chemistry.

This sentence has been removed.

Thanks, we have consulted them and we refer to Schmidt et al. as they did a similar study as we did.

Line 788: This is very special and only relevant in case of perturbations by solar particles at solar maximum conditions. Chlorine should be low (in total < 0.6ppbv).

Indeed the role if Cl should not be large, we have added that now. We would like to refer to Schmidt et al., 2006. They don't use the WACCM model, but HAMMONIA, however this is coupled to the same chemical model MOZART3. In their study, they perform a similar experiment as we do in our study, however they start from 360 ppm of CO2 and double to 720, while we start from the pre-industrial level of CO2 (280 ppm). Our results are very similar. Schmidt et al. (2006) write that "The ozone decrease in the polar winter around 0.1 hPa (about 65 km) is mainly caused by the increase of NO and (to a lesser extent) Cl mixing ratios due to stronger subsidence of NO and Cl-rich air." We see a similar increase of Cl as they do.

The section about the ozone feedback has been rewritten completely. We hope that is now clear what is done.

[revised manuscript text omitted]

Line 928: This sentence is a mess. Focus on temperature dependence of reaction rates.

This sentence has been rewritten. The aim of this paper is not the explain the changes in ozone concentration. As we only would like to calculate the temperature effect of these, so we have left this part out.

*The ozone concentration changes due to changes in the $CO_2$-concentration as well as by changes in the SSTs. The temperature changes caused by this change in ozone concentration generally mitigate the cooling caused by the direct forcing of $CO_2$. However, we have also seen in that in the tropical lower stratosphere and in some regions of the mesosphere, the ozone feedback cools these regions further.*

Line 944ff: This is uncertain, please include remark or leave out here.

A remark has been included.

*Using CFRAM on WACCM data shows that the temperature change in the lower stratosphere is influenced by the water vapour feedback and to a lesser degree by the cloud and albedo feedback, while these feedbacks play no role in the upper stratosphere and the mesosphere. It should be noted that climate models currently have a limited representation of the processes determining the distribution and variability of lower stratospheric water vapour. This means that the temperature response to the water vapour feedback might be different using a different model.*

**Technical corrections**

Table 1: use subscripts.

Subscripts added.

Line 710: Typo

Corrected.

Line 736: Subscripts

Subscripts added.

line 749: Typo, also include 'e.g.' in citation.

Corrected and e.g. added.

*Ozone plays a crucial role in the chemical and radiative budget of the middle atmosphere. The distribution of ozone in the middle atmosphere is determined by both chemical and dynamical processes. Most of the ozone production takes place in the tropical stratosphere, as a result of photochemical processes, which involve oxygen. Meridional circulation then transports ozone to other parts of the middle atmosphere (Langematz, 2019). The production of ozone is largely balanced by catalytic destruction cycles involving $NO_x$, $HO_x$ and $Cl_x$ radicals. $HO_x$ dominates ozone destruction in the mesosphere and lower stratosphere, while $NO_x$ and $Cl_x$ dominate this process in the middle and upper stratosphere (e.g. Cariolle, 1983).*

References: Several entries are incomplete, e.g. page numbers and/or DOI missing.

DOI has been added for all entries. Page number added for those which had page numbers specified.

Line 1121: incomplete journal name.

This reference is no longer there.

**Response to reviewer 3:**

I am sorry to say that I still have reservations about the paper in
its present form. I am afraid that there are some rather fundamental
disagreements between the authors and the reviewer so that my comments
are rather general.

*We thank reviewer 3 for his/her comments.*

My major point is that the paper is still missing a clear message,
going beyond what is already known.

This is the first study that allows for calculating how much of the temperature
response in the middle atmosphere is due to which feedback process. We have now
also made clearer in the introduction what we mean with feedbacks in this paper.

Yes, in a doubled CO2 climate, the direct radiative forcing of CO2 will
lead to a cooling in the upper stratosphere, but this is certainly
well known (e.g. WMO 2018).

What is new in this paper, is that we calculate exactly how much of the temperature
change at a certain point of the middle atmosphere is due to which processes. We
can calculate the temperature response due to the direct forcing of CO2, as well as
the temperature responses due to the changes in ozone concentration, water
vapour concentration, clouds, albedo and dynamics. This is the first paper that
applies the CFRAM method to the middle atmosphere in this way and thus first
study that allows for a calculation the partial temperature responses.

The cooling in the upper stratosphere, by affecting upper
stratospheric ozone chemistry (but which ozone chemistry exactly, see
below), leads to an increase in ozone (referred to in the abstract as
"ozone feedback", which leads to an increase in short-wave heating
(i.e. yields a radiative feedback that generally mitigates this
cooling) -- I agree, but again, what is new here?

What is new in this paper, is that we calculate exactly how much of the temperature
change at a certain point of the middle atmosphere is due the changes in ozone
concentration. More traditional methods of studying the ozone feedback in the
middle atmosphere don't allow for calculating the temperature response to the
changes in ozone concentration.

The abstract states that the "temperature response due to dynamical
feedbacks is small in global average", but I find it difficult to
understand which dynamical feedbacks are meant here -- perhaps
enhanced tropical upwelling in a 2xCO2 run?

This has now been made clear in the introduction. CFRAM allow to calculate the
radiative contribution to the temperature response for different feedback processes.
However, changes in the dynamics also influence the temperature response and

are calculate separately (see also equation A4 in the appendix and *Zhu et al.*, 2016 for comparison).

This has been added to the introduction:
*In this study, we investigate the effects of doubling the $CO_2$-concentration and the accompanying sea surface temperature change on the temperature in the middle atmosphere as compared to the pre-industrial state. We use CFRAM to calculate the radiative contribution to the temperature change due to changes in carbon dioxide directly as well as due to changes in ozone, water vapour, albedo, and clouds. We refer to the changes in ozone, water vapour, albedo and clouds in response to changes in the $CO_2$-concentration as the ozone, water vapour, albedo and cloud feedbacks.*

*The circulation in the middle atmosphere is driven by waves. Wave forcing drives the temperatures in the middle atmosphere far away from radiative equilibrium. In the mesosphere, there is a zonal forcing, which yields a summer to winter transport. In the polar winter stratosphere, there is a strong forcing that consists of rising motion in the tropics, poleward flow in the stratosphere and sinking motion in the middle and high latitudes. This circulation is referred to as the 'Brewer-Dobson circulation' (Brewer, 1949; Dobson, 1956).*

*Dynamical effects make important contributions to the middle-atmosphere energy budget, both through eddy heat flux divergence and through adiabatic heating due to vertical motions. It is therefore important that we also consider changes to the middle-atmosphere climate due to dynamics. We refer to this as the 'dynamical feedback' (Zhu et al., 2016).*

*The main goal of this paper is to calculate the contribution to the temperature change due to changes in carbon dioxide directly as well as due to changes in ozone, water vapour, albedo, clouds and dynamics in the middle atmosphere under a double $CO_2$-scenario using CFRAM. Our intention is not to give a complete account of the exact mechanisms behind the changes in ozone, water vapour, albedo, clouds and dynamics.*

Further, it is stated that the "temperature change in the lower stratosphere is influenced by the water vapour feedback", but again, the processes in question here remain unclear. I suggest to state at least what the sign of the temperature change is (increase or decrease?) and what "water vapour feedback" means (increase or decrease of water vapour? chemical impact of water vapour on ozone or radiative effects of water vapour?).

We have quantified this now and added the caveat that this might be model dependent.

*Using CFRAM on WACCM data shows that the change in water vapour leads to a cooling of up to 2 K in the lower stratosphere. It should be noted that climate models currently have a limited representation of the processes determining the distribution*

*and variability of lower stratospheric water vapour. This means that the temperature response to the water vapour feedback might be different using a different model.*

In the introduction we have now also made clear what we mean with feedbacks:

*In this study, we investigate the effects of doubling the $CO_2$-concentration and the accompanying sea surface temperature change on the temperature in the middle atmosphere as compared to the pre-industrial state. We use CFRAM to calculate the radiative contribution to the temperature change due to changes in carbon dioxide directly as well as due to changes in ozone, water vapour, albedo, and clouds. We refer to the changes in ozone, water vapour, albedo and clouds in response to changes in the $CO_2$-concentration as the ozone, water vapour, albedo and cloud feedbacks.*

*The circulation in the middle atmosphere is driven by waves. Wave forcing drives the temperatures in the middle atmosphere far away from radiative equilibrium. In the mesosphere, there is a zonal forcing, which yields a summer to winter transport. In the polar winter stratosphere, there is a strong forcing that consists of rising motion in the tropics, poleward flow in the stratosphere and sinking motion in the middle and high latitudes. This circulation is referred to as the 'Brewer-Dobson circulation' (Brewer, 1949; Dobson, 1956).*

*Dynamical effects make important contributions to the middle-atmosphere energy budget, both through eddy heat flux divergence and through adiabatic heating due to vertical motions. It is therefore important that we also consider changes to the middle-atmosphere climate due to dynamics. We refer to this as the 'dynamical feedback' (Zhu et al., 2016).*

What remains is an evaluation of CFRAM as a tool to study climate change. There could be new developments here, but if the focus were on CFRAM than the nature of the paper would be much more methodological that it is in its present form.

CFRAM is the first study that allows for calculating how much of the temperature response is due to which feedback process. The main goal of this paper is to calculate the contribution to the temperature change due to changes in carbon dioxide directly as well as due to changes in ozone, water vapour, albedo, clouds and dynamics in the middle atmosphere under a double $CO_2$-scenario using CFRAM. Our intention is not to give a complete account of the exact mechanisms behind the changes in ozone, water vapour, albedo, clouds and dynamics. We have given a detailed account of the method, but the focus is on the results.

I see also remaining disagreements between the authors and the reviewer judging from the reply to my comments.

The authors state "We are not speaking here about the changes in O3-concentration due to the ozone hole, but rather changes in ozone concentration that are resulting from changes in the CO2 concentration." I did not talk about the "ozone hole" either, what I am talking about

is the upper stratospheric ozone loss, which is also chlorine
driven. And enhanced levels of stratospheric chlorine are around for
many decades to come. From reading the manuscript, I assume that Cl
was set to zero, or perhaps 0.6 ppb -- but I am not sure (see also
below). I think this issue could be at least briefly addressed (if
chlorine is not relevant for the present study it should be stated in
the paper). Other chemical effects could be due to changing N2O
levels. I think it is not a good idea to remove the reference to the
WMO ozone assessment (WMO 2018) entirely.

In our experiment, the levels of Cl and other fields are set to pre-industrial levels (as
in the F_1850 compset of WACCM) in WACCM.  However, the increase in $CO_2$-
concentration brings about different concentrations of reactive consituents affecting
the ozone concentration. The changes in the above-mentioned constituents are
simulated using the interactive chemistry as in the MOZART3 model (*Kinnison et
al.,* 2007).

In the paper we have now given an overview of the percentage change in the zonal
and monthly mean concentration of Cl (a), NO (b), O (c), OH (d), $CH_4$ (e) and $NO_x$
(f) and $N_2O$ (g) in July due to combined effect of the $CO_2$ increase and SSTs
changes (experiment (C2 vs B1), as simulated by WACCM.

The reference to the WMO ozone assessment has been added.

I stated in my previous review that "it should be clearly said that
the middle atmosphere is *not* in radiative equilibrium" -- in
response the authors changed the discussion, which clearly improved
the presentation of this aspect. However I find the sentence "In the
absence of eddy motions the zonal-mean temperature would relax to a
radiatively determined state" still misleading. This sounds a bit as
this were still a possible state of the atmosphere; note that
without "eddy motions", i.e. without atmospheric waves, in a radiative
equilibrium there are no heating or cooling terms, i.e. no transport
across isentropic surfaces, in other words no Brewer-Dobson
circulation. (And I think that throughout most of the troposphere,
outside of convection, the troposphere is not dynamically unstable).

*This sentence has now been removed.*

A few details:

I 214: WACCMs chemical model is associated here with CO2 changes?
What is the CO2 chemistry in WACCM? Or is it transport and changing
CO2 emissions that are relevant here?

A more detailed description of the experimental setup has now been added (see
below).

We would also like to refer to Schmidt et al., 2006. They don't use the WACCM model, but HAMMONIA, however this is coupled to the same chemical model MOZART3. In their study, they perform a similar experiment as we do in our study, however they start from 360 ppm of CO2 and double to 720, while we start from the pre-industrial level of CO2 (280 ppm). Our results are very similar.

*In this study, the F_1850 compset (component set) of the model is used, i.e. the model assumes pre-industrial (PI) conditions. This compset simulates an equilibrium state, which means that it runs a perpetual year 1850. Four experiments have been performed for this study (see Table 1).*

*Experiment C1 is the control run, with the pre-industrial $CO_2$ concentration (280 ppm) and forced with pre-industrial ocean surface conditions such as sea surface temperature and sea ice (referred to SSTs from now on). These SSTs are generated from the CMIP5 pre-industrial control simulation by the fully coupled Earth system model CESM. The atmospheric component of CESM is the same as WACCM, but does not include stratospheric chemistry (Hurrell et al., 2013). This latter aspect is not considered in this study.*

*Experiment C2 represents the experiment with the $CO_2$ concentration doubled as compared to the pre-industrial state (560 ppm) and forced with the same pre-industrial SSTs as in experiment C1. In WACCM, the $CO_2$-concentration does not double everywhere in the atmosphere. Only the surface level $CO_2$ mixing ratio is doubled, and elsewhere in the atmosphere is calculated according to WACCM's chemical model.*

*The compset used in this experiment and all the following ones is still F_1850, which means that other radiatively and chemically active gases, such as ozone, will change only because of the changes in the $CO_2$-concentration, due to WACCM's interactive chemistry. Chlorofluorocarbons (CFCs), which have a major impact on the ozone concentration in the real atmosphere, don't play a role in our experiments.*

*In experiment S1, we simulate the scenario, in which there is the SSTs forcing from the coupled CESM for double $CO_2$ condition, but the pre-industrial $CO_2$-concentration of 280 ppm. S2 represents the experiment with the $CO_2$-concentration in the atmosphere doubled to 560 ppm and the SSTs prescribed for the double $CO_2$-climate. Experiment C1, C2, S1 and S2 will be also referred to hereafter by PI, the simulation with high CO2, the simulation with high SSTs and the simulation with high $CO_2$ and SSTs, respectively.*

*The experimental setup of this study is similar to the setup performed with the Canadian Middle Atmosphere Model (CMAM) by Fomichev et al. (2007) and with the Hamburg Model of the Neutral and Ionized Atmosphere (HAMMONIA) by Schmidt et al. (2006). The HAMMONIA model is coupled to the same chemical model as WACCM: MOZART3. The setup in their study is similar, however, in their study, they double the $CO_2$-concentration from 360 ppm to 720 ppm, while in our study, we double from the pre-industrial level of $CO_2$ (280 ppm).*

l. 224: I think you need more documentation here on the WACCM run. I
see pre-industrial CO2 and doubled CO2 (also SSTs are
mentioned), but a lot of other fields are unclear;
e.g. stratospheric chlorine, N2O, CH4. Aerosol loading? Are
these compounds (and the entire setup) based on
"pre-industrial"? Could tropospheric ozone be relevant
(different between the runs). Perhaps you could use a
citation, where all the assumptions of the WACCM run are
described? Note that when you use pre-industrial with (just)
CO2 doubled, these doubled CO2 runs do not describe a future

A more detailed description of the experimental setup has now been added (the
response the to the previous question).

It is correct that the doubles CO2 run don't describe a realistic future scenario.
However, these experiments can still teach us something. We write this in the paper
as well (in the introduction);

We acknowledge that such idealized equilibrium simulation cannot reproduce the
complexity of the atmosphere, in which the $CO_2$-concentration is changing
gradually. However, simulating a double $CO_2$-scenario still allows us to identify
robust feedback processes in the middle atmosphere.
We mention this now also in section 2.2:

*Note that experiment S1 and C2 are not representing scenarios that could happen*
*in the real atmosphere. These experiments have been used to study the effect of*
*the SSTs separately. Experiment S2 doesn't take into account other*
*(anthropogenic) changes in the atmosphere not caused by changes in the $CO_2$-*
*concentration and the SSTs.*

We would also like to refer to *Schmidt et al.,* 2006. They don't use the WACCM
model, but HAMMONIA, however this is coupled to the same chemical model:
MOZART3.  In their study, they perform a similar experiment as we do in our study,
however they start from 360 ppm of CO2 and double to 720, while we start from the
pre-industrial level of CO2 (280 ppm). Our results are very similar.

In our experiment, the levels of Cl and other fields are set to pre-industrial levels (as
in the F_1850 compset of WACCM) in WACCM.  However, the increase in $CO_2$-
concentration brings about different concentrations of reactive consituents affecting
the ozone concentration. The changes in the above-mentioned constituents are
simulated using the interactive chemistry as in the MOZART3 model (*Kinnison et*
*al.,* 2007).

In the paper we have now given an overview of the percentage change in the zonal
and monthly mean concentration of Cl (a), NO (b), O (c), OH (d), $CH_3$ (e) and $NO_x$
(f) and $N_2O$ (g) in July due to combined effect of the $CO_2$ increase and SSTs
changes (experiment (C2 vs B1), as simulated by WACCM.

l 745-748: You state "Ozone plays a major role in the chemical and
radiative budget of the middle atmosphere. The ozone

distribution in the mesosphere is maintained by a balance between transport processes and various catalytic cycles involving nitrogen oxides, HOx and Clx radicals." Which transport processes with an impact on ozone are you referring to here?

*This part has been rewritten:*

*Ozone plays a crucial role in the chemical and radiative budget of the middle atmosphere. The distribution of ozone in the middle atmosphere is determined by both chemical and dynamical processes. Most of the ozone production takes place in the tropical stratosphere, as a result of photochemical processes, which involve oxygen. Meriodional circulation then transports ozone to other parts of the middle atmosphere (Langematz, 2019). The production of ozone is largely balanced by catalytic destruction cycles involving $NO_x$, $HO_x$ and $Cl_x$ radicals. $HO_x$ dominates ozone destruction in the mesosphere and lower stratosphere, while $NO_x$ and $Cl_x$ in the middle and upper stratosphere (Cariolle, 1983).*

*Since the 1970s ozone in the middle atmosphere began to decline globally, due to increased production of ozone depleting substances (ODSs). The Montreal Protocol, adopted in 1987 to stop this threat, eventually led to a slow recovery of the stratospheric ozone over the recent two decades. In our study, we don't consider the effect of anthropogenic ODSs since pre-industrial times (Langematz, 2019).*

*In this study, we are interested in the temperature response to changes in ozone concentration induced by the increased $CO_2$ concentration and/or the changes in SST in WACCM. Under enhanced $CO_2$ concentrations, the ratio between $O_3$ and $O$ mixing ratios is generally shifted toward a higher concentration of ozone, which is caused by the strong temperature dependency of the ozone production reaction ($O + O_2 + M \rightarrow O_3 + M$)*

**Major changes made to the manuscript:**

- Substantial rewriting of abstract, introduction, model and methods.
- Restructuring of section 3, 4 and 5.
- More quantification of our results in the text.
- Relating clearer to earlier studies and more emphasis on the fact that this is the first method to calculate the partial temperature changes at each point of the atmosphere.
- Clearer explanation of the atmospheric chemistry of WACCM and the changes in other chemical constituents due to changes in $CO_2$ and/or SSTs.

**Using the climate feedback response method to quantify climate feedbacks in the middle atmosphere in WACCM**

Maartje Sanne Kuilman[1], Qiong Zhang[2], Ming Cai[3], Qin Wen[1,4]

1. Department of Meteorology and Bolin Centre for Climate Research, Stockholm University, Stockholm, Sweden
2. Department of Physical Geography and Bolin Centre for Climate Research, Stockholm University, Stockholm, Sweden
3. Department of Earth, Ocean and Atmospheric Science, Florida State University, Tallahassee, Florida, USA
4. Laboratory for Climate and Ocean-Atmosphere Studies (LaCOAS), Department of Atmospheric and Oceanic Sciences, School of Physics, Peking University, Beijing, China

Corresponding author: Maartje Sanne Kuilman (maartje.kuilman@misu.su.se)

**Abstract**

Over recent decades it has become clear that the middle atmosphere has a significant impact on surface and tropospheric climate. A better understanding of the middle atmosphere and how it reacts to the current increase of the concentration of carbon dioxide ($CO_2$) is therefore necessary. In this study, we investigate the response to a doubling $CO_2$ in the middle atmosphere using the Whole Atmosphere Community Climate Model (WACCM). We use the climate feedback response analysis method (CFRAM) to calculate the partial temperature changes due to an external forcing and climate feedbacks in the atmosphere. As this method has the unique feature of additivity, these partial temperature changes are linearly addable. In this study, we discuss the direct forcing of $CO_2$ and the effects of the ozone, water vapour, cloud, albedo and dynamical feedbacks.

As expected, our results show that the direct forcing of $CO_2$ cools the middle atmosphere. This cooling becomes stronger with increasing height: the cooling in the upper stratosphere is about three times as strong as the cooling in the lower stratosphere. The ozone feedback yields a radiative feedback that mitigates this cooling in most regions of the middle atmosphere. However, in the tropical lower stratosphere and in some regions of the mesosphere, the ozone feedback has a cooling effect. The increase in $CO_2$-concentration causes the dynamics to change. The temperature response due to this dynamical feedback is small in the global average, although there are large temperature changes due to this feedback locally. The temperature change in the lower stratosphere is influenced by the water vapour feedback and to a lesser degree by the cloud and albedo feedback. These feedbacks play no role in the upper stratosphere and the mesosphere. We find that the effects of the changed SSTs on the middle atmosphere are relatively small as compared to the effects of changing the $CO_2$. However, the changes in SSTs are responsible for dynamical feedbacks that cause large temperature
* * *
**Margin annotations (tracked changes):**

Moved (insertion) [1]

¶
In a double $CO_2$ climate, the direct forcing of $CO_2$ would lead to a cooling which increases with increasing height in

¶
The ozone feedback yields a radiative feedback that generally mitigates this cooling. The dynamical feedback is another important feedback with large effects locally, while the effects of the water vapour feedback and especially the cloud and albedo feedbacks are only of importance in the lower stratosphere. ¶
¶
CFRAM is very powerful tool to study climate feedbacks in the middle atmosphere however, there is an error term caused by the linearization in the method.¶ ... [1]

Moved up [1]: Abstract¶

Formatted ... [3]

[revised manuscript text omitted]

**Moved down [29]:** ¶

**Moved down [30]:** We add the values for each

**Moved down [31]:** shows the radiative feedbacks

**Moved down [32]:** We also observe that there is a

**Moved down [33]:** ¶

**Moved down [34]:** ozone feedback generally yields a

**Moved down [35]:** With CFRAM, it is possible to

**Moved down [36]:** This can be understood as waves

**Moved down [41]:** The Non-LTE effects are also

**Moved down [37]:** ¶

**Moved down [38]:** ¶

**Moved down [39]:** We also observe that in the

**Moved down [40]:** In some places, this warming

**Moved down [42]:** by WACCM, the columns $CO_2$,

**Moved down [43]:** ¶

**Formatted** … [31]

[Figure]

(a) July ΔT CO₂: CO₂ high      (b) January ΔT CO₂: CO₂ high

[revised manuscript text omitted]
 R = \Delta F^{ext} + \Delta S + \Delta Q^{conv} + \Delta Q^{turb} - \Delta D^v - \Delta D^h + \Delta W^{fric} \qquad (A2)$$

In which the delta ($\Delta$) stands for the difference between the perturbation run and the control run.

CFRAM takes advantage of the fact that the infrared radiation is directly related to the temperatures in the entire column. The temperature changes in the equilibrium response to perturbations in the energy flux terms can be calculated. This is done by requiring that the temperature-induced changes in infrared radiation balance the non-temperature induced energy flux perturbations.

Equation (A2) can also be written as:

$$\Delta(S - R)_{total} + \Delta dyn = 0 \qquad (A3)$$

The term $\Delta(S - R)$ can be calculated as the longwave heating rate and the solar heating rate are output variables of the model simulations. We take the time mean of the WACCM data and perform the calculations for each grid point of the WACCM data. This means that in the end, we will have the temperature changes at each latitude, longitude and height.

We then calculate the difference in these heating rates for the perturbation simulation and the control simulation.

We use the term $\Delta(S - R)_{total}$ to calculate the dynamics term $\Delta dyn$.

$$\Delta dyn = -\Delta(S - R)_{total} \qquad (A4)$$

WACCM provides us with a heating rate in $Ks^{-1}$. For the CFRAM calculations, we need the energy flux in $Wm^{-2}$. We can calculate the energy flux by multiplying with the mass of different layers in the atmosphere and the specific heat capacity.

$$\Delta(S - R) = \Delta(S - R)_{(WACCM)} * mass_k * c_p \qquad (A5)$$

In which $\Delta(S-R)$ is the difference in the shortwave radiation $(S)$ and longwave radiation $(R)$ between the perturbation run and the control run as a flux in Wm⁻², while $\Delta(S-R)_{(WACCM)}$ is this difference as heating rate in Ks⁻¹ in WACCM, with $mass_k = \frac{p_{k+1} - p_k}{g}$ with p in Pa, $c_p = 1004$ J $kg^{-1} K^{-1}$ the specific heat capacity at constant pressure and $g$ the gravitational acceleration 9.81 $ms^{-2}$.

WACCM includes a non-local thermal equilibrium (non-LTE) radiation scheme above 50 km. It consists of a long-wave radiation (LW) part and a short-wave radiation (SW) part which includes the extreme ultraviolet (EUV) heating rate, chemical potential heating rate, $CO_2$ near-infrared (NIR) heating rate, total auroral heating rate and non-EUV photolysis heating rate.

Therefore, we split the term $\Delta(S-R)_{total}$ in an LTE and a non-LTE term:

$$\Delta(S-R)_{total} = \Delta(S-R)_{LTE} + \Delta(S-R)_{non-LTE} \qquad (A6)$$

WACCM provides us with the total longwave heating rate as well as the total solar heating rate and the non-LTE longwave and shortwave heating rates for the different runs. This means that we can calculate the term $\Delta(S-R)_{non-LTE}$ as well, where we again need to convert our result from Ks⁻¹ to Wm⁻²:

$$\Delta(S-R)_{non-LTE} = \Delta(S-R)_{non-LTE(WACCM)} \; mass_k * c_p \qquad (A7)$$

This term can be inserted in equation (3):

$$\Delta(S-R)_{LTE} + \Delta(S-R)_{non-LTE} + \Delta dyn = 0 \qquad (A8)$$

The central step in CFRAM is to decompose the radiative flux vector, using a linear approximation.

We start by decomposing the LTE infrared radiative flux vector $\Delta R$

$$\Delta R_{LTE} = \frac{\partial R}{\partial T}\Delta T + \Delta R_{CO_2} + \Delta R_{O_3} + \Delta R_{H_2O} + \Delta R_{albedo} + \Delta R_{cloud} \qquad (A9)$$

where $\Delta R_{CO_2}$, $\Delta R_{O_3}$, $\Delta R_{H_2O}$, $\Delta R_{albedo}$, $\Delta R_{cloud}$ are the changes in infrared radiative fluxes due to the changes in $CO_2$, ozone, water vapour, albedo and clouds, respectively.

For equation (A9), we assumed that radiative perturbations can be linearized by neglecting the higher order terms of each thermodynamic feedback and the interactions between these feedbacks. This is also commonly done in the PRP method (*Bony et al.,* 2006).

The term $\frac{\partial R}{\partial T}\Delta T$ represents the changes in the IR radiative fluxes related to the temperature changes in the entire atmosphere-surface column. The matrix $\frac{\partial R}{\partial T}$

is the Planck feedback matrix, in which the vertical profiles of the changes in the divergence of radiative energy fluxes due to a temperature change are represented.

We calculate this feedback matrix using the output variables of the perturbation and the control run of WACCM and inserting these in the CFRAM radiation code: atmospheric temperature, surface temperature, reference height temperature, ozone, surface pressure, solar insolation, downwelling solar flux at the surface, net solar flux at the surface, dew point temperature, cloud fraction, cloud ice amount, cloud liquid amount, ozone and specific humidity.

Similarly, the changes in the LTE shortwave radiation flux can be written as the sum of the change in shortwave radiation flux due to the direct forcing of $CO_2$ and the different feedbacks:

$$\Delta S_{LTE} = \Delta S_{CO_2} + \Delta S_{O_3} + \Delta S_{H_2O} + \Delta S_{albedo} + \Delta S_{cloud} \qquad (A10)$$

Similarly, to equation (A9), we perform a linearization.

Substituting (A9) and (A10) in equation (A8) yields:

$$\Delta(S-R)_{CO_2} + \Delta(S-R)_{O_3} + \Delta(S-R)_{H_2O} + \Delta(S-R)_{albedo} + \Delta(S-R)_{cloud} - \frac{\partial R}{\partial T}\Delta T$$
$$+\Delta(S-R)_{non-LTE} + \Delta dyn = 0 \qquad (A11)$$

This can be written as:

$$\Delta T = \left(\frac{\partial R}{\partial T}\right)^{-1} \left\{\Delta(S-R)_{CO_2} + \Delta(S-R)_{O_3} + \Delta(S-R)_{H_2O} + \Delta(S-R)_{albedo} + \right.$$
$$\left. \Delta(S-R)_{cloud} + \Delta(S-R)_{non-LTE} + \Delta dyn\right\} \qquad (
[revised manuscript text omitted]

| Page 1: [1] Deleted | Maartje Kuilman | 17/08/2020 22:31:00 |
|---|---|---|
| Page 1: [2] Deleted | Maartje Kuilman | 17/08/2020 22:31:00 |

| Page 1: [3] Formatted | Maartje Kuilman | 17/08/2020 22:31:00 |
|---|---|---|

Font colour: Auto, Not Superscript/ Subscript

| Page 2: [4] Deleted | Maartje Kuilman | 17/08/2020 22:31:00 |
|---|---|---|

| Page 7: [5] Formatted | Maartje Kuilman | 17/08/2020 22:31:00 |
|---|---|---|

x_body, Line spacing:  At least 12,65 pt, Widow/Orphan control, Adjust space between Latin and Asian text, Adjust space between Asian text and numbers

| Page 7: [6] Formatted | Maartje Kuilman | 17/08/2020 22:31:00 |
|---|---|---|

Font colour: Text 1

| Page 7: [7] Formatted | Maartje Kuilman | 17/08/2020 22:31:00 |
|---|---|---|

Font: 12 pt, Font colour: Black

| Page 7: [8] Formatted | Maartje Kuilman | 17/08/2020 22:31:00 |
|---|---|---|

Font: Italic

| Page 7: [9] Deleted | Maartje Kuilman | 17/08/2020 22:31:00 |
|---|---|---|

| Page 7: [10] Deleted | Maartje Kuilman | 17/08/2020 22:31:00 |
|---|---|---|

| Page 7: [11] Deleted | Maartje Kuilman | 17/08/2020 22:31:00 |
|---|---|---|

| Page 7: [12] Deleted | Maartje Kuilman | 17/08/2020 22:31:00 |
|---|---|---|

| Page 7: [13] Deleted | Maartje Kuilman | 17/08/2020 22:31:00 |
|---|---|---|

| Page 7: [14] Deleted | Maartje Kuilman | 17/08/2020 22:31:00 |
|---|---|---|
| Page 10: [15] Deleted | Maartje Kuilman | 17/08/2020 22:31:00 |
| Page 10: [16] Deleted | Maartje Kuilman | 17/08/2020 22:31:00 |

| Page 10: [17] Deleted | Maartje Kuilman | 17/08/2020 22:31:00 |
|---|---|---|

| Page 10: [18] Deleted | Maartje Kuilman | 17/08/2020 22:31:00 |
|---|---|---|

| Page 10: [18] Deleted | Maartje Kuilman | 17/08/2020 22:31:00 |
|---|---|---|

| Page 10: [19] Deleted | Maartje Kuilman | 17/08/2020 22:31:00 |
|---|---|---|

| Page 10: [20] Deleted | Maartje Kuilman | 17/08/2020 22:31:00 |

| Page 10: [21] Deleted | Maartje Kuilman | 17/08/2020 22:31:00 |

| Page 10: [22] Deleted | Maartje Kuilman | 17/08/2020 22:31:00 |

| Page 10: [22] Deleted | Maartje Kuilman | 17/08/2020 22:31:00 |

| Page 10: [23] Deleted | Maartje Kuilman | 17/08/2020 22:31:00 |

| Page 10: [24] Deleted | Maartje Kuilman | 17/08/2020 22:31:00 |

| Page 10: [25] Deleted | Maartje Kuilman | 17/08/2020 22:31:00 |

| Page 10: [26] Deleted | Maartje Kuilman | 17/08/2020 22:31:00 |

| Page 10: [27] Deleted | Maartje Kuilman | 17/08/2020 22:31:00 |

| Page 10: [28] Deleted | Maartje Kuilman | 17/08/2020 22:31:00 |

| Page 10: [28] Deleted | Maartje Kuilman | 17/08/2020 22:31:00 |

| Page 10: [29] Deleted | Maartje Kuilman | 17/08/2020 22:31:00 |

| Page 10: [30] Deleted | Maartje Kuilman | 17/08/2020 22:31:00 |

| Page 10: [30] Deleted | Maartje Kuilman | 17/08/2020 22:31:00 |

| Page 10: [31] Formatted | Maartje Kuilman | 17/08/2020 22:31:00 |

Font colour: Text 1

| Page 10: [31] Formatted | Maartje Kuilman | 17/08/2020 22:31:00 |

Font colour: Text 1

| Page 10: [32] Deleted | Maartje Kuilman | 17/08/2020 22:31:00 |

| Page 10: [32] Deleted | Maartje Kuilman | 17/08/2020 22:31:00 |

| Page 11: [33] Formatted | Maartje Kuilman | 17/08/2020 22:31:00 |

Font colour: Auto

| Page 11: [33] Formatted | Maartje Kuilman | 17/08/2020 22:31:00 |

Font colour: Auto

| Page 11: [34] Formatted | Maartje Kuilman | 17/08/2020 22:31:00 |

Font colour: Auto

| Page 11: [35] Formatted | Maartje Kuilman | 17/08/2020 22:31:00 |

Font colour: Auto

| Page 11: [36] Formatted | Maartje Kuilman | 17/08/2020 22:31:00 |

Font colour: Auto

| Page 11: [37] Deleted | Maartje Kuilman | 17/08/2020 22:31:00 |

| Page 11: [38] Formatted | Maartje Kuilman | 17/08/2020 22:31:00 |

Font colour: Auto

| Page 11: [39] Formatted | Maartje Kuilman | 17/08/2020 22:31:00 |

Font colour: Auto

| Page 11: [40] Formatted | Maartje Kuilman | 17/08/2020 22:31:00 |

Font colour: Auto

| Page 11: [41] Deleted | Maartje Kuilman | 17/08/2020 22:31:00 |

| Page 11: [41] Deleted | Maartje Kuilman | 17/08/2020 22:31:00 |

| Page 11: [42] Deleted | Maartje Kuilman | 17/08/2020 22:31:00 |

3.2

| Page 11: [43] Formatted | Maartje Kuilman | 17/08/2020 22:31:00 |

Normal (Web), Line spacing: single, Outline numbered + Level: 2 + Numbering Style: 1, 2, 3, … + Start at: 2 + Alignment: Left + Aligned at: 0 cm + Indent at: 0,71 cm

| Page 11: [44] Deleted | Maartje Kuilman | 17/08/2020 22:31:00 |

| Page 11: [45] Formatted | Maartje Kuilman | 17/08/2020 22:31:00 |

Normal (Web), Line spacing: single

| Page 11: [46] Deleted | Maartje Kuilman | 17/08/2020 22:31:00 |

| Page 11: [47] Deleted | Maartje Kuilman | 17/08/2020 22:31:00 |
|---|---|---|

| Page 11: [48] Deleted | Maartje Kuilman | 17/08/2020 22:31:00 |
|---|---|---|

| Page 11: [49] Deleted | Maartje Kuilman | 17/08/2020 22:31:00 |
|---|---|---|
| Page 11: [50] Deleted | Maartje Kuilman | 17/08/2020 22:31:00 |

| Page 11: [51] Formatted | Maartje Kuilman | 17/08/2020 22:31:00 |
|---|---|---|

Font: 12 pt

| Page 11: [51] Formatted | Maartje Kuilman | 17/08/2020 22:31:00 |
|---|---|---|

Font: 12 pt

| Page 11: [51] Formatted | Maartje Kuilman | 17/08/2020 22:31:00 |
|---|---|---|

Font: 12 pt

| Page 11: [52] Deleted | Maartje Kuilman | 17/08/2020 22:31:00 |
|---|---|---|
| Page 11: [52] Deleted | Maartje Kuilman | 17/08/2020 22:31:00 |
| Page 11: [52] Deleted | Maartje Kuilman | 17/08/2020 22:31:00 |
| Page 11: [52] Deleted | Maartje Kuilman | 17/08/2020 22:31:00 |
| Page 11: [52] Deleted | Maartje Kuilman | 17/08/2020 22:31:00 |
| Page 11: [52] Deleted | Maartje Kuilman | 17/08/2020 22:31:00 |
| Page 11: [52] Deleted | Maartje Kuilman | 17/08/2020 22:31:00 |
| Page 11: [53] Formatted | Maartje Kuilman | 17/08/2020 22:31:00 |

Font colour: Text 1

| Page 11: [53] Formatted | Maartje Kuilman | 17/08/2020 22:31:00 |
|---|---|---|

Font colour: Text 1

| Page 11: [54] Formatted | Maartje Kuilman | 17/08/2020 22:31:00 |
|---|---|---|

Font colour: Text 1

| Page 11: [55] Formatted | Maartje Kuilman | 17/08/2020 22:31:00 |
|---|---|---|

Font colour: Text 1

| Page 11: [56] Formatted | Maartje Kuilman | 17/08/2020 22:31:00 |
|---|---|---|

Font colour: Text 1

| Page 11: [56] Formatted | Maartje Kuilman | 17/08/2020 22:31:00 |
|---|---|---|

Font colour: Text 1

| Page 11: [56] Formatted | Maartje Kuilman | 17/08/2020 22:31:00 |
|---|---|---|

Font colour: Text 1

| Page 11: [57] Formatted | Maartje Kuilman | 17/08/2020 22:31:00 |
|---|---|---|

Font colour: Text 1

| Page 11: [58] Formatted | Maartje Kuilman | 17/08/2020 22:31:00 |
|---|---|---|

Font: Italic, Font colour: Text 1

| Page 11: [58] Formatted | Maartje Kuilman | 17/08/2020 22:31:00 |
|---|---|---|

Font: Italic, Font colour: Text 1

| Page 11: [59] Deleted | Maartje Kuilman | 17/08/2020 22:31:00 |
|---|---|---|

| Page 11: [60] Formatted | Maartje Kuilman | 17/08/2020 22:31:00 |
|---|---|---|

Normal (Web), Line spacing:  single

| Page 11: [61] Deleted | Maartje Kuilman | 17/08/2020 22:31:00 |
|---|---|---|

| Page 12: [62] Deleted | Maartje Kuilman | 17/08/2020 22:31:00 |
|---|---|---|
| Page 12: [62] Deleted | Maartje Kuilman | 17/08/2020 22:31:00 |
| Page 12: [63] Formatted | Maartje Kuilman | 17/08/2020 22:31:00 |

English (US)

| Page 12: [63] Formatted | Maartje Kuilman | 17/08/2020 22:31:00 |
|---|---|---|

English (US)

| Page 12: [64] Deleted | Maartje Kuilman | 17/08/2020 22:31:00 |
|---|---|---|

| Page 12: [64] Deleted | Maartje Kuilman | 17/08/2020 22:31:00 |
|---|---|---|

| Page 12: [65] Deleted | Maartje Kuilman | 17/08/2020 22:31:00 |
|---|---|---|

| Page 12: [65] Deleted | Maartje Kuilman | 17/08/2020 22:31:00 |
|---|---|---|

| Page 12: [66] Formatted | Maartje Kuilman | 17/08/2020 22:31:00 |
|---|---|---|

Font colour: Auto

| Page 12: [66] Formatted | Maartje Kuilman | 17/08/2020 22:31:00 |
|---|---|---|

Font colour: Auto

| Page 12: [67] Formatted | Maartje Kuilman | 17/08/2020 22:31:00 |
|---|---|---|

Font colour: Auto

| Page 12: [67] Formatted | Maartje Kuilman | 17/08/2020 22:31:00 |
|---|---|---|

Font colour: Auto

| Page 12: [68] Formatted | Maartje Kuilman | 17/08/2020 22:31:00 |
|---|---|---|

Font colour: Auto

| Page 12: [68] Formatted | Maartje Kuilman | 17/08/2020 22:31:00 |
|---|---|---|

Font colour: Auto

| Page 12: [68] Formatted | Maartje Kuilman | 17/08/2020 22:31:00 |
|---|---|---|

Font colour: Auto

| Page 12: [68] Formatted | Maartje Kuilman | 17/08/2020 22:31:00 |
|---|---|---|

Font colour: Auto

| Page 12: [69] Formatted | Maartje Kuilman | 17/08/2020 22:31:00 |
|---|---|---|

Font colour: Auto

| Page 12: [69] Formatted | Maartje Kuilman | 17/08/2020 22:31:00 |
|---|---|---|

Font colour: Auto

| Page 12: [70] Deleted | Maartje Kuilman | 17/08/2020 22:31:00 |
|---|---|---|

| Page 12: [70] Deleted | Maartje Kuilman | 17/08/2020 22:31:00 |
|---|---|---|

| Page 12: [70] Deleted | Maartje Kuilman | 17/08/2020 22:31:00 |
|---|---|---|

| Page 12: [71] Deleted | Maartje Kuilman | 17/08/2020 22:31:00 |
|---|---|---|

---

## Editor Decision (ED2)

The abstract and the manuscript reads now much better. On the abstract I have the following two comments (P1, L46):

(1) Abbreviation SST has not been introduced.
(2) Further, it has not been mentioned anywhere yet that changed SSTs have been considered. Either you add a sentence here or you add that the SSTs have been changed as well in line 32 where you write what is done is this study.

On the main text I have the following comments:

P5, L218: What do you mean with the latter is not considered in this study? No chemistry is considered!? Please rephrase the sentence to be clearer.

P5, L229-230: This sentence is also very weird. Why do they do not play a role in your experiments? The question from the referee was quite simple and could have been answered with yes or no. Either CFCs are considered or not. Since you use pre-industrial conditions I would assume that these are either zero or low since to my knowledge anthropogenic production (and thus the increase) of CFCs started later than 1850. This is something which can easily be checked. So please check and rephrase the sentence accordingly.

P6, Table 1: The naming in the table is quite misleading and I needed a while to understand how your four experiments differentiate from each other. I would suggest to do the following change (and simplification) to the table to be clearer:

| Experiment | CO2 | SSTs |
|---|---|---|
| C1 | PI | PI |
| C2 | double | PI |
| S1 | PI | double (or high) |
| S2 | double | double (or high) |

Further, it would be more logical if the naming would be C1, S1, S2 and S3 since you have one control run and three scenarios.

P7, L298-312: This text part is also quite difficult to read. I would suggest that you repeat here the sentences from the appendix what R, S and the deltas are. Further, I would suggest to move the second paragraph to the appendix or to make a list with bullets for CO2, H2O, O3, cloud and Albedo where then is written what has been done for each species or you just simplify the text. In general the difference between control and experiment is considered, so just mention then what is the difference for the species. In your text especially the phrase "the other variables" is not clear and rather confusing.

P9, 392: add "(C1)" after control simulation.

P14, Figure 5 caption: Shouldn't it read "S1 - C1" and "C2 - C1", respectively? The same holds for Fig7, Fig 9, Fig. 10, Fig 11 and Fig 13.

P18, L697: Be more precise than "higher up". Above which level?

P24, L870: Sentence not clear. Please rephrase. Which exact mechanism are you talking about? What exactly is out of the scope of the paper? Do you mean to investigate the exact mechanism why the non-LTE effects are small is outside the scope of the paper?

P26, L897: Which errors? Do you mean the bars in the bar chart of Figure 13?

P30, L1122: What is the PRP method? Please add more details.

**Other technical issues:**

P2, L64: the -> this

P3, L122: these -> "the" or "the respective"

P3, L126: use plural, thus "forcing and feedbacks"

P3, L143: such idealized -> such "an" idealized

P6, L264: effect -> affect

P9, L391: add "to" before "the"

P10, L433: Past should be used here: "In this section, we have discussed……."

P10, L444: add "is found" after CO2-concentration.

P11, L452: singular should be used here -> run

P13, L536: add "by" so that it reads "could be caused by an increase…."

P14, L545: add "the" so that it reads "….due to the……".

P15, L589: add "scale" so that it reads "……..synoptic scale waves"

P16, L645: add "concentration" after CO2.

P17, L652: add "to" before "the" so that it reads "…with respect to the….."

P18, L698: add "vapour" so that it reads "water vapour".

P20, L759: stratosphere -> stratospheric

P20, L760: until -> up to

P21, L771: add "average" so that it reads "by taking the average in this way".

P22, L814: add "is" so that it reads "is a bit smaller".

P22, L822-824: "before" appears in the sentence twice. I would suggest to delete the latter one.

P22, L832: move "mostly" before "contributes" so that it reads "which mostly contributes…".

P24, L874: "for due" is not correct, this "for" is obsolete here. Please remove.

P26, L890: add "the" so that it reads "shows the"

P26, L931-932: "before" twice. One is therefore obsolete. I would suggest to remove the latter one.

---

## Author Response (AR3)

The abstract and the manuscript reads now much better.

We thank the editor for her positive feedback and for her further comments.

On the abstract I have the following two comments (P1, L46):
(1) Abbreviation SST has not been introduced.
(2) Further, it has not been mentioned anywhere yet that changed SSTs have been considered. Either you add a sentence here or you add that the SSTs have been changed as well in line 32 where you write what is done is this study.

This has been implemented:

*"In this study, we investigate the response of the middle atmosphere to a doubling of the $CO_2$-concentration and the associated changes in sea surface temperatures (SSTs) using the Whole Atmosphere Community Climate Model (WACCM)."*

On the main text I have the following comments:
P5, L218: What do you mean with the latter is not considered in this study? No chemistry is considered!? Please rephrase the sentence to be clearer.
This sentence has been rephrased:

*"The SSTs might be slightly different when they would be generated using a model that also includes atmospheric chemistry, however, this aspect is not considered in this study."*

P5, L229-230: This sentence is also very weird. Why do they do not play a role in your experiments? The question from the referee was quite simple and could have been answered with yes or no. Either CFCs are considered or not. Since you use pre-industrial conditions I would assume that these are either zero or low since to my knowledge anthropogenic production (and thus the increase) of CFCs started later than 1850. This is something which can easily be checked. So please check and rephrase the sentence accordingly.
It has been checked the CFC are zero (there are some values $10^{-22}$ mol/mol, but mostly zeros), this also doesn't change between the control run and the perturbation runs. So, the short answer to the reviewer's question would be no, these are not considered.

The text has now been rewritten:

*"The compset used in this experiment and all the following ones is still F_1850, which means that other radiatively and chemically active gases, such as ozone, will change only because of the changes in the $CO_2$-concentration, due to WACCM's interactive chemistry. This also means that the effects of chlorofluorocarbons (CFCs) are not considered in our experiments, as anthropogenic production of CFCs started later than 1850."*

P6, Table 1: The naming in the table is quite misleading and I needed a while to understand how your four experiments differentiate from each other. I would suggest to do the following change (and simplification) to the table to be clearer:
Experiment CO2 SSTs
C1 PI PI
C2 double PI
S1 PI double (or high)
S2 double double (or high)
*The simplification has been implemented.*

Further, it would be more logical if the naming would be C1, S1, S2 and S3 since you have one control run and three scenarios.
*This has been implemented.*

P7, L298-312: This text part is also quite difficult to read. I would suggest that you repeat here the sentences from the appendix what R, S and the deltas are.
*This has been implemented.*
*"In which $\vec{R}$ represents the vertical profile of the net long-wave radiation emitted by each layer in the atmosphere and by the surface. $\vec{S}$ is the vertical profile of the solar radiation absorbed by each layer. The matrix $\left(\frac{\partial \vec{R}}{\partial \vec{T}}\right)$*
*is the Planck feedback matrix, in which the vertical profiles of the changes in the divergence of radiative energy fluxes due to a temperature change are represented. $\Delta T$ represents the temperature change."*

Further, I would suggest to move the second paragraph to the appendix or to make a list with bullets for CO2, H2O, O3, cloud and Albedo where then is written what has been done for each species or you just simplify the text. In general the difference between control and experiment is considered, so just mention then what is the difference for the species. In your text especially the phrase "the other variables" is not clear and rather confusing.
*The authors agree that the former formulation was confusing. The following has been added to the appendix:*

*The factors $\Delta(\vec{S} - \vec{R})_{CO_2}$ , $\Delta(\vec{S} - \vec{R})_{O_3}$, $\Delta(\vec{S} - \vec{R})_{H_2O}$ , $\Delta(\vec{S} - \vec{R})_{albedo}$ and $\Delta(\vec{S} - \vec{R})_{cloud}$ in eqs (1-5) are calculated by inserting the output variables from WACCM in the radiation code of CFRAM. Here, one takes the output variables from the control run, apart from the variable that is related to the direct forcing or the feedback. The table below shows which variables have been taken from the perturbation runs for each feedback.*

| Direct forcing/feedback | Changed variables in the radiation code |
|---|---|
| $CO_2$ | $CO_2$ |
| Ozone | $O_3$ |
| Water vapour | Specific humidity
Surface pressure
Surface temperature
Dew point temperature |
| Albedo | Downwelling solar flux at surface |

| | Net solar flux at surface |
|---|---|
| Cloud | Cloud fraction |
| | Cloud ice |
| | Cloud liquid amount |

*Table A1: The variables from the perturbation runs inserted in the radiation code of CFRAM to calculate the temperature change in response to the changes in $CO_2$, $O_3$, water vapour, cloud and albedo.*

The main text now refers to the appendix in the following way:

*Here, one takes the output variables from the control run, apart from the variable that is related to the direct forcing or the feedback. Table A1 in the Appendix shows which variables from the perturbation runs have been inserted in the radiation code of CFRAM in order to calculate $\Delta(\vec{S} - \vec{R})_{CO_2}$, $\Delta(\vec{S} - \vec{R})_{O_3}$, $\Delta(\vec{S} - \vec{R})_{H_2O}$, $\Delta(\vec{S} - \vec{R})_{albedo}$ and $\Delta(\vec{S} - \vec{R})_{cloud}$ and eventually the associated temperature changes.*

P9, 392: add "(C1)" after control simulation.
This has been added (also for the other simulations):

*"Figure 1: The total change in temperature in July (top) and January (bottom) for (a,d) the simulation with high $CO_2$ and SSTs (S3), (b,e) the simulation with high $CO_2$ (S1), (c,f) the simulation with high SSTs (S2), all as compared to the pre-industrial control simulation (C1). The dotted regions indicate the regions where the data reaches a confidence level of 95%. The black line indicates the tropopause height for the experiments S3 (a,d), S1 (b,e) and S2 (c,f)."*

P14, Figure 5 caption: Shouldn't it read "S1 - C1" and "C2 - C1", respectively? The same holds for Fig7, Fig 9, Fig. 10, Fig 11 and Fig 13.
Thanks for this comment, this was indeed incorrect. It has now been corrected.

P18, L697: Be more precise than "higher up". Above which level?
This has been clarified:

*"Fig. 8 shows that above 1 hPa, there are also large percentage changes in water vapour. However, the absolute concentration of water vapour is small there, which explains why there is no temperature response to these changes."*

P24, L870: Sentence not clear. Please rephrase. Which exact mechanism are you talking about? What exactly is out of the scope of the paper? Do you mean to investigate the exact mechanism why the non-LTE effects are small is outside the scope of the paper?
This has been rewritten:
*"We find that there are also some small temperature changes due to non-LTE effect above 0.1 hPa. How the non-LTE effects exactly cause the small temperature changes in this region is outside the scope of this paper and needs further investigation."*

P26, L897: Which errors? Do you mean the bars in the bar chart of Figure 13?

Yes, that is correct. I have made this clearer in the text.

*"As in Fig. 11, the 'Error'-column in Fig. 13 shows the difference between temperature change in WACCM and the sum of the calculated temperature responses in CFRAM."*

We have also clarified this bit in the description in Fig. 11:

*"This term shows the difference between temperature change in WACCM and the sum of the calculated temperature responses in CFRAM (see eq. 9 in section 2.3)."*

P30, L1122: What is the PRP method? Please add more details.

This method and abbreviation had introduced in the introduction. Now more details have been given at this point in the paper as well:

*"This is also commonly done in the partial radiative perturbation (PRP) method, in which partial derivatives of the model top of the atmosphere radiation are evaluated with respect to changes in model parameters by diagnostic rerunning the model's radiation code (Bony et al., 2006)."*

**Other technical issues:**

P2, L64: the -> this
This has been corrected.

P3, L122: these -> "the" or "the respective"
This has been corrected.

P3, L126: use plural, thus "forcing and feedbacks"
This has been corrected.

P3, L143: such idealized -> such "an" idealized
This has been corrected.

P6, L264: effect -> affect
This has been corrected.

P9, L391: add "to" before "the"
This has been corrected.

P10, L433: Past should be used here: "In this section, we have discussed......."
It was meant to be in the current section (4), we will discuss. This has now been changed in the text to make it clearer:

*"In the following subsections 4.1-4.5, we will discuss the meridional-vertical profiles of the temperature responses to the direct forcing and the various feedbacks during*

P10, L444: add "is found" after CO2-concentration.
This has been added.

P11, L452: singular should be used here -> run
This has been corrected.

P13, L536: add "by" so that it reads "could be caused by an increase…."
This has been corrected.

P14, L545: add "the" so that it reads "….due to the……".
This has been corrected.

P15, L589: add "scale" so that it reads "……..synoptic scale waves"
This has been added.

P16, L645: add "concentration" after CO2.
This has been added.

P17, L652: add "to" before "the" so that it reads "…with respect to the….."
This has been added.

P18, L698: add "vapour" so that it reads "water vapour".
This has been added.

P20, L759: stratosphere -> stratospheric
This has been corrected.

P20, L760: until -> up to
This has been changed.

P21, L771: add "average" so that it reads "by taking the average in this way".
This has been changed:

*"By calculating the average in this way, we can directly compare the vertical values in different regions of the atmosphere."*

P22, L814: add "is" so that it reads "is a bit smaller".
This has been added.

P22, L822-824: "before" appears in the sentence twice. I would suggest to delete the latter one.
The latter 'before' has been removed.

P22, L832: move "mostly" before "contributes" so that it reads "which mostly contributes…".
'Mostly' has been moved.

P24, L874: "for due" is not correct, this "for" is obsolete here. Please remove.
'For' has been removed.

P26, L890: add "the" so that it reads "shows the"
'The' has been added.

P26, L931-932: "before" twice. One is therefore obsolete. I would suggest to remove the latter one.
'Before' has been removed.

[revised manuscript text omitted]

$$\quad \Delta T_{CO_2} = \left(\frac{\partial R}{\partial T}\right)^{-1} \Delta(S - R)_{CO_2} \tag{1}$$

$$\quad \Delta T_{O_3} = \left(\frac{\partial R}{\partial T}\right)^{-1} \Delta(S - R)_{O_3} \tag{2}$$

$$\quad \Delta T_{H_2O} = \left(\frac{\partial R}{\partial T}\right)^{-1} \Delta(S - R)_{H_2O} \tag{3}$$

$$\quad \Delta T_{albedo} = \left(\frac{\partial R}{\partial T}\right)^{-1} \Delta(S - R)_{albedo} \tag{4}$$

$$\quad \Delta T_{cloud} = \left(\frac{\partial R}{\partial T}\right)^{-1} \Delta(S - R)_{cloud} \tag{5}$$

In which $R$ represents the vertical profile of the net long-wave radiation
emitted by each layer in the atmosphere and by the surface. $S$ is the vertical
profile of the solar radiation absorbed by each layer. The matrix $\left(\frac{\partial R}{\partial T}\right)$
is the Planck feedback matrix, in which the vertical profiles of the changes in
the divergence of radiative energy fluxes due to a temperature change are
represented. $\Delta T$ represents the temperature change.
The factors $\Delta(S - R)_{CO_2}$ , $\Delta(S - R)_{O_3}$, $\Delta(S - R)_{H_2O}$ , $\Delta(S - R)_{albedo}$ and
$\Delta(S - R)_{cloud}$ are calculated by inserting the output variables from WACCM in
the radiation code of CFRAM. Here, one takes the output variables from the
control run, apart from the variable that is related to the direct forcing or the
feedback. Table A1 in the Appendix shows which variables from the
perturbation runs have been inserted in the radiation code of CFRAM in order to calculate $\Delta(S-R)_{CO_2}$, $\Delta(S-R)_{O_3}$, $\Delta(S-R)_{H_2O}$, $\Delta(S-R)_{albedo}$ and $\Delta(S-
[revised manuscript text omitted]
 R = \Delta F^{ext} + \Delta S + \Delta Q^{conv} + \Delta Q^{turb} - \Delta D^v - \Delta D^h + \Delta W^{fric} \qquad \text{(A2)}$$
In which the delta ($\Delta$) stands for the difference between the perturbation run
and the control run.
CFRAM takes advantage of the fact that the infrared radiation is directly
related to the temperatures in the entire column. The temperature changes in
the equilibrium response to perturbations in the energy flux terms can be
calculated. This is done by requiring that the temperature-induced changes in
infrared radiation balance the non-temperature induced energy flux
perturbations.
Equation (A2) can also be written as:
$$\Delta(S-R)_{total} + \Delta dyn = 0 \qquad \text{(A3)}$$
The term $\Delta(S-R)$ can be calculated as the longwave heating rate and the
solar heating rate are output variables of the model simulations. We take the
time mean of the WACCM data and perform the calculations for each grid
point of the WACCM data. This means that in the end, we will have the
temperature changes at each latitude, longitude and height.
We then calculate the difference in these heating rates for the perturbation
simulation and the control simulation.
We use the term $\Delta(S-R)_{total}$ to calculate the dynamics term $\Delta dyn$.
$$\Delta dyn = -\Delta(S-R)_{total} \qquad \text{(A4)}$$
WACCM provides us with a heating rate in $Ks^{-1}$. For the CFRAM calculations,
we need the energy flux in $Wm^{-2}$. We can calculate the energy flux by
multiplying with the mass of different layers in the atmosphere and the specific
heat capacity.
$$\Delta(S-R) = \Delta(S-R)_{(WACCM)} * mass_k * c_p \qquad \text{(A5)}$$

In which $\Delta(S - R)$ is the difference in the shortwave radiation $(S)$ and
longwave radiation $(R)$ between the perturbation run and the control run as a
flux in Wm$^{-2}$, while $\Delta(S - R)_{(WACCM)}$ is this difference as heating rate in Ks$^{-1}$ in
WACCM, with $mass_k = \frac{p_{k+1} - p_k}{g}$ with p in Pa, $c_p = 1004$ J $kg^{-1} K^{-1}$ the
specific heat capacity at constant pressure and $g$ the gravitational
acceleration 9.81 $ms^{-2}$.
WACCM includes a non-local thermal equilibrium (non-LTE) radiation scheme
above 50 km. It consists of a long-wave radiation (LW) part and a short-wave
radiation (SW) part which includes the extreme ultraviolet (EUV) heating rate,
chemical potential heating rate, $CO_2$ near-infrared (NIR) heating rate, total
auroral heating rate and non-EUV photolysis heating rate.
Therefore, we split the term $\Delta(S - R)_{total}$ in an LTE and a non-LTE term:
$$\Delta(S - R)_{total} = \Delta(S - R)_{LTE} + \Delta(S - R)_{non-LTE} \quad\quad\quad (A6)$$
WACCM provides us with the total longwave heating rate as well as the total
solar heating rate and the non-LTE longwave and shortwave heating rates for
the different runs. This means that we can calculate the term $\Delta(S - R)_{non-LTE}$
as well, where we again need to convert our result from Ks$^{-1}$ to Wm$^{-2}$:
$$\Delta(S - R)_{non-LTE} = \Delta(S - R)_{non-LTE(WACCM)} \; mass_k * c_p \quad\quad (A7)$$
This term can be inserted in equation (3):
$$\Delta(S - R)_{LTE} + \Delta(S - R)_{non-LTE} + \Delta dyn = 0 \quad\quad\quad (A8)$$
The central step in CFRAM is to decompose the radiative flux vector, using a
linear approximation.
We start by decomposing the LTE infrared radiative flux vector $\Delta R$
$$\Delta R_{LTE} = \frac{\partial R}{\partial T}\Delta T + \Delta R_{CO_2} + \Delta R_{O_3} + \Delta R_{H_2O} + \Delta R_{albedo} + \Delta R_{cloud} \quad\quad (A9)$$
where $\Delta R_{CO_2}$, $\Delta R_{O_3}$, $\Delta R_{H_2O}$, $\Delta R_{albedo}$, $\Delta R_{cloud}$ are the changes in infrared
radiative fluxes due to the changes in $CO_2$, ozone, water vapour, albedo and
clouds, respectively.
For equation (A9), we assumed that radiative perturbations can be linearized
by neglecting the higher order terms of each thermodynamic feedback and
the interactions between these feedbacks. This is also commonly done in the
partial radiative perturbation (PRP) method, in which partial derivatives of the
model top of the atmosphere radiation are evaluated with respect to changes
in model parameters by diagnostic rerunning the model's radiation code (*Bony*
*et al.,* 2006).

The term $\frac{\partial R}{\partial T}\Delta T$ represents the changes in the IR radiative fluxes related to the temperature changes in the entire atmosphere-surface column. The matrix $\frac{\partial R}{\partial T}$

is the Planck feedback matrix, in which the vertical profiles of the changes in the divergence of radiative energy fluxes due to a temperature change are represented.

We calculate this feedback matrix using the output variables of the perturbation and the control run of WACCM and inserting these in the CFRAM

radiation code: atmospheric temperature, surface temperature, reference height temperature, ozone, surface pressure, solar insolation, downwelling solar flux at the surface, net solar flux at the surface, dew point temperature, cloud fraction, cloud ice amount, cloud liquid amount, ozone and specific humidity.

Similarly, the changes in the LTE shortwave radiation flux can be written as the sum of the change in shortwave radiation flux due to the direct forcing of

$CO_2$ and the different feedbacks:

$\Delta S_{LTE} = \Delta S_{CO_2} + \Delta S_{O_3} + \Delta S_{H_2O} + \Delta S_{albedo} + \Delta S_{cloud}$ (A10)

Similarly, to equation (A9), we perform a linearization.

Substituting (A9) and (A10) in equation (A8) yields:

$\Delta(S-R)_{CO_2} + \Delta(S-R)_{O_3} + \Delta(S-R)_{H_2O} + \Delta(S-R)_{albedo} + \Delta(S-R)_{cloud} - \frac{\partial R}{\partial T}\Delta T$

$+\Delta(S-R)_{non-LTE} + \Delta dyn = 0$ (A11)

This can be written as:

$\Delta T = \left(\frac{\partial R}{\partial T}\right)^{-1} \{\Delta(S-R)_{CO_2} + \Delta(S-R)_{O_3} + \Delta(S-R)_{H_2O} + \Delta(S-R)_{albedo} + $

$\Delta(S-R)_{cloud} + \Delta(S-R)_{non-LTE} + \Delta dyn\}$ (A12)

As described in the main text of this paper, we can solve Eq. (A12) for each of the terms on its right-hand side, based on the linear decomposition principle.

This yields the partial temperature changes due to each specific process. The factors $\Delta(S-R)_{CO_2}$, $\Delta(S-R)_{O_3}$, $\Delta(S-R)_{H_2O}$, $\Delta(S-R)_{albedo}$ and

[revised manuscript text omitted]